# Dynamics of fMRI patterns reflect sub-second activation sequences and reveal replay in human visual cortex

Lennart Wittkuhn [1,2✉] & Nicolas W. Schuck [1,2✉]

Neural computations are often fast and anatomically localized. Yet, investigating such computations in humans is challenging because non-invasive methods have either high temporal or spatial resolution, but not both. Of particular relevance, fast neural replay is known to occur throughout the brain in a coordinated fashion about which little is known. We develop a multivariate analysis method for functional magnetic resonance imaging that makes it possible to study sequentially activated neural patterns separated by less than 100 ms with precise spatial resolution. Human participants viewed five images individually and sequentially with speeds up to 32 ms between items. Probabilistic pattern classifiers were trained on activation patterns in visual and ventrotemporal cortex during individual image trials. Applied to sequence trials, probabilistic classifier time courses allow the detection of neural representations and their order. Order detection remains possible at speeds up to 32 ms between items (plus 100 ms per item). The frequency spectrum of the sequentiality metric distinguishes between sub- versus supra-second sequences. Importantly, applied to resting-state data our method reveals fast replay of task-related stimuli in visual cortex. This indicates that non-hippocampal replay occurs even after tasks without memory requirements and shows that our method can be used to detect such spontaneously occurring replay.

[1] Max Planck Research Group NeuroCode, Max Planck Institute for Human Development, Berlin, Germany. [2] Max Planck UCL Centre for Computational Psychiatry and Ageing Research, Berlin, Germany. ✉email: wittkuhn@mpib-berlin.mpg.de; schuck@mpib-berlin.mpg.de

Many cognitive processes are underpinned by rapidly changing neural activation patterns. Most famously, memory and planning have been linked to fast replay of representation sequences in the hippocampus, happening approximately within 200–300 milliseconds (ms) while the animal is resting or sleeping, e.g.[1–9]. Similar events have been observed during behavior[10,11], as well as outside of the hippocampus[12–17]. Likewise, internal deliberations during choice are reflected in alternations between orbitofrontal value representations that last less than 100 ms[18], while perceptual learning has been shown to result in sub-second anticipatory activation sequences in visual cortex[19–21]. Investigating fast-paced representational dynamics within specific brain areas therefore promises important insights into a variety of cognitive processes. Such investigations could be crucial for understanding replay, which is characterized by a widespread co-occurrence of neural reactivation events throughout the brain of mostly unknown functional significance, in particular outside of the hippocampus, see, e.g.[17,22]. These aspects are still understudied in humans.

Studying fast neural dynamics is particularly difficult in humans because signal recording must mainly occur non-invasively. How fast and anatomically localized neural dynamics can be investigated using non-invasive neuroimaging techniques is therefore a major challenge for human neuroscience, see, e.g.[23,24]. The main concern related to functional magnetic resonance imaging (fMRI) is that this technique measures neural activity indirectly through slow sampling of an extended and delayed blood-oxygen-level-dependent (BOLD) response function[25–27] that can obscure temporal detail. Yet, the problems arising in BOLD fMRI might not be as insurmountable as they seem. First, BOLD signals from the same participant and brain region show reliable timing and last for several seconds. Miezin et al.[28], for instance, reported a between-session reliability of hemodynamic peak times in visual cortex of $r^2 = 0.95$, see also[29,30]. Even for closely timed events, the sequential order can therefore result in systematic differences in activation strength[31] that remain in the signal long after the fast sequence event is over, effectively mitigating the problems that arise from slow sampling. Moreover, Misaki et al.[32] were able to decode onset differences in visual stimulation of only 100 ms when two stimuli were shown to one eye before the other. Interestingly, Misaki et al.[32] indicated that timing differences become most apparent in peak activation strength, rather than temporal aspects of the hemodynamic response function (HRF). A second reason that makes the investigation of fast neural dynamics feasible is that some fast sequence events have properties that make it easier to detect them. Replay events, in particular, involve reactivation of spatially tuned cells in the order of a previously traveled path. But these reactivated paths do not typically span the entire spatial environment and only involve a local subset of all possible places the animal could occupy[7,8]. This locality means that even when measurement noise causes some elements of a fast sequence to remain undetected, or leads to partially re-ordered detection, the set of detected representations will still reflect positions nearby in space. In this case, successive detection of elements nearby in space or time would still identify the fast process under investigation even under noisy conditions.

If fMRI analyses can capitalize on such effects, this could allow the investigation of fast sequential activations. As mentioned above, one important application of such methods would be hippocampal replay, a topic of intense recent interest, for reviews, see, e.g.[24,33–37]. To date, most replay research has studied the phenomenon in rodents because investigations in humans and other primates either required invasive recordings from the hippocampus[38–42], used techniques with reduced hippocampal sensitivity and spatial resolution[43–48], or investigated non-sequential fMRI activation patterns over seconds or minutes[49–53]. Recently, we have hypothesized that the properties of BOLD signals mentioned above should enable the investigation of rapid neural dynamics. Indeed, using fMRI, we identified fast sequential hippocampal pattern reactivation in resting humans[54]. However, Schuck and Niv[54] did not yet answer questions about how fMRI could be used to measure the speed of replay. One additional exploratory question is whether replay occurs outside of the hippocampus, and even following simple visual detection tasks.

Here, we provide and experimentally validate a multivariate analysis approach for fMRI that addresses the challenges and questions outlined above. The main idea of our approach is that fast neural event sequences will cause characteristic time courses of overlapping activation patterns. While the effects of co-occurring activations on individual voxels is complex, we reason that characteristic overlap will nevertheless lead to predictable and simple fluctuations in the time courses of pattern classifiers. The present experiment tests this idea and our results confirm that logistic regression classifier time courses reveal the content and order of fast sequential neural events using fMRI. Importantly, we use this method to ask whether sequential reactivations of sensory events occur outside of the hippocampus, even if task experiences did not require memorization or involve repeated sequential structure. Our study extends our previous work in several ways. First, our controlled experimental design provides evidence for the decodability of fast sequential neural events in a setting where the speed and order of fast neural event sequences are known. We also show that sequence detection can be achieved in the presence of high levels of signal noise and timing uncertainty, and is specific enough to differentiate fast sequences from activation patterns that could reflect slow conscious thinking. Second, we develop a modeling approach of multivariate fMRI pattern classification time courses that validates our experimental results and allows inference of the speed of fast sequential neural processes from the frequency spectra of our fMRI sequentiality metric. Third, we report that our task induced fast sequential replay in sensory brain areas during post-task rest, although it did not require any memorization, did not feature strong sequential structure, and did not elicit systematic hippocampal responses. Finally, our results have implications for the interpretation of our own previous results in Schuck and Niv[54] and future fMRI studies investigating fast neural event sequences, like hippocampal replay.

## Results

As discussed above, we investigated the possibility that fMRI can be used to address two cornerstones of understanding signals resulting from fast activation sequences: *order detection* and *element detection*. The first effect, order detection, pertains to the presence of order structure in the signal that is caused by the sequential order of fast neural events. We evaluated this effect by investigating the impact of item order on (a) the relative strength of activations within a single measurement, and (b) the order of decoded patterns across successive measurements. The second effect, element detection, quantifies to what extent fMRI allows detection of elements that were part of a sequence versus those that were not. While event detection is a standard problem in fMRI, we focused on the special case relevant to our question: detecting neural patterns of brief events that are affected by patterns from other sequence elements occurring only tens of milliseconds before or afterwards, causing backward and forward interference, respectively. Using full sequences of all possible elements in our experimental setup that tested sequence ordering, our design ensured that the two effects can be demonstrated

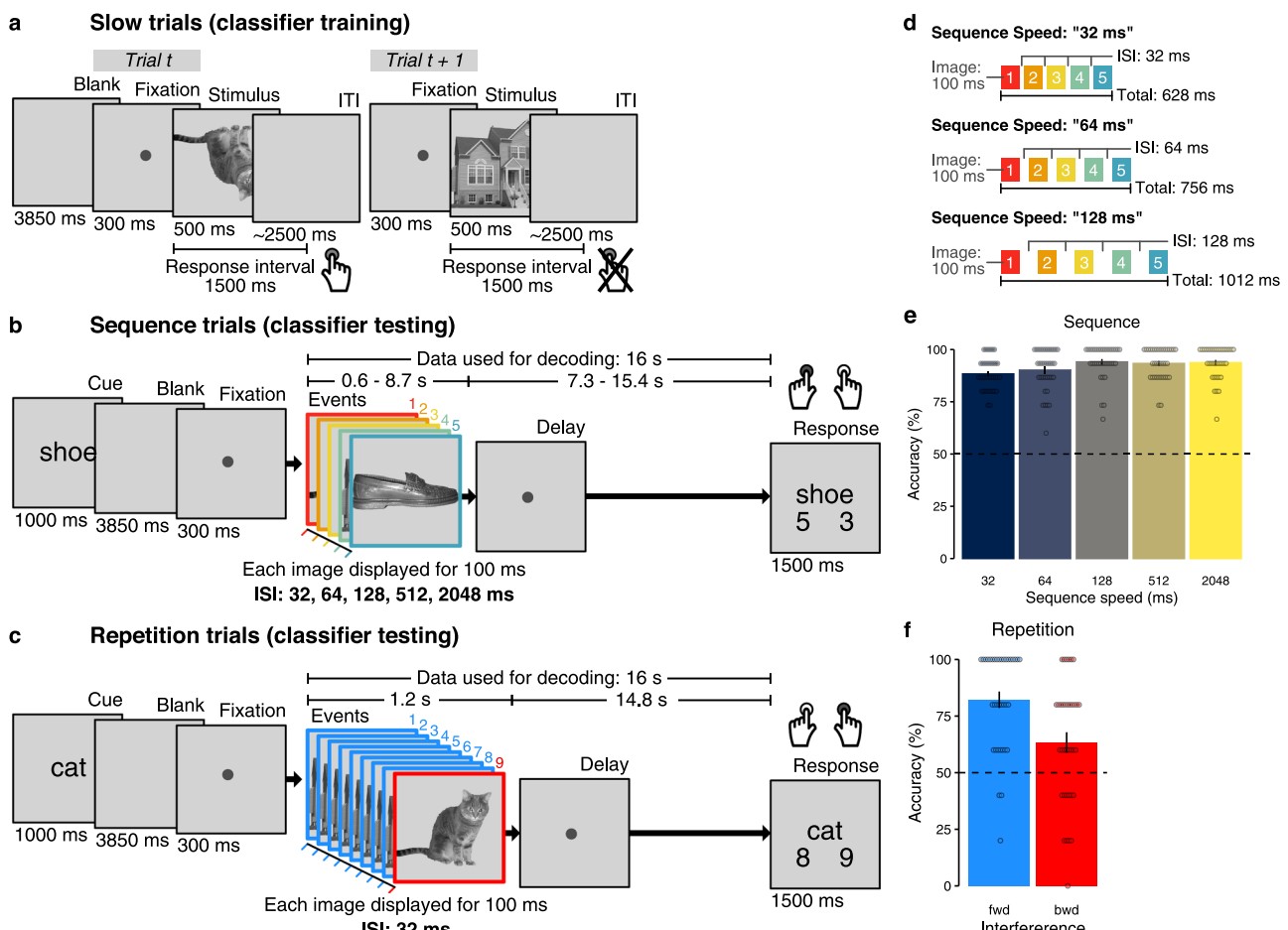

**Fig. 1 Task design and behavioral performance. a** On slow trials, individual images were presented and inter-trial intervals (ITIs) were 2.5 s on average. Participants were instructed to detect upside-down visual stimuli (20% of trials) but not respond to upright pictures. Classifier training was performed on fMRI data from correct upright trials only. **b** Sequence trials contained five unique visual images, separated by five levels of inter-stimulus intervals (ISIs) between 32 and 2048 ms. **c** Repetition trials were always fast (32 ms ISI) and contained two visual images of which either the first or the second was repeated eight times (causing backward and forward interference, respectively). In both task conditions, participants were asked to detect the serial position of a cued target stimulus in a sequence and select the correct answer after a delay period without visual input. One sequence or repetition trial came after five slow trials. fMRI analyses focused on the time from sequence onset to the end of the delay period (16 s ≈ 13 TRs, 1 TR = 1.25 s). **d** Illustration of the three fastest sequence speed conditions of 32, 64, and 128 ms ISI between images. **e** Mean behavioral accuracy in sequence trials (in %) as a function of sequence speed (ISI, in ms; $N = 36$, $ts \geq 23.78$, $ps < 0.001$, $ds \geq 3.96$, linear mixed effects (LME) model and five one-sided one-sample $t$-tests against chance (50%), false discovery rate (FDR) correction). **f** Mean behavioral accuracy in repetition trials (in %), as a function of which sequence item was repeated (fwd = forward, bwd = backward condition; $N = 36$, $ts \geq 2.94$, $ps \leq 0.003$, $ds \geq 0.49$, two one-sided one-sample $t$-tests against chance (50%) with FDR-correction). All error bars represent ±1 standard error of the mean (SEM). All statistics have been derived from data of $N = 36$ human participants who participated in one experiment. The horizontal dashed lines in (e) and (f) indicate 50% chance level. The original authors of Haxby et al.[55] hold the copyright of the stimulus material (individual images of a cat, chair, face, house, and shoe) shown in (a), (b), and (c) and made it available under the terms of the Creative Commons Attribution-Share Alike 3.0 license (see http://data.pymvpa.org/datasets/haxby2001/and http://creativecommons.org/licenses/by-sa/3.0/ for details). Source data are provided as a Source Data file.

independently, i.e., the order effect could not have been a side effect of element detection.

Participants viewed images of five different objects. During *slow trials* (Fig. 1a, 600 trials in total), individual images were shown with inter-trial intervals (ITIs) of approximately 2.5 s, as is common in fMRI decision-making experiments (cf.[44,47,52,54]). In fast trials (120 trials in total), the same images were shown as either a random sequence of all five objects (*sequence trials*, 75 trials, Fig. 1b), or two objects were repeated several times (*repetition trials*, 45 trials, Fig. 1c). Importantly, image presentation rate was greatly increased in sequence and repetition trials, with as little as 32 ms between stimuli and a presentation time of 100 ms per stimulus. Logistic regression classifiers were trained on data from slow trials and applied to sequence and repetition trials,

as well as to resting-state data. We then asked whether the order and the elements of fast sequences are detectable from fMRI signals, depending on sequence speed, number of repetitions, level of background noise, and timing uncertainty. To this end, visual stimuli in sequence and repetition trials were presented in a precisely timed and ordered manner, as detailed below. Since activation patterns were primarily visual in nature, only data from visual and ventral temporal cortex were considered. A corresponding analysis using hippocampal data did not yield comparable results, see below. The analyses included $N = 36$ human participants who underwent two fMRI sessions with four task runs each, i.e., eight runs in total. Four additional participants were excluded from analyses due to insufficient performance, see Methods and Supplementary Information (SI) (Supplementary

Fig. 1a). Sessions were separated by 9 days on average (SD = 6 days, range: 1–24 days).

**Training fMRI pattern classifiers on slow events.** In slow trials, participants repeatedly viewed the same five images individually for 500 ms (images showed a cat, chair, face, house, and shoe, taken from[55]). Temporal delays between images were set to 2.5 s on average, as typical for task-based fMRI experiments[56]. To ensure that image ordering did not yield biased classifiers through biased pattern similarities (cf.[57]), each possible order permutation of the five images was presented exactly once (120 sets of 5 images each). Participants were kept attentive by a cover task that required them to press a button whenever a picture was shown upside-down (20% of trials; mean accuracy = 99.44%; $t_{(35)}$ = 263.27, 95% CI [99.13, $+\infty$]; $p < 0.001$, compared to chance (50%); $d = 43.88$; Supplementary Fig. 1a–c). Using data from correct upright slow trials, we trained five separate multinomial logistic regression classifiers, one for each image category (one-vs.-rest; see Methods for details; cf.[55]). fMRI data were masked by a gray-matter-restricted region of interest (ROI) of occipito-temporal cortex, known to be related to visual object processing (11,162 voxels in the masks on average; cf.[55,58–60]). Spatial patterns associated with image categories indicated a mix of overlapping and non-overlapping sets of voxels, and average correlations between the mean voxel patterns were negative (see SI). We accounted for hemodynamic lag by extracting fMRI data acquired 3.75–5 s after stimulus onset (corresponding to the fourth repetition time (TR), see Methods). Cross-validated (leave-one-run-out) classification accuracy was on average 69.22% (SD = 11.18%; $t_{(35)}$ = 26.41, 95% CI [66.07, $+\infty$], $p < 0.001$, compared to chance (20%); $d = 4.40$; Fig. 2a). In order to examine the sensitivity of the classifiers to pattern activation time courses, we applied them to seven TRs following stimulus onset on each trial. This analysis confirmed delayed and distinct increases in the estimated probability of the true stimulus class given the data, peaking at the fourth TR after stimulus onset, as expected, given that the classifiers were trained on data from the fourth TR following stimulus onset (Fig. 2b). The peak in probability for the true stimulus shown on the corresponding trial was significantly higher than the mean probability of all other stimuli at that time point ($t$s $\geq 17.95$, $p$s $< 0.001$, $d$s $\geq 2.99$; Bonferroni-corrected). Decoding in an anatomical ROI of the hippocampus did not surpass the chance level (decoding accuracy: mean (M) = 20.52%, SD = 1.49%; $t_{35}$ 2.10, 95% CI [20.02, 21.03], $p = 0.05$, compared to chance (20%), $d = 0.35$; using the same decoding approach, see SI for details).

*Single event and event sequence modeling.* The data shown in Fig. 2b highlight that multivariate decoding time courses are delayed and sustained, similar to single-voxel hemodynamics. We captured these dynamics elicited by single events by fitting a sine-based response function to the time courses on slow trials (a single sine wave flattened after one cycle, with parameters for amplitude $A$, response duration $\lambda$, onset delay $d$, and baseline $b$; Fig. 2c and Supplementary Fig. 4; see Methods). Based on this fit to single events, we derived expectations for probabilistic time courses during sequential events. The sequentiality analyses reported below essentially quantify how well successive activation patterns can be differentiated from one another depending on the speed of stimulus sequences. We therefore considered two time-shifted response functions and derived the magnitude and time course of differences between them. Based on the sinusoidal nature of the response function, the time course of this difference can be approximated by a single sine wave with duration $\lambda_\delta = \lambda + \delta$, where $\delta$ is the time between events and $\lambda$ is the average fitted

single event duration, here $\lambda = 5.24$ TRs (see Eqs. (4) and (5), Methods). This average parameter was used for all further analyses (Fig. 2c, d; see Methods). In this model, the amplitude is proportional to the time shift between events (until time shifts become larger than the time-to-peak of the response function). Consequently, after an onset delay ($d = 0.56$ TRs), the difference in probability of two time-shifted events is expected to be positive for the duration of half a cycle, i.e., $0.5\lambda_\delta = 0.5(5.24 + \delta)$ TRs, and negative for the same period thereafter. Simply put, this means that the strength of overlapping activations will initially be ordered forward, in the same way as the sequence, i.e., earlier items will be activated stronger. In a later period, however, this will reverse and result in backwards ordering, i.e., earlier items will be activated less. In summary, three predictions therefore arise from this model: (1) the first event will dominate the signal in earlier TRs, and activation strengths will be proportional to the true event order during the sequential process; (2) in later TRs, the last sequence element will dominate the signal, and activation strengths will be ordered backwards; and (3) the duration and strength of these two effects will depend on the fitted response duration and the timing of the stimuli as specified above (Fig. 2e and Eqs. (1)–(5); see Methods). For sequences with more than two items (as in sequence trials, see below), $\delta$ is defined as the interval between the onsets of the first and last sequence item. To reflect the relation between the true order and the activation strength, we henceforth term the above-mentioned early and late TRs as the forward and backward periods, and consider all results below either separately for these phases, or for both relevant periods combined (calculating periods depending on the timings of image sequences and rounding TRs, see Methods).

*Detecting sequentiality in fMRI patterns following fast and slow neural event sequences.* Our first major aim was to test detection of sequential order of fast neural events with fMRI. We therefore investigated the above-mentioned sequence trials in which participants viewed a series of five unique images at different speeds (Fig. 1b). Sequence speed was manipulated by leaving either 32, 64, 128, 512, or 2048 ms between pictures, while images were always presented briefly (100 ms per image, total sequence duration 0.628–8.692 s). Note, that we refer to the inter-stimulus interval (ISI) as "sequence speed" (see Fig. 1d). Sequences always contained each image exactly once. Every participant experienced 15 randomly selected image orders that ensured that each image appeared equally often at the first and last position of the sequence (all 120 possible orders counterbalanced across participants). The task required participants to indicate the serial position of a verbally cued image 16 s after the first image was presented. This delay between visual events and response (roughly spanning 13 TRs; see x-axes in Fig. 3a, b) allowed us to measure sequence-related fMRI signals without interference from following trials, while the upcoming question did not necessitate memorization of the sequence during the delay period. Performance was high even in the fastest sequence trials (32 ms: M = 88.33%, SD = 7.70, $t_{35}$ = 29.85, 95% CI [86.16, $+\infty$], $p < 0.001$ compared to chance (50%), $d = 4.98$), and only slightly reduced compared to the slowest condition (2048 ms: M = 93.70%, SD = 7.96, $t_{35}$ = 32.95, 95% CI [91.46, $+\infty$], $p < 0.001$ compared to chance (50%), $d = 5.49$; Fig. 1e and Supplementary Fig. 1d).

We investigated whether sequence order was detectable from the relative pattern activation strength within a single measurement. Examining the time courses of probabilistic classifier evidence during sequence trials (Fig. 3a) showed that the time delay between events was indeed reflected in sustained within-TR ordering of probabilities in all speed conditions. Specifically, immediately after sequence onset, the first element (red line) had

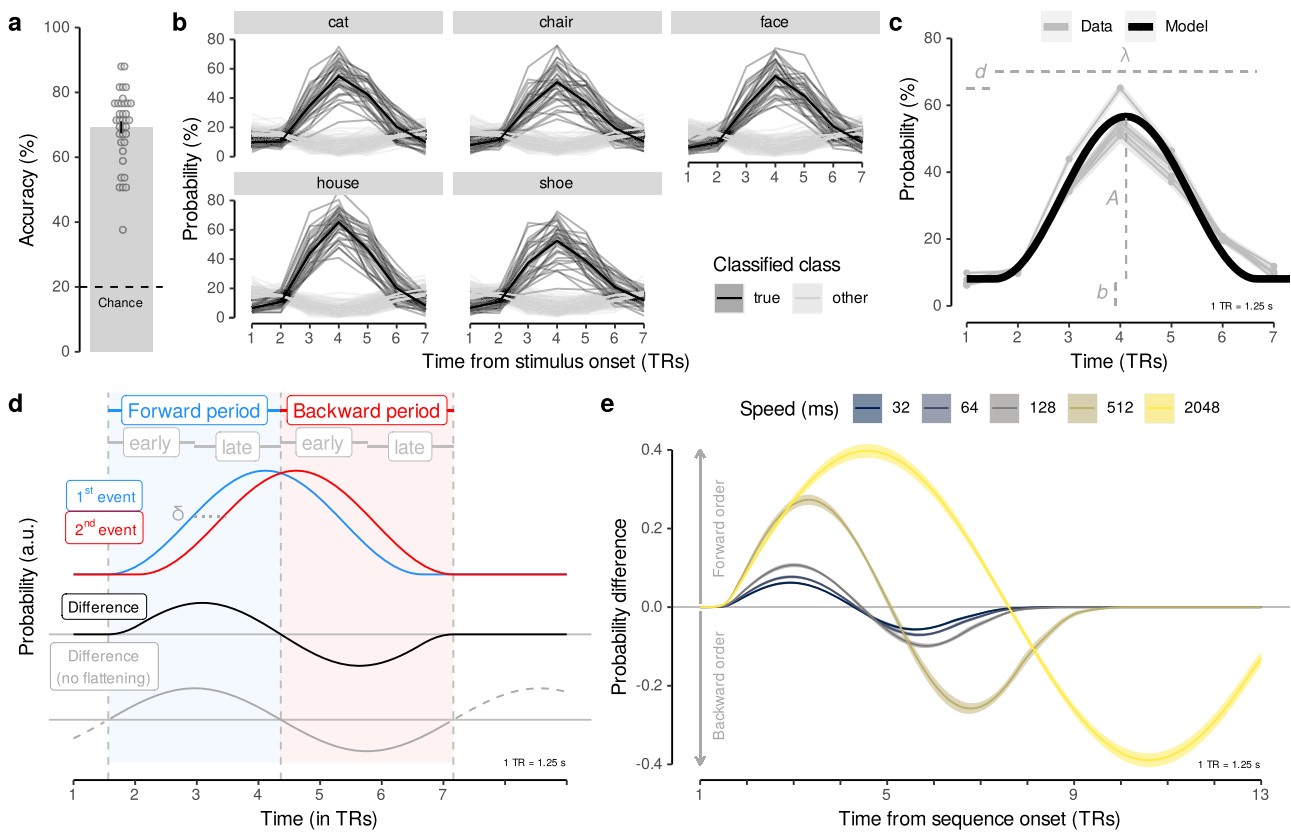

**Fig. 2 Classification accuracy and multivariate response functions. a** Cross-validated classification accuracy in decoding the five unique visual objects in occipito-temporal data during task performance (in %; $N = 36$, $t_{(35)} = 26.41$, 95% CI [66.07, $+\infty$], $p < 0.001$, $d = 4.40$, one one-sided one-sample $t$-test, no multiple comparisons). Chance level is 20% (dashed line). Each dot corresponds to averaged data from one participant. Error bar represents ±1 SEM. **b** Time courses (in TRs from stimulus onset) of probabilistic classification evidence (in %) for all five stimulus classes. Substantial delayed and extended probability increases for the stimulus presented (black lines) on a given trial (gray panels) were found. Each line represents one participant ($N = 36$, $t$s ≥ 17.95, $p$s < 0.001, $d$s ≥ 2.99, 35 two-sided two-sample $t$-tests, Bonferroni-corrected). **c** Average probabilistic classifier response for the five stimulus classes (gray lines) and fitted sine-wave response model using averaged parameters (black line). **d** Illustration of sinusoidal response functions following two neural events (blue and red lines) time-shifted by delta seconds (dashed horizontal line). The resulting difference between event probabilities (black line) establishes a forward (blue area) and backward (red area) time period, split into early and late phases. The sine-wave approximation without flattened tails is shown in gray. **e** Probability differences between two time-shifted events predicted by the sinusoidal response functions depending on the event delays (delta) as they occurred in the five different sequence speed conditions (colors), based on Eq. (6). All statistics have been derived from data of $N = 36$ human participants who participated in one experiment. Source data are provided as a Source Data file.

the highest probability and the last element (blue line) had the lowest probability. This pattern reversed afterwards, following the forward and backward dynamics that were predicted by the time-shifted response functions (Fig. 2d; forward and backward periods adjusted to sequence speed, see above and Methods). A TR-wise linear regression between the serial positions of the images and their probabilities confirmed this impression. In all speed conditions, the mean slope coefficients initially increased above zero (reflecting higher probabilities of earlier compared to later items) and decreased below zero afterwards (Fig. 3b and Supplementary Fig. 6a). Considering mean regression coefficients during the predicted forward and backward periods, we found significant forward ordering in the forward period at ISIs of 128, 512, and 2048 ms ($t$s ≥ 2.85, $p$s ≤ 0.009, $d$s ≥ 0.47) and significant backward ordering in the backward period in all speed conditions ($t$s ≥ 3.89, $p$s < 0.001, $d$s ≥ 0.65, FDR-corrected; Fig. 3c). Notably, the observed time course of regression slopes on sequence trials (Fig. 3b) closely matched the time course predicted by our modeling approach (Fig. 2d), as indicated by strong correlations for all speed conditions between model predictions and the averaged time courses (Fig. 3d; Pearson's $r$s ≥ 0.81, $p$s < 0.001) as well as significant within-participant correlations (Fig. 3e; mean

Pearson's $r$s ≥ 0.23, $t$s ≥ 3.76, $p$s < 0.001, compared to zero, $d$s ≥ 0.63, FDR-corrected).

Choosing a different index of association like rank correlation coefficients (Supplementary Figs. 5a, b and 6c) or the mean step size between probability-ordered events within TRs (Supplementary Figs. 5c, d and 6d) produced qualitatively similar results (for details, see SI). Removing the sequence item with the highest probability at every TR also resulted in similar effects, with backward sequentiality remaining significant at all speeds ($p$ ≤ 0.002) except the 32 and 128 ms conditions ($p$ ≥ 0.20), and forward sequentiality still being evident at speeds of 512 and 2048 ms ($p$ ≤ 0.004; Supplementary Fig. 7a, b). To identify the drivers of the apparent asymmetry in detecting forward and backward sequentiality, we ran two additional control analyses and either removed the probability of the first or the last sequence item (forward and backward periods adjusted accordingly). Removal of the first sequence item had little impact on sequentiality detection (Supplementary Fig. 7c, d and SI), but removing the last sequence item markedly affected the results such that significant forward and backward sequentiality was only evident at speeds of 512 and 2048 ms (Supplementary Fig. 7e, f and SI).

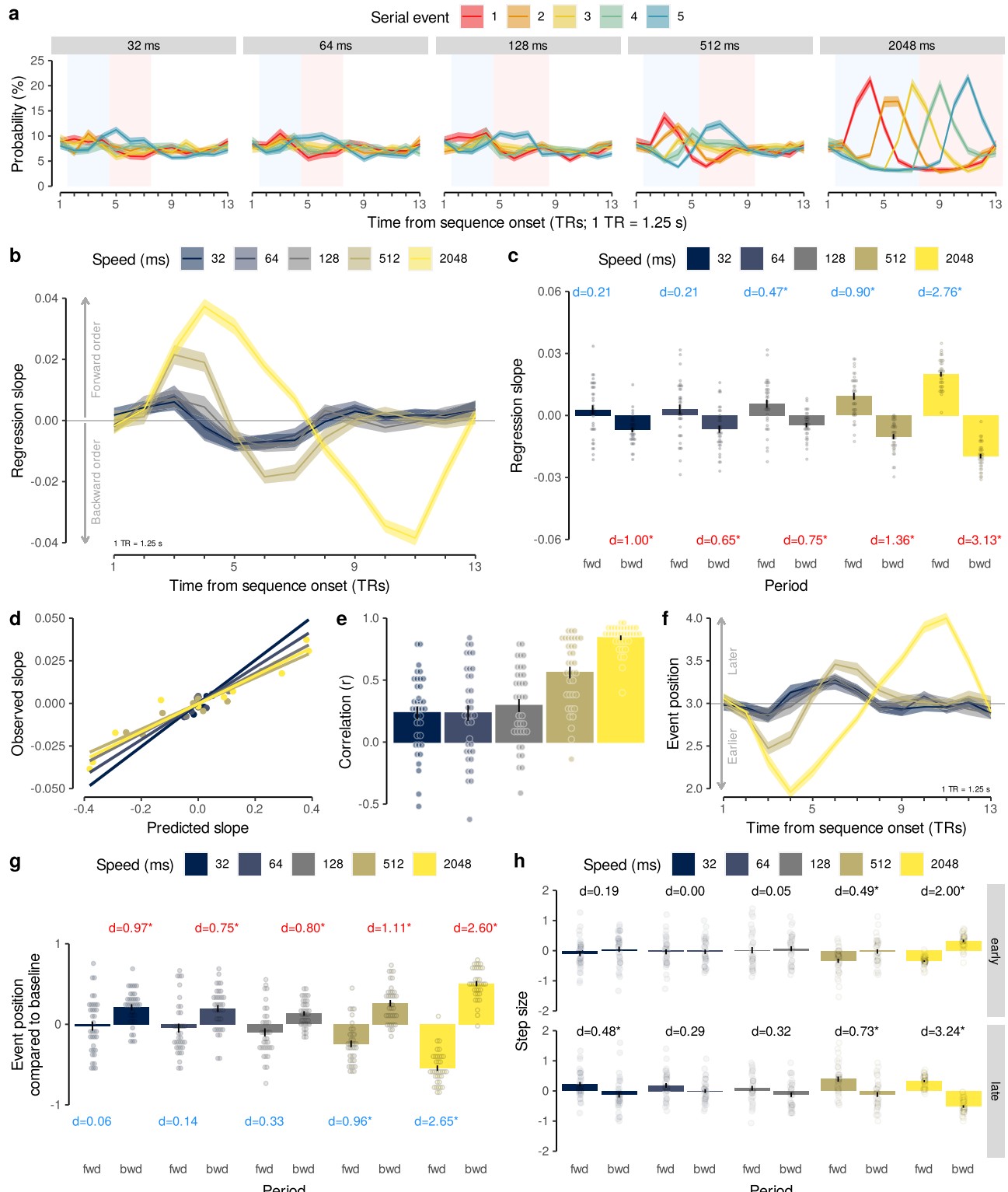

Next, we investigated evidence of pattern sequentiality across successive measurements, similar to Schuck and Niv[54]. Specifically, for each TR we only considered the decoded image with the highest probability and asked whether earlier images were decoded primarily in earlier TRs, and whether later images were primarily decoded in later TRs. In line with this prediction, the average serial position fluctuated in a similar manner as the regression coefficients, with a tendency of early positions to be decoded in early TRs, and later positions in later TRs (Fig. 3f).

The average serial position of the decoded images was therefore significantly different between the predicted forward and backward period at all sequence speeds (all $ps < 0.001$, Fig. 3g, Supplementary Fig. 6d). Compared to baseline (mean serial position of 3), the average serial position during the forward period was significantly lower for speeds of 512 and 2048 ms (all $ps < 0.001$). The average decoded serial position at later time points was significantly higher compared to baseline in all speed conditions, including the 32 ms condition (all $ps < 0.001$). Thus,

**Fig. 3 Sequence order is reflected in probability time courses. a** Time courses (TRs from sequence onset) of classifier probabilities (%) per event (colors) and sequence speed (panels). Forward (blue) and backward (red) periods shaded as in Fig. 2d. **b** Time courses of mean regression slopes between event position and probability for each speed (colors). Positive/negative values indicate forward/backward sequentiality, respectively. **c** Mean slope coefficients for each speed (colors) and period (forward vs. backward; $N = 36$, $ts \geq 2.85$, $ps \leq 0.009$, $ds \geq 0.47$ (significant tests only), ten two-sided one-sample $t$-tests against zero, FDR-corrected). Asterisks indicate significant differences from baseline. **d** Between-participant correlation between predicted (Fig. 2e, Eq. (6)) and observed (b) time courses of mean regression slopes (13 TRs per correlation, Pearson's $rs \geq 0.81$, $ps < 0.001$). Each dot represents one TR. **e** Mean within-participant correlations between predicted and observed slopes as in (d) ($N = 36$, mean Pearson's $rs \geq 0.23$, $ts \geq 3.76$, $ps \leq 0.001$, compared to zero, $ds \geq 0.63$, FDR-corrected). **f** Time courses of mean event position for each speed, as in (b). **g** Mean event position for each period and speed, as in (c) ($N = 36$, $ts \geq 4.78$, $ps < 0.001$, $ds \geq 0.75$ (significant tests only), ten two-sided one-sample $t$-tests against baseline, FDR-corrected). **h** Mean step sizes of early and late transitions for each period and speed ($N = 36$, $ts \geq 2.88$, $ps \leq 0.006$, $ds \geq 0.48$ (significant tests only), ten two-sided one-sample $t$-tests against zero, FDR-corrected). Asterisks indicate differences between periods, otherwise as in (c). Each dot represents data of one participant. Error bars/shaded areas represent ±1 SEM. All statistics have been derived from data of $N = 36$ human participants who participated in one experiment. Effect sizes indicated by Cohen's $d$. Asterisks indicate $p < 0.05$, FDR-corrected. 1 TR = 1.25 s. Source data are provided as a Source Data file.

earlier images were decoded earlier after sequence onset and later images later, as expected.

This sequential progression through the involved sequence elements had implications for transitions between consecutively decoded events. The transitions will be a direct function of the slope of the average decoded position shown in Fig. 3f. When the slope is negative, the steps between successive sequence items are backward and reflect the transition from a later position to an earlier position. When the slope is positive, the steps are forward, reflecting a progression from an earlier event position to a later event position. This can be verified by computing the step sizes between consecutively decoded serial events as in Schuck and Niv[54]. For example, observing a $2 \rightarrow 4$ transition of decoded events in consecutive TRs would correspond to a forward step of size $+2$, while a $3 \rightarrow 2$ transition would reflect a backward step of size $-1$. As can be seen from Fig. 3f, both the early and late phase of the response (see phases in Fig. 2d) included periods with a negative and a positive slope, in line with our predictions (formally, the prediction can be obtained by taking the derivative with respect to time of Eq. (6), see Methods, i.e., the function shown in Fig. 2e). We therefore considered the periods with a positive and negative position slope separately for the early and late phase. As expected, the early transitions were mainly forward during the period of a positive slope as compared to the negative slope periods for speed conditions of 512 and 2048 ms ($ps \leq 0.01$, Fig. 3h). Similarly, the late transitions were also forward and backward during the positive and negative slope periods, respectively, and differed in all speed conditions ($ps \leq 0.01$, Fig. 3h), except the 64 and 128 ms conditions ($p = 0.12$ and $p = 0.10$; FDR-corrected). This analysis suggests that transitions between decoded items reflect the ordered progression from early to late and then from late to early sequence events, even when events were separated only by tens of milliseconds.

*Detecting sequence elements: asymmetries and interference effects.* We next turned to our second main question, asking whether we can detect which patterns were part of a fast sequence and which were not. One important reason why detecting which patterns were activated during sequence events might be more difficult than in a standard setting is that co-activation of multiple patterns close in time could lead to interference. We therefore investigate such interference in detail below.

We analyzed classification time courses in repetition trials, in which only two out of the five possible images were shown. One of the two images was repeated, while the other one was shown only once. This setup allowed us to study to what extent another activation (the repeated image) can interfere with the detection of a brief activation pattern of interest (the image shown only once). The repeating image was shown eight times, which created maximally adverse effects for the detection of the single image. To

ask if detection of brief activations is differently affected by events occurring before versus after the single event, we varied whether the single item was preceded or followed by the repeated item. We pose this question because the backward effects were consistently larger than forward effects in our sequentiality analyses reported above (Fig. 3c), suggesting asymmetric detection sensitivity. This implies that one briefly presented item at the end of a sequence will be easier to detect than a briefly presented item at the beginning of a sequence, even though both were equally close in time to another strong activation signal. To test this idea, we considered the two order conditions described above. We will term the case in which the first image was shown briefly once and followed immediately by eight repetitions of a second image the *forward interference* condition, because the forward phase of the sequential responses suffers from interference. Correspondingly, trials in which the first image was repeated eight times and the second image was shown once will be termed the *backward interference* condition. In all cases, images were separated by only 32 ms. Participants were kept attentive by the same cover task used in sequence trials (Fig. 1c). Average behavioral accuracy was high on repetition trials (M = 73.46%, SD = 9.71%; Fig. 1f and Supplementary Fig. 1a) and clearly differed from a 50% chance level ($t_{(35)} = 14.50$, 95% CI [70.72, $+\infty$], $p < 0.001$, $d = 2.42$). Splitting up performance into forward and backward interference trials showed performance above chance level in both conditions (M = 82.22% and M = 63.33%, respectively, $ts \geq 2.94$, $ps \leq 0.003$, $ds \geq 0.49$, Fig. 1f).

As before, we applied the classifiers trained on slow trials to the data acquired in repetition trials and obtained the estimated probability of every class given the data for each TR (Fig. 4a and Supplementary Fig. 9). The expected relevant time period was determined to be from TRs 2 to 7 and used in all analyses (see rectangular areas in Fig. 4a).

We first asked whether our classifiers indicated that the two events that were part of the sequence were more likely decoded than items that were not part of the sequence. Indeed, the event types (first, second, non-sequence) had significantly different mean decoding probabilities, with sequence items having a higher probability (first: M = 20.19%; second: M = 24.78%) compared to non-sequence items (M = 7.72%; both $ps < 0.001$, corrected; main effect: $F_{2,57.78} = 110.13$, $p < 0.001$; Fig. 4b). Moreover, the probability of decoding within-sequence items depended on the condition and whether the item was repeated or not. Considering both interference conditions (forward/backward) in the same analysis revealed a main effect of condition, $F_{2,41.64} = 146.15$, $p < 0.001$, as well as an interaction between condition and whether the item was repeated, $F_{2,140.00} = 122.59$, $p < 0.001$. This indicated that the forward phase suffered from much stronger interference than the backward phase. In the *forward interference* condition, the repeated second event had an approximately 18% higher

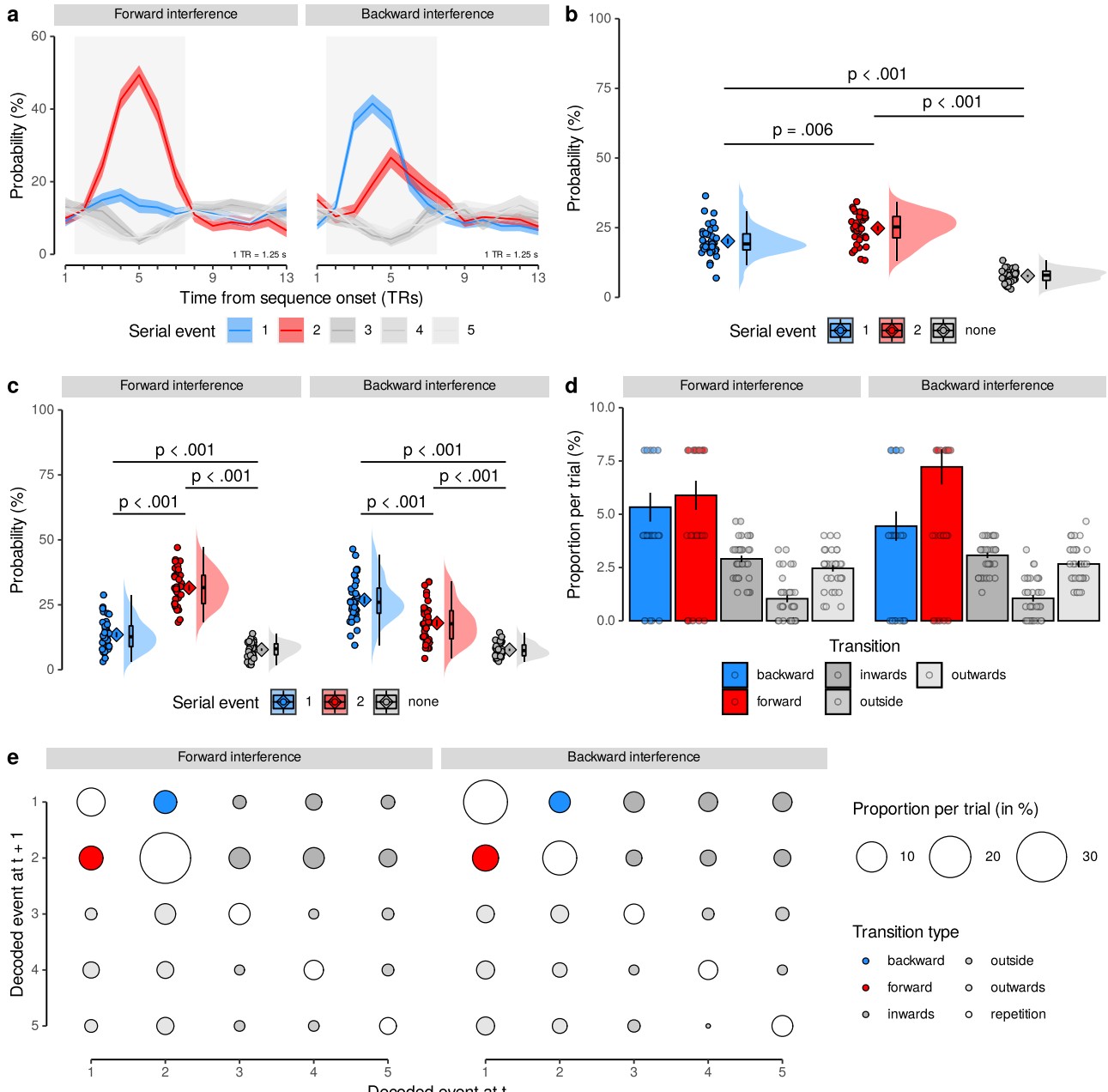

**Fig. 4 Ordering of two-item sequences on repetition trials. a** Time courses (in TRs from sequence onset) of probabilistic classifier evidence (in %) in repetition trials, color-coded by event type (first, second and the three remaining non-sequence items, see legend). Data shown separately for forward (left) and backward (right) interference conditions. Gray background indicates relevant time period independently inferred from response functions (Fig. 2d). Shaded areas represent ±1 SEM. 1 TR = 1.25 s. **b** Mean probability of event types averaged across all TRs in the relevant time period, as in (a). Each dot represents one participant, the probability density of the data is shown as `rain cloud plots` (cf.[141]). Boxplots indicate the median and interquartile range (IQR, i.e., distance between the first and third quartiles). The lower and upper hinges correspond to the first and third quartiles (the 25th and 75th percentiles). The upper whisker extends from the hinge to the largest value no further than 1.5* IQR from the hinge. The lower whisker extends from the hinge to the smallest value at most 1.5* IQR of the hinge. The diamond shapes show the sample mean and error bars indicate ±1 SEM (N = 36, $ts \geq 3.31$, $ps \leq 0.006$, LME model with post hoc Tukey's honest significant difference (HSD) tests). **c** Average probability of event types, separately for forward/backward conditions as in (a), plots as in (b) (N = 36, $ts \geq 4.14$, $ps < 0.001$, LME model with post hoc Tukey's HSD tests). **d** Mean trial-wise proportion of each transition type, separately for forward/backward conditions, as in (a) (N = 36, $ts \geq 4.64$, $ps < 0.001$, four two-sided paired t-tests, Bonferroni-corrected). **e** Transition matrix of decoded images indicating mean proportions per trial, separately for forward/backward conditions, as in (a). Transition types highlighted in colors (see legend). All statistics have been derived from data of N = 36 human participants who participated in one experiment. Source data are provided as a Source Data file.

probability than the single first event (31.55% vs. 13.50%, $p < 0.001$). In the *backward interference* condition, the repeated first event had only 9% higher probability than the single second event (26.87% vs. 18.00%, $p < 0.001$, corrected). This means that the

item shown only once was easier to detect when it followed a sustained activation of a different pattern, compared to when it preceded an interfering activation (Fig. 4c). We found no main effect of repetition, $p = 0.91$ (Fig. 4c).

Importantly, however, both sequence elements still differed from non-sequence items even under conditions of interference (forward: 7.75% and backward: 7.69%, respectively, all $ps < 0.001$, corrected), indicating that sequence element detection remains possible under such circumstances. Using data from all TRs revealed qualitatively similar significant effects ($p \leq 0.04$ for all but one test after correction, see SI). Repeating all analyses using proportions of decoded classes (the class with the maximum probability was considered decoded at every TR), or considering all repetition trial conditions, also revealed qualitatively similar results. Thus, brief events can be detected despite significant interference.

We next asked which implications these findings have for the observed pattern transitions (cf.[54]). To this end, we analyzed the trial-wise proportions of transitions between consecutively decoded events, and asked whether forward transitions between sequence items were more likely than transitions between a sequence and a non-sequence item (*outward transitions*) or between two non-sequence items (*outside transitions*; for details, see Methods). This analysis revealed that forward transitions (5.89%) were more frequent than both outward transitions (2.46%), and outside transitions (1.04%, both $ps < 0.001$, $ts \geq 4.64$, Bonferroni-corrected; Fig. 4d) in the forward interference condition. The same was true in the backward interference condition (forward transitions: 7.22%; outward transitions: 2.67%; outside transitions: 1.06%, all $ps < 0.001$, $ts \geq 5.14$). The full transition matrix is shown in Fig. 4e. Repetitions of the first or second item are shown on the upper two diagonal elements (with all consecutive repetitions of items labeled *repetition* in Fig. 4e), and were not considered in this analysis.

Together, the results from repetition trials indicated that: (1) within-sequence items could be clearly detected despite inter-ference from other sequence items; (2) event detection was asymmetric, such that items occurring at the end of sequences can be detected more easily than those occurring at the beginning; and (3) the detection of sequence items made it possible to observe within-sequence transitions between decoded items.

Note that our analyses focused on the two extreme cases of repetition trials with one versus eight repetitions of the first or second item while the experiment also included repetition trials with intermediate levels of repetitions (see SI). Specifically, other repetition trials included cases in which the second item began to appear at each possible position from 2 to 9. The other repetition trials could therefore include, for instance, three repetitions of the first and six repetitions of the second image, or four repetitions of the first and five repetitions of the second item, etc. The results reported in the SI indicate that effects in these trials show smooth transition between the extremes shown in the main manuscript.

*Detecting sparse sequence events with lower signal-to-noise ratio (SNR).* The results above indicate that detection of fast sequences is possible if they are under experimental control. In most applications of our method, however, this will not be the case. When detecting replay, for instance, sequential events will occur spontaneously during a period of noise. We therefore next assessed the usefulness of our method under such circumstances.

We first characterized the behavior of sequence detection metrics during periods of noise. To this end, we applied the logistic regression classifiers to fMRI data acquired from the same participants ($n = 32$ out of 36) during a 5-min (233 TRs) resting period before any task exposure. Classifier probabilities during rest fluctuated wildly, often with a single category having a high probability, while all other categories had probabilities close to zero. During fast sequence periods, in contrast, the near-simultaneous activation of stimulus-driven activity led to reduced

probabilities, such that category probabilities tended to be closer together and less extreme. In consequence, the average standard deviation of the probabilities per TR during rest and slow (2048 ms) sequence periods was higher (M = 0.23 and M = 0.22, respectively) compared to the average standard deviation in the fast sequence condition (32 ms; M = 0.20; $ts \geq 4.17$; $ps < 0.001$; $ds \geq 0.74$; Fig. 5a).

As before, we fitted regression coefficients through the classifier probabilities of the rest data and, for comparison, concatenated data from the 32 and 2048 ms sequence trials (Fig. 5b, c). As predicted by our modeling approach (Fig. 2e), and shown in the previous section (Fig. 3b), the time courses of regression coefficients in the sequence conditions were characterized by rhythmic fluctuations whose frequency and amplitude differed between speed conditions (Fig. 5c). To quantify the magnitude of this effect, we calculated frequency spectra of the time courses of the regression coefficients in rest and concatenated sequence data (Fig. 5d; using the Lomb-Scargle method, e.g.[61] to account for potential artifacts due to data concatenation, see Methods). This analysis revealed that frequency spectra of the sequence data differed from rest frequency spectra in a manner that depended on the speed condition (Fig. 5d, e). As foreshadowed by our model, power differences appeared most pronounced in the predicted frequency ranges (Fig. 5e; $ps \leq 0.002$; see Eq. (5) and Methods). Specifically, when the 32 ms condition was considered, the analyses revealed an increased power around 0.17 Hz, which corresponds to the frequency predicted to occur by our model. Data from the 2048 ms condition, in contrast, exhibited an increased power around 0.07 Hz, as predicted.

Finally, we asked whether these differences would persist if (a) only few sequence events occurred during a 5-min rest period, while (b) their onset was unknown, and (c) their SNR was lower. To this end, we synthetically generated data containing a variable number of sequence events that were inserted at random times into the resting-state data acquired before any task exposure. Specifically, we inserted between 1 and 6 sequence events into the rest period by blending rest data with TRs recorded in fast (32 ms) or slow (2048 ms) sequence trials (12 TRs per trial, random selection of sequence trials and insertion of time points, without replacement). To account for possible SNR reductions, the inserted probability time courses were multiplied by a factor $\kappa$ of $\frac{4}{5}$, $\frac{1}{2}$, $\frac{1}{4}$, $\frac{1}{8}$, or 0 and added to the probability time courses of the inversely scaled ($1-\kappa$) resting-state data. Effectively, this led to a step-wise reduction of the inserted sequence signal from 80% to 0%, relative to the SNR obtained in the experimental conditions reported above. Thus, here we use the term SNR to describe the relative mixing proportion of (a) data from the task, which contain sequential signal, with (b) data from the pre-task resting-state session, which contain only noise. Note that this is different from the common definition of SNR in univariate fMRI as the ratio of average signal to standard deviation over time.

As expected, differences in the above-mentioned standard deviation of the probability gradually increased with both the SNR level and the number of inserted sequence events when either fast or slow sequences were inserted (Fig. 5f). In our case, this led significant differences to emerge with one insert and an SNR reduced to 12.5% in both the fast and slow conditions (Fig. 5g; comparing against zero, the expectation of no difference with a conventional false-positive rate $\alpha$ of 5%; all $ps$ FDR-corrected).

Importantly, the presence of sequence events was also reflected in the frequency spectrum of the regression coefficients. Inserting fast event sequences into rest led to power increases in the frequency range indicative of 32 ms events (~0.17 Hz, Fig. 5h, i, left panel), in line with our findings above. This effect again got

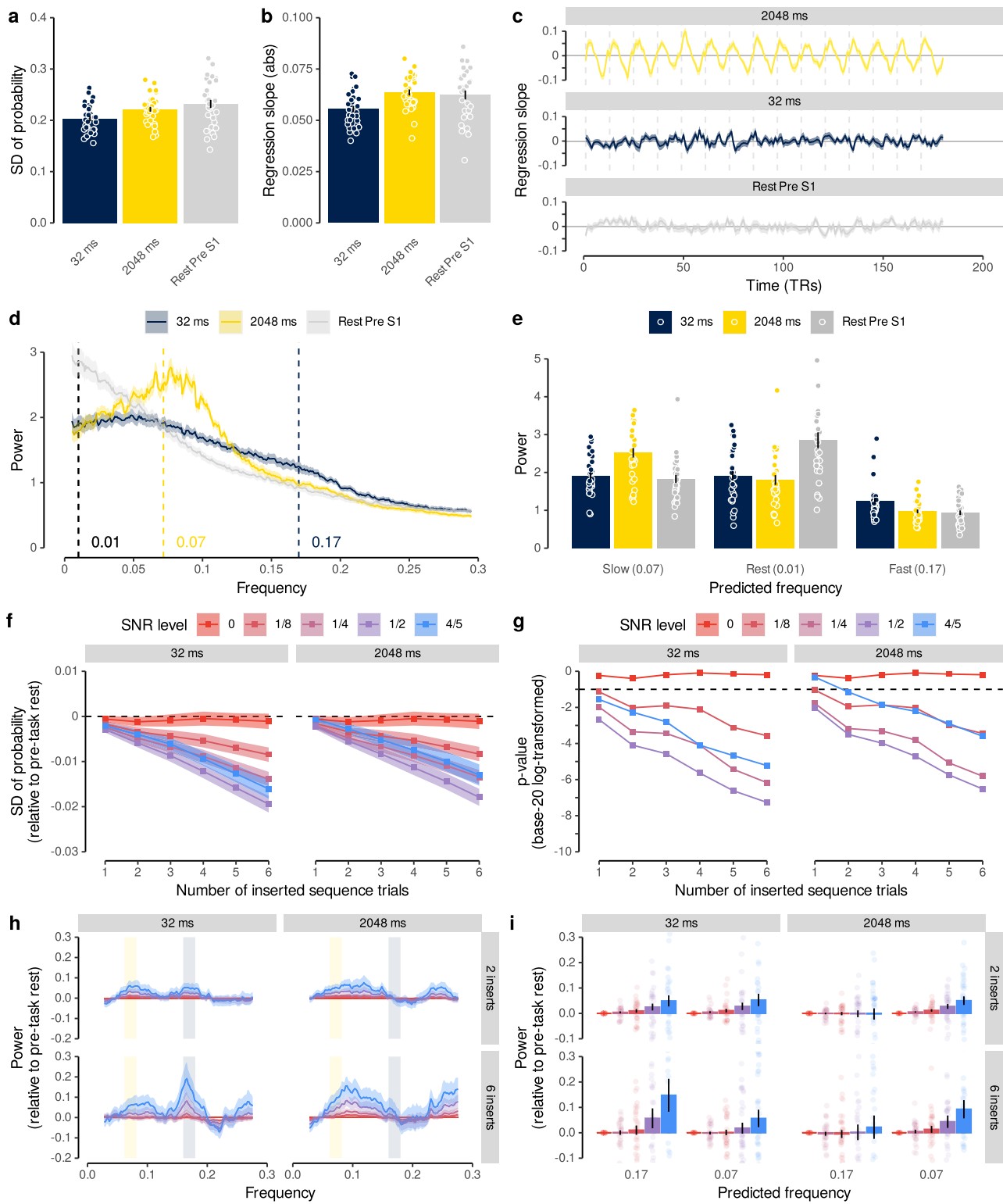

stronger with higher SNR levels and more sequence events. Inserting slow (2048 ms) sequence events into the rest period showed a markedly different frequency spectrum, with an increase around the frequency predicted for this speed (~0.07 Hz, Fig 5h, i, right panel). Comparing the power around the predicted frequency (±0.01 Hz) of both speed conditions indicated significant increases in power compared to sequence-free rest when six sequence events were inserted and the SNR was reduced to 80% ($ts \geq 2.28$, $ps \leq 0.03$, $ds \geq 0.40$). Hence, the

presence of spontaneously occurring sub-second sequences during rest can be detected in the frequency spectrum of our sequentiality measure, and distinguished from slower second-scale sequences that might reflect conscious thinking.

*Detecting fast reactivations in post-task resting-state data.* Finally, we asked whether our task elicited spontaneous replay of image sequences in object-selective brain areas during rest after the task. Based on the above findings, we reasoned that potentially

**Fig. 5 Detecting sparse sequence events with lower SNR. a** Mean standard deviation of classifier probabilities in rest and sequence data ($n = 32$, $ts \geq 4.17$, $ps < 0.001$, $ds \geq 0.74$, two two-sided paired $t$-tests comparing rest and 2048 ms conditions against 32 ms condition, FDR-corrected). **b** Mean absolute regression slopes, as in (a) ($n = 32$, $ts \geq 4.64$, $ps < 0.001$, $ds \geq 0.82$, two two-sided paired $t$-tests comparing rest and 2048 ms conditions against 32 ms condition, FDR-corrected). **c** Time courses of the regression slopes (signed values, not magnitudes) in rest and sequence data. Vertical lines indicate trial boundaries. **d** Normalized frequency spectra of regression slopes in rest and sequence data. Annotations indicate predicted frequencies based on Eq. (5). **e** Mean power of predicted frequencies in rest and sequence data, as in (a). Each dot represents data from one participant ($n = 32$, $ts \geq 3.10$, $ps \leq 0.002$, two-sided paired $t$-tests, FDR-corrected). **f** Mean standard deviation of rest data including a varying number of SNR-adjusted sequence events (fast or slow). Dashed line indicates indifference from sequence-free rest ($n = 32$, $ts \geq 2.22$, $ps \leq 0.04$, 30 two-sided one-sample $t$-tests against chance, FDR-corrected). **g** Base-20 log-transformed $p$ values of $t$-tests comparing the standard deviation of probabilities in (f) with sequence-free rest. Dashed line indicates $p = 0.05$ ($N = 32$, $ts \geq 2.22$, $ps \leq 0.04$, 30 two-sided one-sample $t$-tests against chance, FDR-corrected). **h** Frequency spectra of regression slopes in SNR-adjusted sequence-containing rest relative to sequence-free rest. Rectangles indicate predicted frequencies, as in (d). **i** Mean relative power of predicted frequencies in SNR-adjusted sequence-containing rest ($n = 32$, $ts \geq 2.28$, $ps \leq 0.03$, two-sided $t$-tests against baseline, FDR-corrected). Shaded areas/error bars represent ±1 SEM. All statistics have been derived from data of $n = 32$ human participants who participated in one experiment. 1 TR = 1.25 s. Source data are provided as a Source Data file.

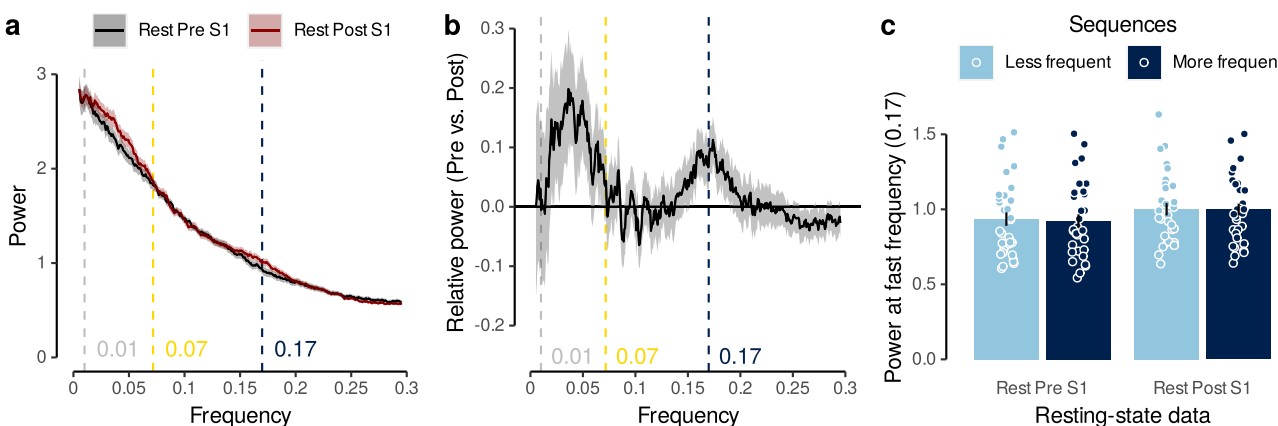

**Fig. 6 Detecting fast task-related reactivations in post-task resting-state data. a** Normalized frequency spectra of regression slopes in pre- and post-task resting-state data. Annotations indicate predicted frequencies based on Eq. (5). Shaded areas represent ±1 SEM. **b** Relative power (difference between pre- and post-task rest) of normalized frequency spectra shown in (a) ($n = 32$, $F_{1,94.99} = 6.17$, $p = 0.02$, LME model comparing pre- vs. post-task resting-state data at 0.17 Hz). **c** Mean power at predicted fast frequency (0.17 Hz) in pre- and post-task resting-state data for less and more frequent stimulus sequences ($n = 32$, $ts \geq 4.17$, $ps < 0.001$, $ds \geq 0.74$, two two-sided paired $t$-tests comparing rest and 2048 ms conditions against 32 ms condition, FDR-corrected). Each dot corresponds to averaged data from one participant. Error bars represent ±1 SEM. All statistics have been derived from data of $n = 32$ human participants who participated in one experiment. Source data are provided as a Source Data file.

reactivated sequences should become apparent in a frequency spectrum analysis. We therefore applied this analysis to resting-state data recorded after participants performed the task. Crucially, because the true sequence of potential replay events was not known, we repeated the analyses for all possible image orders, averaged the resulting frequency spectra, and compared the results to the same analysis performed on the pre-task rest session (see Methods). As shown in Fig. 6, the frequency spectrum analyses revealed a significant increase specifically in the power spectrum of the high frequency range (Fig. 6a, $F_{1,94.99} = 6.17$, $p = 0.02$ when testing pre- versus post-task data at the predicted frequency of 0.17 Hz, as before). Directly comparing pre- versus post-task rest revealed a large power difference at 0.17 Hz, indicative of replayed sequence speeds of 32 ms, as in our fastest sequence speed condition (Fig. 6b). In addition, we found a second peak at around 0.04 Hz, indicating long activations of individual items of several seconds. Thus, post-task rest seemed to be characterized by fast sequential reactivations as well as longer constant activations. We next asked whether specific sequences that had been experienced slightly more often by participants were more likely to be reactivated than less frequent sequences. During slow trials, all participants experienced all 120 possible sequential combinations of images. But in addition, each participant experienced only a subset of 15 image orders during the sequence trials. Hence, image orders experienced in sequence

trials were slightly more frequent and we asked if they were reactivated more strongly during the post-task resting-state session. This was not the case. A power increase in the fast frequency range when comparing pre- to post-task rest was found for both sets of sequences, i.e., the 15 image orders that occurred in sequence and slow trials, and the 105 that occurred only in slow trials (Fig. 6c, $ps \geq 0.13$). In summary, applying the frequency spectrum analyses to post-task resting-state therefore suggests that (1) task stimuli are reactivated during post-task rest, and (2) this reactivation happens fast, but (3) appears unspecific and not directly related to the sequences presented more frequently to participants during the task.

## Discussion
Here, we demonstrated that BOLD fMRI in combination with multivariate probabilistic decoding can be used to detect sub-second sequences of closely timed neural events non-invasively in humans. We combined probabilistic multivariate pattern analysis with time course modeling and investigated human brain activity recorded following the presentation of sequences of visual objects at varying speeds, as well as activity during rest. In the fastest case a sequence of five images was displayed within 628 ms (32 ms between pictures). Stimulus sequences were not masked. Even when using a TR of 1.25 s, achievable with conventional multi-

band (MB) echo-planar imaging (EPI), the image order could be detected from activity patterns in visual and ventral temporal cortex. Detection of briefly presented sequence items was also possible when their activation was affected by interfering signals from a preceding or subsequent sequence item and could be differentiated from images that were not part of the sequence. Our results withstood several robustness tests, and also indicated that detection is biased to most strongly reflect the last event of a sequence. Analyses of augmented resting data, in which neural event sequences occurred rarely, at unknown times, and with reduced signal strength, showed that our method could detect sub-second sequences even under such adverse conditions. Moreover, we showed that frequency spectrum analyses can be used to distinguish sub-second from supra-second sequences under such circumstances. Our approach therefore promises to expand the scope of BOLD fMRI to fast, sequential neural representations by extending multivariate decoding approaches into the temporal domain, in line with our previous findings[54].

Importantly, we applied this method not only to experimentally controlled data, but also used it to ask whether task experience might elicit spontaneous replay of sequential stimuli in post-task resting-state data, as suggested by previous studies, for reviews, see, e.g.[33,35,62,63]. Indeed, our results indicate that such reactivations occur during post-task rest and can be detected using the proposed analysis. Our analyses suggest that the reactivated sequences were fast and occurred at replay-like speeds, similar to the fastest sequence trials used in our task (32 ms between activations). Evidence for fast sequential replay was accompanied by a relative increase in power in the slower frequency range (peaking at 0.04 Hz). This could reflect an increase of slower long-lasting activations, possibly reflecting conscious thinking about the task. This supports our conclusion that the frequency spectrum of the sequentiality metric is a useful approach to detect fast replay and to distinguish it from slow activations. Our analysis did not find any evidence that only those sequences were replayed that were more frequent than others or that were presented at a fast speed during the task. Rather, our results suggest that replay seemed to equally involve all stimulus orders. However, it is important to note that our task was not optimized to elicit replay of particular sequences at all. In fact, the more frequent sequences were arranged such that the same stimuli appeared equally often at the first and last position, which makes it difficult to distinguish them from other sequences.

Of note, replay during post-task rest reflected cortical reactivations in occipito-temporal brain regions. Given that we were not able to decode on-task stimulus representations in the hippocampus, it remains unclear if reactivations occurred independently from (task-related) involvement of the hippocampus or if we were simply not able to detect concurrent reactivation in the hippocampus. This possibility of hippocampus-independent cortical reactivations raises important questions regarding the functional significance of such events. One potential reason why we found no hippocampus involvement could be that the oddball detection paradigm used for slow trials to train the classifiers involved no mnemonic task component, and therefore was not suitable to activate the hippocampus. Our previous work[54] has already demonstrated the success of our methods in hippocampal data. Taken together, our results indicate that our method allows the uncovering of fast task-related reactivations during rest and highlight the importance of task design for detecting replay in humans using fMRI.

This contrasts with previous fMRI studies in humans (for reviews see, e.g.[24,64]) that measured non-sequential reactivation as increased similarity of multi-voxel patterns during experience and extended post-encoding rest compared to pre-encoding baseline[49–51,53,65–69] or functional connectivity of hippocampal,

cortical, and dopaminergic brain structures that support post-encoding systems-level memory consolidation[66–68,70–72]. In the current study we open the path toward a better understanding of the speed and sequential nature of the observed phenomena.

The fastest sequences studied in our experiments lasted 628 ms and were therefore longer than the average hippocampal replay event of about 300 ms, e.g.[17]. Yet, several factors support the idea that our method is still relevant for the study of replay. First, previous studies have shown that a significant proportion of replay events indeed lasts much longer than 300 ms. Davidson et al.[7] report sequence lengths of up to 1000 ms and the data by Kaefer et al.[17] indicate that about 20% of events in the hippocampus are longer than 500 ms. In addition, the median duration of replay events in medial prefrontal cortex (PFC) reported in Kaefer et al.[17] was 740 ms. This indicates that a significant proportion of replay events will be covered by our method. Second, our ISI was as fast as 32 ms, which corresponds to the time lag between activations reported in magnetoencephalography (MEG) studies (e.g.,[47]) and therefore might capture the important aspect of temporal separation between activation patterns well. Third, while effect sizes showed a pronounced decrease when comparing the slower conditions (2048 ms: 3.13; 512 ms: 1.36; 128 ms: 0.75, for the backwards effect of regression slopes, Fig. 3c, effect sizes indicate Cohen's $d$), accelerating sequence speeds beyond 128 ms seemed not to be associated with a comparable decrease in effect sizes (64 ms: 0.65, 32 ms: 1.00). This indicates that the sensitivity of our methods for even faster event sequences might not be catastrophically diminished. Fourth, the sequence duration of 628 ms was to a large extent due to the stimulus duration of 100 ms. Evidence from previous work using electroencephalography (EEG) suggests that the neural response to successive visual stimuli is more strongly influenced by the ISI than the stimulus duration[73,74]. Hence, we speculate that our methods may also work in cases with shorter pattern activations and thereby overall shorter sequences.

Our results deepen the understanding of our previous findings[54] in two ways. First, we provide additional empirical evidence that our sequentiality analyses based on multivariate fMRI pattern classification are indeed sensitive to fast neural event sequences. To this end, we used an experimental setup where the order of sequential events is known—in contrast to analyses of resting-state data in Schuck and Niv[54] where the order and speed of event sequences can only be assumed. Second, Schuck and Niv[54] observed forward-ordered replay. Our present study clarifies the origins of forward and backward ordering of fMRI activation patterns. We show that probabilistic classifier evidence in earlier TRs reflects the forward order of the sequences while this pattern reverses in later TRs. Importantly, we demonstrate an asymmetry in decoding early versus late sequential events. This can therefore lead fMRI pattern sequences to appear in the reverse order relative to the underlying neural sequences. This represents a crucial insight, given the different functional roles assigned to forward and backward replay (see e.g.[33]). We note that future research should be careful when interpreting directionality, as the relationship between decoded and true directionality is not straightforward. One approach in this context could be to investigate the order of sequence direction itself. If items appear to be ordered first in direction A, and a few TRs later in direction B, then direction A seems to be the true one. Probabilistic classifiers might prove particularly useful for such analyses as they make it possible to characterize sequential ordering within a single measurement. The origins of this asymmetry are not entirely clear. It seems possible that they reflect the benefits of the last item not being followed by another activation that could impede its detection. A relation to the asymmetric shape of the HRF, to changing HRF variability with

time and even to inhibitory retrograde neurotransmitters (e.g.[75]) cannot be ruled out. Third, we have shown that the interference of activation patterns of fast sequential neural events is stronger for early events compared to late events. Importantly, early events remained detectable despite this interference, demonstrating that our method can detect the elements of a replay event with fMRI despite interference effects. The prominence of the last sequence item implies that the apparent over-occurrence of one particular item might reflect that this item was a frequent start or end point of replayed trajectories. Past research has shown that task aspects, such as goals, heavily influence which items the replayed sequences start or end with[76].

In addition, our study introduces important methodological advancements that go beyond our original publication[54]. We show that the analyses of classifier probabilities provide major statistical improvements compared to analyses focused on the decoded category with the highest classification probability (as in Schuck and Niv[54]). The key advantage is that probabilistic classifiers provide a continuous metric of classification evidence and thereby allow the detection of sequential ordering within a single measurement (i.e., within a single TR). This results in significant information gain compared to the assessment of sequential ordering that considers only a single label per TR. Moreover, we leverage frequency spectrum analysis in an approach to make inferences about the speed of the sequential neural process. Although the sampling rate (i.e., the TR) of fMRI is usually less than the speed of replay events, frequency spectrum analyses can characterize the speed of fast sequential events during rest. Together, these methodological advances offer insights into previous fMRI studies investigating hippocampal replay in humans, including our own work[54].

Additionally, some caveats have to be noted. Our results indicate that the sequentiality in fMRI analyses is mainly influenced by the first and last element of a fast sequence. Given that replay events are often structured by task-relevant features like the start and goal location in a spatial environment (e.g.,[76]), analyzing the transitions between the corresponding decoded events will offer insights into the content and functional role of fast replay events. Moreover, it is important to keep in mind that the benefits of our experimental setting came at the cost that they also introduced important differences from a replay study in various regards, including the focus on extra-hippocampal activations and sensory stimulation.

Our fMRI-based approach has advantages as well as disadvantages compared to existing EEG and MEG approaches[44,46,47]. In particular, it seems likely that our method has limited resolution of sequence speed. While we could distinguish between supra- and sub-second sequences, a much finer distinction might prove difficult in practice. Yet, EEG and MEG investigations suggest that the extent of temporal compression of previous experience is an important aspect of replay and other reactivation phenomena[45,77–80]. In addition, the differential sensitivity to activity depending on sequence position complicates interpretations of findings, and can lead to statistical aliasing of sequences with the same start and end elements but different elements in the middle. Finally, because a single sequence causes forward and backward ordering of signals, it can be difficult to determine the direction of a hypothesized sequence. One major advantage of fMRI is that it does not suffer from the low sensitivity to hippocampal activity and limited ability to anatomically localize effects that characterizes EEG and MEG. This is particularly important in the case of replay, which is hippocampus-centered but co-occurs with fast neural event sequences in other parts of the brain including primary visual cortex[12], auditory cortex[15], PFC[13,14,16,17,81], entorhinal cortex[22,82,83], and ventral striatum[84]. Importantly, replay events occurring in different brain areas might not be mere copies of each other, but can differ regarding their timing, content, and relevance for cognition, e.g.[16,17]. Precise characterization of replay events occurring in different anatomical regions is therefore paramount. The present finding of fast and slow reactivations in visual cortex underlines the importance of knowing the anatomical origin of replay events. Because EEG and MEG cannot untangle the co-occurring events and animal research is often restricted to a single recording site, much remains to be understood about the distributed and coordinated nature of replay. One particular problem is that localizer tasks frequently used to train classifiers in MEG studies might only partially reflect hippocampal activity. In fact, our own data here show that simple visual tasks do not elicit reliable hippocampal activation patterns. Thus, EEG or MEG classifiers trained on such data risk to not reflect any hippocampal activity.

Finally, our study provides additional insights for future research. We have shown that the mere fact that detecting which elements were part of a sequence is beneficial if sequences mostly contain a local subset of all possible events. Thus, experimental setups with a larger number of possible events will be insightful. At the same time, a larger number of to-be-decoded events will likely impair baseline classification accuracy, which in turn impairs sequence detection. Researchers should thus take the trade-off between these two aspects into account. Moreover, several other factors could influence the success of future investigations: the sampling rate (the TR); the choice of brain region; and the properties of the resulting HRFs[23]. Whether an increased sampling rate would be beneficial for the detection of fast event sequences is difficult to predict. First, longer TRs provide better SNR as they allow more time for longitudinal magnetization. In addition, faster sampling will not affect the underlying (slow) HRF dynamics that impede the identification of temporal order of fast neural event sequences. Sampling the activation time courses at a faster rate might not reveal more information about the sequential process under investigation. Whether shorter TRs can make up for the downsides in spatial resolution and SNR therefore seems an empirical question. Moreover, the choice of brain region will impact results only if the stability of the HRF within that brain region is low, whereas between-region differences between HRF parameters might have less impact. But HRF stability is generally high[30,85–87], and previous research noting this fact has therefore already indicated possibilities of disentangling temporally close events[28–31,88,89]. Further, increased spatial resolution might improve detection due to less partial volume averaging of non-activation-related signals. Our approach has shown how multivariate and modeling approaches can help exploit these HRF properties in order to enhance our understanding of the human brain.

## Methods

**Participants**. In all, 40 young and healthy adults were recruited from an internal participant database or through local advertisement and fully completed the experiment. No statistical methods were used to predetermine the sample size but it was chosen to be larger than similar previous neuroimaging studies, e.g.[51,52,54]. Four participants were excluded from further analysis because their mean behavioral performance was below the 50% chance level in either or both the sequence and repetition trials suggesting that they did not adequately process the visual stimuli used in the task. Please note that this exclusion was based on mean behavioral performance across all conditions of sequence and repetition trials. This means that participants who, for example, performed below chance in only one of the conditions of either the sequence or the repetition trials, but above chance in all other conditions, might still be included in the final sample because their mean behavioral performance across all conditions ended up to be above the level of chance performance. Thus, the final sample consisted of 36 participants (age: M = 24.61 years, SD = 3.77 years, range: 20–35 years, 20 female, 16 male). All participants were screened for magnetic resonance imaging (MRI) eligibility during a telephone screening prior to participation and again at the beginning of each study session according to standard MRI safety guidelines (e.g., asking for metal implants, claustrophobia, etc.). None of the participants reported to have any major physical or mental health problems. All participants were required to be right-handed, to have corrected-to-normal vision, and to speak German fluently. Furthermore, only participants with a head circumference of 58 cm or less could be

included in the study. This requirement was necessary as participants' heads had to fit the MRI head coil together with MRI-compatible headphones that were used during the experimental tasks. The ethics commission of the German Psychological Society (DGPs) approved the study protocol (reference number: NS 012018). All volunteers gave written informed consent prior to the beginning of the experiments. Every participant received 40.00 Euro and a performance-based bonus of up to 7.20 Euro upon completion of the study. None of the participants reported to have any prior experience with the stimuli or the behavioral task.

## Task

*Stimuli.* All stimuli were gray-scale images of a cat, chair, face, house, and shoe taken from Haxby et al.[55] with a size of $400 \times 400$ pixels each, which have been shown to reliably elicit object-specific neural response patterns in several previous studies, e.g.,[55,58–60]. Participants received auditory feedback to signal the accuracy of their responses. A high-pitch coin sound confirmed correct responses, whereas a low-pitch buzzer sound signaled incorrect responses. The sounds were the same for all task conditions and were presented immediately after participants entered a response or after the response time had elapsed. Auditory feedback was used to anatomically separate the expected neural activation patterns of visual stimuli and auditory feedback. While auditory feedback is more likely to engage primarily temporal brain regions, visual stimuli are more likely to activate primarily occipital brain regions. We recorded the presentation time stamps of all visual stimuli and confirmed that all experimental components were presented as expected. The task was programmed in MATLAB (version R2012b; Natick, MA, USA; The MathWorks Inc.) using the Psychophysics Toolbox extensions (Psychtoolbox; version 3.0.11)[90–92] and run on a Windows XP computer with a monitor refresh-rate of 16.7 ms.

*Slow trials.* The slow trials of the task were designed to elicit object-specific neural response patterns of the presented visual stimuli. The resulting patterns of neural activation were later used to train the classifiers. In order to ensure that participants maintained their attention and processed the stimuli adequately, they were asked to perform an oddball detection task (for a similar approach, see[44,47]). Specifically, participants were instructed to press a button each time an object was presented upside-down. Participants could answer using either the left or the right response button of an MRI-compatible button box. In contrast to similar approaches, e.g.,[44,47], we intentionally did not ask participants for a response on trials with upright stimuli to avoid neural activation patterns of motor regions in our training set which could influence later classification accuracy on the test set.

Participants were rewarded with 3 cents for each oddball (i.e., stimulus presented upside-down) that was correctly identified (i.e., hit) and punished with a deduction of 3 cents for (incorrect) responses (i.e., false alarms) on non-oddball trials (i.e., when stimuli were presented upright). In case participants missed an oddball (i.e., miss), they also missed out on the reward. Auditory feedback (coin and buzzer sound for correct and incorrect responses, respectively) was presented immediately after the response (in case of hits and false alarms) or at the end of the response time limit (in case of misses) using MRI-compatible headphones (VisuaStimDigital, Resonance Technology Company Inc., Northridge, CA, USA). Correct rejections (i.e., no responses to upright stimuli) were not rewarded and were consequently not accompanied by auditory feedback. Together, participants could earn a maximum reward of 3.60 Euro in this task condition.

Across the entire experiment, all five unique images were presented in all possible sequential combinations which resulted in $5! = 120$ sequences with each of the five unique visual objects in a different order. Thus, across the entire experiment, participants were shown $120 \times 5 = 600$ visual objects in total for this task condition. Of all visual objects, 20% were presented upside-down (i.e., 120 oddball stimuli). All unique visual objects were shown upside-down equally often, which resulted in $120/5 = 24$ oddballs for each individual visual object category. The order of sequences as well as the appearances of oddballs were randomly shuffled for each participant and across both study sessions.

Each trial (for the trial procedure, see Fig. 1a) started with a waiting period of 3.85 s during which a blank screen was presented. This ITI ensured a sufficient time delay between each slow trial and the preceding trial (either a sequence or a repetition trial). The five visual object stimuli of the current trial were then presented as follows: after the presentation of a short fixation dot for a constant duration of 300 ms, a stimulus was shown for a fixed duration of 500 ms followed by a variable ISI during which a blank screen was presented again. The duration of the ISI for each trial was randomly drawn from a truncated exponential distribution with a mean of 2.5 s and a lower limit of 1 s. We expected that neural activation patterns elicited by the stimuli can be well recorded during this average time period of 3 s (for a similar approach, see[55]). Behavioral responses were collected during a fixed time period of 1.5 s after each stimulus onset. In case participants missed an oddball target, the buzzer sound (signaling an incorrect response) was presented after the response time limit had elapsed. Only neural activation patterns related to correct trials with upright stimuli were used to train the classifiers. Slow trials were interleaved with sequence and repetition trials such that each of the 120 slow trials was followed by either one of the 75 sequence trials or 45 repetition trials (details on these trial types are given below).

*Sequence trials.* In the sequence trials of the task, participants were shown sequences of the same five unique visual objects at varying presentation speeds. In total, 15 different sequences were selected for each participant. Sequences were chosen such that each visual object appeared equally often at the first and last position of the sequence. Given five stimuli and 15 sequences, for each object category this was the case for 3 out of the 15 sequences. Furthermore, we ensured that all possible sequences were chosen equally often across all participants. Given 120 possible sequential combinations in total, the sequences were distributed across eight groups of participants. Sequences were randomly assigned to each participant following this pseudo-randomized procedure.

To investigate the influence of sequence presentation speed on the corresponding neural activation patterns, we systematically varied the ISI between consecutive stimuli in the sequence. Specifically, we chose five different speed levels of 32, 64, 128, 512, and 2048 ms, respectively (i.e., all exponents of 2 for good coverage of faster speeds). Each of the 15 sequences per participant was shown at each of the 5 different speed levels. The occurrence of the sequences was randomly shuffled for each participant and across sessions within each participant. This resulted in a total of 75 sequence trials presented to each participant across the entire experiment. To ensure that participants maintained attention to the stimuli during the sequence trials, they were instructed to identify the serial position of a previously cued target object within the shown stimulus sequence and indicate their response after a delay period without visual input.

During a sequence trial (for the trial procedure, see Fig. 1b) the target cue (the name of the visual object, e.g., *shoe*) was shown for a fixed duration of 1000 ms, followed by a blank screen for a fixed duration of 3850 ms. A blank screen was used to reduce possible interference of neural activation patterns elicited by the target cue with neural response patterns following the sequence of visual objects. A short presentation of a gray fixation dot for a constant duration of 300 ms signaled the onset of the upcoming sequence of visual objects. All objects in the sequence were presented briefly for a fixed duration of 100 ms. The ISI for each trial was determined based on the current sequence speed (see details above) and was the same for all stimuli within a sequence. The sequence of stimuli was followed by a delay period with a gray fixation dot that was terminated once a fixed duration of 16 s since the onset of the first sequence object had elapsed. This was to ensure sufficient time to acquire the aftereffects of neural responses following the sequence of objects even at a sequence speed of 2048 ms. During this waiting period, participants were listening to bird sounds in order to keep them moderately entertained without additional visual input. Subsequently, the name of the target object as well as the response mapping was presented for a fixed duration of 1.5 s (same fixed response time limit as for the slow trials, see above). In this response interval, participants had to choose the correct serial position of the target object from two response options that were presented on the left and right side of the screen. The mapping of the response options was balanced for left and right responses (i.e., the correct option appeared equally often on the left and right side; 37 times each with the mapping of the last trial being determined randomly) and shuffled randomly for every participant. The serial position of the target for each trial was randomly drawn from a Poisson distribution with $\lambda = 1.9$ and truncated to an interval from 1 to 5. Thus, across all trials, the targets appeared more often at the later compared to earlier positions of the sequence. This was done to reduce the likelihood that participants stopped to process stimuli or diverted their attention after they identified the position of the target object. The serial position of the alternative response option was drawn from the same distribution as the serial position of the target. As for the slow trials, auditory feedback was presented immediately following a response. The coin sound indicated a reward of 3 cents for correct responses, whereas the buzzer sound signaled incorrect or missed responses (however, there was no deduction of 3 cents for incorrect responses or misses). Together, participants could earn a maximum reward of 2.25 Euro in this task condition.

*Repetition trials.* We included so-called *repetition trials* to investigate how decoding time courses would be affected by (1) the number of fast repetitions of the same neural event and (2) their interaction with the position of the switch to a subsequent stimulus category. Repetition trials included varying repetitions of two images in a sequence of nine items in total. All analyses reported in the Results section focused on the two most extreme cases, (1) the first image shown once followed by eight repetitions of the second image, and (2) eight repetitions of the first image followed by the second image shown once. Analyses of all intermediate levels of repetitions are reported in the SI. Each of the five stimulus categories was selected as the preceding stimulus for eight sequences in total. For each of these eight sequences, we systematically varied the position of the switch to the second stimulus category from serial position 2 to 9. Overall, the transition to the second stimulus happened five times at each serial position with varying stimulus material on each trial. Across the eight trials for each stimulus category, we ensured that each preceding stimulus category was followed by each of the remaining four stimulus categories equally often. Specifically, a given preceding stimulus category was followed by each of the remaining four stimulus categories two times. Also, the average serial position of the first occurrence of each of the subsequent stimuli was the same for all subsequent stimuli. That is to say, the same subsequent stimulus appeared either on position 9 and 2, 8 and 3, 7 and 4, or 6 and 5, resulting in an average first occurrence of the subsequent stimulus at position 5.5. All stimulus sequences of the repetition trials were presented with a fixed ISI of 32 ms. Note that

this is the same presentation speed as the fastest ISI of the sequence trials. Similar to the sequence trials, participants were instructed to remember the serial position at which the second stimulus within the sequence appeared for the first time. For example, if the switch to the second stimulus happened at the fifth serial position, participants had to remember this number.

Similar to the trial procedure of the sequence trials, each repetition trial (Fig. 1c) began with the presentation of the target cue (name of the visual object, e.g., *cat*), which was shown for a fixed duration of 500 ms. The target cue was followed by a blank screen that was presented for a fixed duration of 3.85 s. A briefly presented fixation dot announced the onset of the sequential visual stimuli. Subsequently, the fast sequence of visual stimuli was presented with a fixed duration for visual stimuli (100 ms each) and the ISI (32 ms on all trials). As for sequence trials, the sequence of stimuli on repetition trials was followed by a variable delay period until 16 s from sequence onset had elapsed. On repetition trials, participants had to choose the correct serial position of the first occurrence of the target stimulus from two response options. The incorrect response option was a random serial position that was at least two positions away from the correct target position. For example, if the correct option was 5, the alternative target position could either be earlier (1, 2, or 3) or later (7, 8, or 9). This was done to ensure that the task was reasonably easy to perform. Finally, we added five longer repetition trials with 16 elements per sequence. Here, the switch to the second sequential stimulus always occurred at the last serial position. Each of the five stimulus categories was the preceding stimulus once. The second stimulus of each sequence was any of the other four stimulus categories. In doing so, in the long repetition trials each stimulus category was the preceding and subsequent stimulus once. Repetition trials were randomly distributed across the entire experiment and (together with the sequence trials) interleaved with the slow trials.

**Study procedure**. The study consisted of two experimental sessions. During the first session, participants were informed in detail about the study, screened for MRI eligibility, and provided written informed consent if they agreed to participate in the study. Then they completed a short demographic questionnaire (assessing age, education, etc.) and a computerized version of the Digit-Span Test, assessing working memory capacity[93]. Next, they performed a 10-min practice of the main task. Subsequently, participants entered the MRI scanner. After a short localizer, we first acquired a 5-min resting-state scan for which participants were asked to stay awake and focus on a white fixation cross presented centrally on a black screen. Then, we acquired four functional task runs of about 11 min during which participants performed the main task in the MRI scanner. After the functional runs, we acquired another 5-min resting-state, 5-min fieldmaps, as well as a 4-min anatomical scan. The second study session was identical to the first session, except then participants entered the scanner immediately after another short assessment of MRI eligibility. In total, the study took about 4 h to complete (2.5 and 1.5 h for Session 1 and 2, respectively).

**MRI data acquisition**. All MRI data were acquired using a 32-channel head coil on a research-dedicated 3-Tesla Siemens Magnetom TrioTim MRI scanner (Siemens, Erlangen, Germany) located at the Max Planck Institute for Human Development in Berlin, Germany. The scanning procedure was exactly the same for both study sessions. For the functional scans, whole-brain images were acquired using a segmented k-space and steady-state T2*-weighted multi-band (MB) echo-planar imaging (EPI) single-echo gradient sequence that is sensitive to the BOLD contrast. This measures local magnetic changes caused by changes in blood oxygenation that accompany neural activity (sequence specification: 64 slices in interleaved ascending order; anterior-to-posterior (A–P) phase-encoding direction; TR = 1250 ms; echo time (TE) = 26 ms; voxel size = 2 × 2 × 2 mm; matrix = 96 × 96; field of view (FOV) = 192 × 192 mm; flip angle (FA) = 71°; distance factor = 0%; MB acceleration factor = 4). Slices were tilted for each participant by 15° forwards relative to the rostro-caudal axis to improve the quality of fMRI signal from the hippocampus (cf.[94]) while preserving good coverage of occipito-temporal brain regions. Each MRI session included four functional task runs. Each run was about 11 min in length, during which 530 functional volumes were acquired. For each functional run, the task began after the acquisition of the first four volumes (i.e., after 5 s) to avoid partial saturation effects and allow for scanner equilibrium. We also recorded two functional runs of resting-state fMRI data, one before and one after the task runs. Each resting-state run was about 5 min in length, during which 233 functional volumes were acquired. After the functional task runs, two short acquisitions with six volumes each were collected using the same sequence parameters as for the functional scans but with varying phase-encoding polarities, resulting in pairs of images with distortions going in opposite directions between the two acquisitions (also known as the *blip-up/blip-down* technique). From these pairs the displacement maps were estimated and used to correct for geometric distortions due to susceptibility-induced field inhomogeneities as implemented in the fMRIPrep preprocessing pipeline[95]. In addition, a whole-brain spoiled gradient recalled (GR) field map with dual echo-time images (sequence specification: 36 slices; A–P phase-encoding direction; TR = 400 ms; TE1 = 4.92 ms; TE2 = 7.38 ms; FA = 60°; matrix size = 64 × 64; FOV = 192 × 192 mm; voxel size = 3 × 3 × 3.75 mm) was obtained as a potential alternative to the method described above. However, as this field map data were not successfully recorded for four participants, we used the *blip-up/blip-down* technique for distortion correction (see details on MRI data preprocessing

below). Finally, high-resolution T1-weighted (T1w) anatomical Magnetization Prepared Rapid Gradient Echo (MPRAGE) sequences were obtained from each participant to allow registration and brain-surface reconstruction (sequence specification: 256 slices; TR = 1900 ms; TE = 2.52 ms; FA = 9°; inversion time (TI) = 900 ms; matrix size = 192 × 256; FOV = 192 × 256 mm; voxel size = 1 × 1 × 1 mm). We also measured respiration and pulse during each scanning session using pulse oximetry and a pneumatic respiration belt.

**MRI data preparation and preprocessing**. Results included in this manuscript come from preprocessing performed using *fMRIPrep* 1.2.2 (Esteban et al.[95,96]; RRID:SCR_016216), which is based on *Nipype* 1.1.5 (Gorgolewski et al.[97,98]; RRID: SCR_002502). Many internal operations of *fMRIPrep* use *Nilearn* 0.4.2[99]; RRID: SCR_001362, mostly within the functional processing workflow. For more details of the pipeline, see https://fmriprep.readthedocs.io/en/1.2.2/workflows.html the section corresponding to workflows in *fMRIPrep*'s documentation.

*Conversion of data to the brain imaging data structure (BIDS) standard*. The majority of the steps involved in preparing and preprocessing the MRI data employed recently developed tools and workflows aimed at enhancing standardization and reproducibility of task-based fMRI studies, for a similar preprocessing pipeline, see[100]. Following successful acquisition, all study data were arranged according to the BIDS specification[101] using the HeuDiConv tool (version 0.6.0. dev1; freely available from https://github.com/nipy/heudiconv) running inside a Singularity container[102,103] to facilitate further analysis and sharing of the data. Dicoms were converted to the NIfTI-1 format using dcm2niix (version 1.0.20190410 GCC6.3.0)[104]. In order to make identification of study participants unlikely, we eliminated facial features from all high-resolution structural images using pydeface (version 2.0; available from https://github.com/poldracklab/pydeface). The data quality of all functional and structural acquisitions was evaluated using the automated quality assessment tool MRIQC (for details, see[105], and the https://mriqc.readthedocs.io/en/stable/MRIQC documentation. The visual group-level reports of the estimated image quality metrics confirmed that the overall MRI signal quality of both anatomical and functional scans was highly consistent across participants and runs within each participant.

*Preprocessing of anatomical MRI data*. A total of two T1-weighted images were found within the input BIDS dataset, one from each study session. All of them were corrected for intensity non-uniformity (INU) using N4BiasFieldCorrection (Advanced Normalization Tools (ANTs) 2.2.0)[106]. A T1w-reference map was computed after registration of two T1w images (after INU-correction) using mri_robust_template (FreeSurfer 6.0.1)[107]. The T1w reference was then skull-stripped using antsBrainExtraction.sh (ANTs 2.2.0), using OASIS as target template. Brain surfaces were reconstructed using recon-all (FreeSurfer 6.0.1,RRID:SCR_001847)[108], and the brain mask estimated previously was refined with a custom variation of the method to reconcile ANTs-derived and FreeSurfer-derived segmentations of the cortical gray-matter of Mindboggle (RRID:SCR_002438)[109]. Spatial normalization to the ICBM 152 Nonlinear Asymmetrical template version 2009c[110] (RRID:SCR_008796) was performed through nonlinear registration with antsRegistration (ANTs 2.2.0,RRID: SCR_004757)[111], using brain-extracted versions of both T1w volume and template. Brain tissue segmentation of cerebrospinal fluid (CSF), white-matter (WM), and gray-matter (GM) was performed on the brain-extracted T1w using fast (FSL 5.0.9,RRID:SCR_002823)[112].

*Preprocessing of functional MRI data*. For each of the BOLD runs found per participant (across all tasks and sessions), the following preprocessing was performed. First, a reference volume and its skull-stripped version were generated using a custom methodology of *fMRIPrep*. The BOLD reference was then co-registered to the T1w reference using bbregister (FreeSurfer) which implements boundary-based registration[113]. Co-registration was configured with nine degrees of freedom to account for distortions remaining in the BOLD reference. Head-motion parameters with respect to the BOLD reference (transformation matrices, and six corresponding rotation and translation parameters) are estimated before any spatiotemporal filtering using mcflirt (FSL 5.0.9)[114]. BOLD runs were slice-time-corrected using 3dTshift from AFNI 20160207[115] (RRID:SCR_005927). The BOLD time-series (including slice-timing correction) were resampled onto their original, native space by applying a single, composite transform to correct for head-motion and susceptibility distortions. These resampled BOLD time-series will be referred to as preprocessed BOLD in original space, or just preprocessed BOLD. The BOLD time-series were resampled to MNI152NLin2009cAsym standard space, generating a preprocessed BOLD run in MNI152NLin2009cAsym space. First, a reference volume and its skull-stripped version were generated using a custom methodology of *fMRIPrep*. Several confounding time-series were calculated based on the preprocessed BOLD: frame-wise displacement (FD), DVARS, and three region-wise global signals. FD and DVARS are calculated for each functional run, both using their implementations in *Nipype* (following the definitions by Power et al.)[116]. The three global signals are extracted within the CSF, the WM, and the whole-brain masks. Additionally, a set of physiological regressors were extracted to allow for component-based noise correction (*CompCor*)[117]. Principal components are estimated after high-pass filtering the preprocessed BOLD time-series (using a

discrete cosine filter with 128 s cut-off) for the two *CompCor* variants: temporal (*tCompCor*) and anatomical (*aCompCor*). Six *tCompCor* components are then calculated from the top 5% variable voxels within a mask covering the subcortical regions. This subcortical mask is obtained by heavily eroding the brain mask, which ensures it does not include cortical GM regions. For *aCompCor*, six components are calculated within the intersection of the aforementioned mask and the union of CSF and WM masks calculated in T1w space, after their projection to the native space of each functional run (using the inverse BOLD-to-T1w transformation). The head-motion estimates calculated in the correction step were also placed within the corresponding confounds file. The BOLD time-series were resampled to surfaces on the following spaces: fsnative, fsaverage. All resamplings can be performed with *a single interpolation step* by composing all the pertinent transformations (i.e., head-motion transform matrices, susceptibility distortion correction when available, and co-registrations to anatomical and template spaces). Gridded (volumetric) resamplings were performed using antsApplyTransforms (ANTs), configured with Lanczos interpolation to minimize the smoothing effects of other kernels[118]. Non-gridded (surface) resamplings were performed using mri_- vol2surf (FreeSurfer). Following preprocessing using fMRIPrep, the fMRI data were spatially smoothed using a Gaussian mask with a standard deviation (full-width at half-maximum (FWHM) parameter) set to 4 mm using an example Nipype smoothing workflow (see the Nipype documentation for details) based on the SUSAN algorithm as implemented in the FMRIB Software Library (FSL)[119].

## Multivariate fMRI pattern analysis

*Leave-one-run-out cross-validation procedure*. All fMRI pattern classification analyses were conducted using open-source packages from the Python (Python Software Foundation, Python Language Reference, version 3.7) modules Nilearn (version 0.5.0)[99] and scikit-learn (version 0.20.3)[120]. fMRI pattern classification was performed using a leave-one-run-out cross-validation procedure for which data from seven task runs were used for training and data from the left-out run (i.e., the eighth run) were used for testing. This procedure was repeated eight times so that each task run served as the testing set once. We trained an ensemble of five independent classifiers, one for each of the five stimulus classes (cat, chair, face, house, and shoe). For each class-specific classifier, labels of all other classes in the data were relabeled to a common *other* category. In order to ensure that the classifier estimates were not biased by relative differences in class frequency in the training set, the weights associated with each class were adjusted inversely proportional to the class frequencies in each training fold. Given that there were five classes to decode, the frequencies used to adjust the classifiers' weights were 1/5 for the class of interest, and 4/5 for the *other* class, comprising any other classes. Adjustments to minor imbalances caused by the exclusion of erroneous trials were performed in the same way. Training was performed on data from all trials of the seven runs in the respective cross-validation fold using only the trials of the slow task where the visual object stimuli were presented upright and participants did not respond correctly (i.e., correct rejection trials). In each iteration of the classification procedure, the classifiers trained on seven out of eight runs were then applied separately to the data from the left-out run. Specifically, the classifiers were applied to (1) data from the slow trials of the left-out run, selecting volumes capturing the expected activation peaks to determine classification accuracy, (2) data from the slow trials of the left-out run, selecting all volumes from stimulus onset to the end of the trial (seven volumes in total per trial) to identify temporal dynamics of classifier predictions on a single trial basis, (3) data from the sequence trials of the left-out run, selecting all volumes from sequence onset to the end of the delay period (13 volumes in total per trial), and (4) data from the repetition trials of the left-out run, also selecting all volumes from sequence onset to the end of the delay period (13 volumes in total per trial). When the classifiers were applied to sequence and repetition trials, data from both accurate and inaccurate trials were used to allow for an equal number of test trials across participants and maximize statistical power within the current study design. As shown in Fig. 1e, f, behavioral performance on sequence and repetition trials was high and significantly above chance.

We used separate multinomial logistic regression classifiers with identical parameter settings. All classifiers were regularized using L2 regularization. The *C* parameter of the cost function was fixed at the default value of 1.0 for all participants. The classifiers employed the lbfgs algorithm to solve the multi-class optimization problem and were allowed to take a maximum of 4000 iterations to converge. Pattern classification was performed within each participant separately, never across participants. For each stimulus in the training set, we added 4 s to the stimulus onset and chose the volume closest to that time point (i.e., rounded to the nearest volume) to center the classifier training on the expected peaks of the BOLD response (for a similar approach, see, e.g.[49]). At a TR of 1.25 s, this corresponded to the fourth MRI volume which thus compromised a time window of 3.75–5 s after each stimulus onset. We detrended the fMRI data separately for each run across all task conditions to remove low-frequency signal intensity drifts in the data due to signal noise from the MRI scanner. For each classifier and run, the features were standardized (*z*-scored) by removing the mean and scaling to unit variance separately for each test set.

For fMRI pattern classification analysis performed on resting-state data, we created a new mask for each participant through additive combination of the eight masks used for cross-validation (see above). This mask was then applied to all task and resting-state fMRI runs which were then separately detrended and

standardized (*z*-scored). The classifiers were trained on the peak activation patterns from all slow trials combined.

*Feature selection*. Feature selection is commonly used in multi-voxel pattern analysis (MVPA) to determine the voxels constituting the activation patterns used for classification in order to improve the predictive performance of the classifier[121,122]. Here, we combined a functional ROI approach based on thresholded *t*-maps with anatomical masks to select image-responsive voxels within a predefined anatomical brain region.

We ran eight standard first-level general linear models (GLMs) for each participant, one for each of the eight cross-validation folds using SPM12 (version 12.7219; https://www.fil.ion.ucl.ac.uk/spm/software/spm12/) running inside a Singularity container built using neurodocker (version 0.7.0; https://github. com/ReproNim/neurodocker) implemented in a custom analysis workflow using Nipype (version 1.4.0)[97]. In each cross-validation fold, we fitted a first-level GLM to the data in the training set (e.g., data from run 1 to 7) and modeled the stimulus onset of all trials of the slow task when a stimulus was presented upright and was correctly rejected (i.e., participants did not respond correctly). These trial events were modeled as boxcar functions with the length of the modeling event corresponding to the duration of the stimulus on the screen (500 ms for all events). If present in the training data, we also included trials with hits (correct response to upside-down stimuli), misses (missed response to upside-down stimuli), and false alarms (incorrect response to upright stimuli) as regressors of no interest, thereby explicitly modeling variance attributed to these trial types (cf.[123]). Finally, we included the following nuisance regressors estimated during preprocessing with fMRIPrep: the frame-wise displacement for each volume as a quantification of the estimated bulk-head-motion, the six rigid-body motion-correction parameters estimated during realignment (three translation and rotation parameters, respectively), and six noise components calculated according to the anatomical variant of *CompCorr* (for details, see[95], and the https://fmriprep.readthedocs.io/en/ stable/ fMRIPrep documentation). All regressors were convolved with a canonical HRF and did not include model derivatives for time and dispersion. Serial correlations in the fMRI time-series were accounted for using an autoregressive AR (1) model. This procedure resulted in fold-specific maps of *t*-values that were used to select voxels from the left-out run of the cross-validation procedure. Note that this approach avoids circularity (or so-called *double-dipping*) as the selective analysis (here, fitting of the GLMs to the training set) is based on data that are fully independent from the data that voxels are later selected from (here, testing set from the left-out run; cf.[124]).

The resulting brain maps of voxel-specific *t*-values resulting from the estimation of the described *t*-contrast were then combined with an anatomical mask of occipito-temporal brain regions. All participant-specific anatomical masks were created based on automated anatomical labeling of brain-surface reconstructions from the individual T1w-reference image created with Freesurfer's recon-all[108] as part of the fMRIPrep workflow[95], in order to account for individual variability in macroscopic anatomy and to allow reliable labeling[125,126]. For the anatomical masks of occipito-temporal regions we selected the corresponding labels of the cuneus, lateral occipital sulcus, pericalcarine gyrus, superior parietal lobule, lingual gyrus, inferior parietal lobule, fusiform gyrus, inferior temporal gyrus, parahippocampal gyrus, and the middle temporal gyrus (cf.[55]). Only gray-matter voxels were included in the generation of the masks as BOLD signal from non-gray-matter voxels cannot be generally interpreted as neural activity[122]. However, note that due to the whole-brain smoothing performed during preprocessing, voxel activation from brain regions outside the anatomical mask but within the sphere of the smoothing kernel might have entered the anatomical mask (thus, in principle, also including signal from surrounding non-gray-matter voxels).

Finally, we combined the *t*-maps derived in each cross-validation fold with the anatomical masks. All voxels with *t*-values above or below a threshold of $t = 3$ (i.e., voxels with the most negative and most positive *t*-values) inside the anatomical mask were then selected for the left-out run of the classification analysis and set to 1 to create the final binarized masks ($M = 11{,}162$ voxels on average, $SD = 2{,}083$).

*Classification accuracy and multivariate decoding time courses*. In order to assess the classifiers' ability to differentiate between the neural activation patterns of individual visual objects, we compared the predicted visual object of each example in the test set to the visual object that was actually shown to the participant on the corresponding trial. We obtained an average classification accuracy score for each participant by calculating the mean proportion of correct classifier predictions across all correctly answered, upright slow trials (Fig. 2a). The mean accuracy scores of all participants were then compared to the chance baseline of $100\%/5 = 20\%$ using a one-sided one-sample *t*-test, testing the a priori hypothesis that classification accuracy would be higher than the chance baseline. The effect size (Cohen's *d*) was calculated as the difference between the mean of accuracy scores and the chance baseline, divided by the standard deviation of the data[127]. Furthermore, we assessed the classifiers' ability to accurately detect the presence of visual objects on a single trial basis. For this analysis, we applied the trained classifiers to seven volumes from the volume closest to the stimulus onset and examined the time courses of the probabilistic classification evidence in response to the visual stimuli on a single trial basis (Fig. 2b). In order to test if the time series of classifier probabilities reflected the expected increase of classifier probability for the

stimulus shown on a given trial, we compared the time series of classifier probabilities related to the classified class with the mean time courses of all other classes using a two-sided paired $t$-test at every time point (i.e., at every TR). Here, we used the Bonferroni-correction method[128] across time points and stimulus classes to adjust for multiple comparisons of 35 observations (7 TRs and 5 stimulus classes). In the main text, we only report the results for the peak in classification probability of the true class, corresponding to the fourth TR after stimulus onset. The effect size (Cohen's $d$) was calculated as the difference between the means of the probabilities of the current versus all other stimuli, divided by the standard deviation of the difference[127].

*Response and difference function modeling.* As reported above, analyzing probabilistic classifier evidence on single slow trials revealed multivariate decoding time courses that can be characterized by a slow response function that resembles single-voxel hemodynamics. For simplicity, we modeled this response function as a sine wave that was flattened after one cycle, scaled by an amplitude, and adjusted to baseline. The model was specified as follows:

$$h(t) = \frac{A}{2}\sin(2\pi f t - 2\pi f d - 0.5\pi) + b + \frac{A}{2} \tag{1}$$

whereby $t$ is time, $A$ is the response amplitude (the peak deviation of the function from baseline), $f$ is the angular frequency (unit: 1/ TR, i.e., 0.8 Hz), $d$ is the onset delay (in TRs), and $b$ is the baseline (in %). The restriction to one cycle was achieved by converting the sine wave in accordance with the following piece-wise function:

$$H(t) = \begin{cases} h(t) & \text{if } d \leq t \leq (d + \frac{1}{f}) \\ b & \text{otherwise} \end{cases} \tag{2}$$

We fitted the four model parameters ($A$, $f$, $d$, and $b$) to the mean probabilistic classifier evidence of each stimulus class at every TR separately for each participant. For convenience, we count time $t$ in TRs. To approximate the time course of the difference between two response functions, we utilized the trigonometric identity for the subtraction of two sine functions, e.g.[129]:

$$\cos(z_1) - \cos(z_2) = -2\sin\left(\frac{z_1 + z_2}{2}\right)\sin\left(\frac{z_1 - z_2}{2}\right) \tag{3}$$

Considering the case of two sine waves with identical frequency but differing by a temporal shift $\delta$ one obtains

$$\cos(2\pi f t) - \cos(2\pi f t - 2\pi f \delta) = -2\sin\left(\frac{4\pi f t - 2\pi f \delta}{2}\right)\sin\left(\frac{2\pi f \delta}{2}\right)$$
$$= -2\sin\left(2\pi f \frac{\delta}{2}\right)\sin\left(2\pi f t - 2\pi f \frac{\delta}{2}\right) \tag{4}$$

which corresponds to a flipped sine function with an amplitude scaled by $2\sin(2\pi f\frac{\delta}{2})$, a shift of $\frac{\delta}{2}$ and an identical frequency $f$.

To apply this equation to our scenario, two adjustments have to be made since the single-cycle nature of our response function is not accounted for in Eq. (3). First, one should note that properties of the amplitude term in Eq. (4) only hold as long as shifts of no greater than half a wavelength are considered (the wavelength $\lambda$ is the inverse of the frequency $f$). The term $\sin(2\pi f\frac{\delta}{2})$ can be written as $\sin(2\pi\frac{\delta}{2\lambda})$, which illustrates that the term monotonically increases until $\delta > \frac{1}{2}\lambda$. Second, the frequency term has to be adapted as follows: The flattening of the sine waves to the left implies that the difference becomes positive at 0 rather than $\frac{\delta}{2}$, thus undoing the phase shift and stretching the wave by $\frac{1}{2}\delta$ TRs. The flattening on the right also leads to a lengthening of the wave by an additional $\frac{1}{2}\delta$ TRs, since the difference becomes 0 at $2\pi f + 2\pi f\delta$, instead of only $2\pi f + 2\pi f\frac{\delta}{2}$. Thus, the total wavelength has to be adjusted by a factor of $\delta$ TRs, and no phase shift relative to the first response is expected. The difference function therefore has frequency

$$f_\delta = (f^{-1} + \delta)^{-1} = \frac{f}{1 + f\delta} \tag{5}$$

instead of $f$, and Eq. (4) becomes $-2A\sin(2\pi f\frac{\delta}{2})\sin(2\pi\frac{f}{1+f\delta}t)$. We can now apply Eq. (3) to the fitted response function as follows:

$$\begin{aligned} h_\delta(t) &= \left(\frac{1}{2}\hat{A}\cos(2\pi\hat{f}t - 2\pi\hat{f}\hat{d} - 0.5\pi) + \hat{b} + \frac{1}{2}\hat{A}\right) \\ &\quad - \left(\frac{1}{2}\hat{A}\cos(2\pi\hat{f}t - 2\pi\hat{f}\hat{d} - 2\pi\hat{f}\delta - 0.5\pi) + \hat{b} + \frac{1}{2}\hat{A}\right) \\ &= -\hat{A}\sin\left(2\pi\hat{f}\frac{\delta}{2}\right)\sin\left(2\pi\frac{\hat{f}}{1+\hat{f}\delta}t - 2\pi\frac{\hat{f}}{1+\hat{f}\delta}d - \pi\right) \\ &= \hat{A}\sin\left(2\pi\hat{f}\frac{\delta}{2}\right)\sin(2\pi\hat{f}_\delta t - 2\pi\hat{f}_\delta d) \end{aligned} \tag{6}$$

whereby $\hat{f}$, $\hat{d}$, $\hat{b}$, and $\hat{A}$ indicate fitted parameters. We determined the relevant TRs in the forward and backward periods for sequence trials by calculating $\delta$ depending on the sequence speed (the ISI). The resulting values for $\delta$ and corresponding forward and backward periods are shown

**Table 1 Relevant time periods depending on sequence speed.**

| Speed (in ms) | $\delta$ (in TRs) | Forward period | Backward period |
|---|---|---|---|
| 32 | 0.42 | TRs 2–4 | TRs 5–7 |
| 64 | 0.52 | TRs 2–4 | TRs 5–7 |
| 128 | 0.73 | TRs 2–4 | TRs 5–8 |
| 512 | 1.96 | TRs 2–5 | TRs 6–9 |
| 2048 | 6.87 | TRs 2–7 | TRs 8–13 |

Forward periods were calculated as $[0.56; 0.5 * \lambda_\delta + d = 0.5 * (5.24 + \delta) + 0.56]$. Backward period were calculated as $[0.5 * \lambda_\delta + d = 0.5 * (5.24 + \delta) + 0.56; \lambda_\delta + d = 5.24 + \delta + 0.56]$. $\delta$ reflects the interval between the onsets of the first and last of five sequence items that is dependent on the sequence speed (the ISI) and the stimulus duration (here, 100 ms). For example, for an ISI of 32 ms, $\delta$ (in TRs) is calculated as $(0.032 * 4 + 0.1 * 4)/1.25 = 0.42$ TRs. $d$ reflects the fitted onset delay (here, 0.56 TRs). All values were then rounded to the closest TRs resulting in the speed-adjusted time periods (two rightmost columns).

in Table 1. Model fitting was performed using NLoptr, an R interface to the NLopt library for nonlinear optimization[130] employing the COBYLA (Constrained Optimization BY Linear Approximation) algorithm (version 1.2.2.1)[131,132]. The resulting parameters were then averaged across participants, yielding the mean parameters reported in the main text. To assess if the model fitted the data reasonably, we inspected the fits of the sine-wave response function for each stimulus class and participant using individual parameters (Supplementary Fig. 4).

*Detecting sequentiality in fMRI patterns on sequence trials.* In order to analyze the neural activation patterns following the presentation of sequential visual stimuli for evidence of sequentiality, we first determined the true serial position of each decoded event for each trial. Specifically, applying the trained classifiers to each volume of the sequence trials yielded a series of predicted event labels and corresponding classification probabilities that were assigned their sequential position within the true sequence that was shown to participants on the corresponding trial.

The main question we asked for this analysis was to what extent we can infer the serial order of image sequences from relative activation differences in fMRI pattern strength within single measurements (a single TR). To this end, we applied the trained classifiers to a series of 13 volumes following sequence onset (spanning a total time window of about 16 s) on sequence trials and analyzed the time courses of the corresponding classifier probabilities related to the five image categories (Fig. 3a). Classification probabilities were normalized by dividing the probabilities by their trial-wise sum for each image class. As detailed in the task description, the time window was selected such that the neural responses to the image sequences could be fully captured without interference from upcoming trials. We examined relative differences in decoding probabilities between serial events at every time point (i.e., at every TR) and quantified the degree of sequential ordering in two different analyses.

First, we conducted a linear regression between the serial position of the five images and their classification probabilities at every TR in the relevant forward and backward period (adjusted by sequence speed) and extracted the slope of the linear regression as an index of linear association. The slopes were then averaged at every TR separately for each participant and sequence speed across data from all 15 sequence trials (Fig. 3b). Here, if later events have a higher classification probability compared to earlier events, the slope coefficient will be negative. In contrast, if earlier events have a higher classification probability compared to later events, the slope coefficient will be positive. Note that, for convenience, we flipped the sign of the mean regression slopes so that positive values indicate forward ordering and negative values indicate backward ordering. To determine if we can find evidence for significant sequential ordering of classification probabilities in the forward and backward periods, we conducted a series of ten separate two-tailed one-sample $t$-tests comparing the mean regression slope coefficients of each speed condition against zero (the expectation of no order information). All $p$ values were adjusted for ten comparisons by controlling the FDR (Fig. 3c;[133]). As an estimate of the effect size, we calculated Cohen's $d$ as the difference between the sample mean and the null value in units of the sample standard deviation[127]. As reported in the main text, we conducted the same analysis using rank correlation coefficients (Kendall's $\tau$) and the mean step size between probability-ordered events within TRs as alternative indices of linear association (for details, see SI). In order to directly compare the predicted time courses of regression slopes based on our modeling approach with the observed time courses, we computed the Pearson's correlation coefficient between the two time series, both on data averaged across participants and within each participant (Fig. 2d, e). The mean within-participant correlation coefficients were tested against zero (the expectation of no correlation) using a separate two-sided one-sample $t$-test for each speed condition. All $p$ values were adjusted for five comparisons by controlling the FDR[133].

We hypothesized that sequential order information of fast neural events will translate into order structure in the fMRI signal and successively decoded events in turn. Therefore, we analyzed the fMRI data from sequence trials for evidence of sequentiality across consecutive measurements. The analyses were restricted to the

expected forward and backward periods which were adjusted depending on the sequence speed. For each TR, we obtained the image with the most likely fMRI signal pattern based on the classification probabilities. First, we asked if we are more likely to decode earlier serial events earlier and later serial events later in the decoding time window of 13 TRs. To this end, we averaged the serial position of the most likely event at every TR, separately for each trial and participant, resulting in a time course of average serial event position across the decoding time window (Fig. 3d). We then compared the average serial event position against the mean serial position (position 3) as a baseline across participants at every time point in the forward and backward period using a series of two-sided one-sample t-tests, adjusted for 38 multiple comparisons (across all five speed conditions and TRs in the forward and backward period) by controlling the FDR[133]. These results are reported in the SI. Next, in order to assess if the average serial position differed between the forward and backward period for the five different speed conditions, we conducted a linear mixed effects (LME) model and entered the speed condition (with five levels) and trial period (forward versus backward) as fixed effects including by-participant random intercepts and slopes. Finally, we conducted a series of two-sided one-sample t-tests to assess whether the mean serial position in the forward and backward periods differed from the expected mean serial position (baseline of 3) for every speed condition (all p values adjusted for 10 comparisons using FDR-correction[133].

Second, we analyzed how this progression through the involved sequence elements affected transitions between consecutively decoded serial events. As before, we extracted the most likely pattern for each TR (i.e., the pattern with the highest classification probability), and calculated the step sizes between consecutively decoded serial events, as in Schuck and Niv[54]. For example, decoding Event 2 → Event 4 in consecutive TRs would correspond to a step size of +2, while a Event 3 → Event 2 transition would reflect a step size of −1, etc. We then calculated the mean step-size of the first (early) and second (late) halves of the forward and backward periods, respectively, which were adjusted for sequence speed. Specifically, the transitions were defined as follows: at speeds of 32, 64, and 128 ms these transitions included the 2 → 3 (early forward), 3 → 4 (late forward), 5 → 6 (early backward), and 6 → 7 (late backward); at speeds of 512 ms these transitions included 2 → 3 (early forward), 4 → 5 (late forward), 6 → 7 (early backward), and 8 → 9 (late backward); at 2048 ms these transitions included 2 → 3 → 4 (early forward), 5 → 6 → 7 (late forward), 8 → 9 → 10 (early backward), and 11 → 12 → 13 (late backward). Finally, we compared the mean step size in the early and late half of the forward versus backward period for every speed condition using ten separate two-sided one-sample t-tests. All p values were adjusted for multiple comparisons by controlling the FDR (cf.[133]).

*Analysis of repetition trials for sensitivity of within-sequence items.* Applying the classifiers trained on slow trials to data from repetition trials yielded a classification probability estimate for each stimulus class given the data at every time point (i.e., at every TR; Fig. 4a and Supplementary Fig. 9). As described in the main text, we then analyzed the classification probabilities to answer which fMRI patterns were activated during a fast sequence under conditions of extreme forward or backward interference. Specifically, sequences with forward interference entailed a brief presentation of a single image that was followed by eight repetitions of a second image; whereas backward interference was characterized by a condition where eight image repetitions were followed by a single briefly presented item. As predicted by the sine-based response functions, the relevant time period included TRs 2–7. All analyses reported in the Results section were conducted using data from these selected TRs as described. Results based on data from all TRs are reported in the SI.

First, we calculated the mean probability of each event type (first, second, and non-sequence events) across all selected TRs and trials in the relevant time period separately for each repetition condition across participants. In order to examine whether the event type (first, second, and non-sequence events) had an influence on the mean probability estimates on repetition trials, we conducted a LME model[134] and entered the event type (with three factor levels: first, second, and non-sequence events) as a fixed effect and included by-participant random intercepts and slopes (Fig. 4b). Post hoc comparisons between the means of the three factor levels were conducted using Tukey's honest significant difference (HSD) test[135].

Second, in order to jointly examine the influence of event duration (number of repetitions) and event type (first, second, and non-sequence events), we conducted a LME model[134] with fixed effects of event type (with three factor levels: first, second, and non-sequence events) and repetition condition (number of individual event repetitions with two factor levels: (1) *forward interference* trials, where one briefly presented event is followed by eight repetitions of a second event, and (2) *backward interference* trials, where eight repetitions of a first event are followed by one briefly presented second event), also adding an interaction term for the two effects. Again, the model included both by-participant random intercepts and slopes (Fig. 4c). Post hoc multiple comparisons among interacting factor levels were performed separately for each repetition condition by conditioning on each level of this factor (i.e., forward interference versus backward interference trials), using Tukey's HSD test.

Third, we asked if we are more likely to find transitions between decoded events that were part of the sequence (the two within-sequence items) compared to items that were not part of the sequence (non-sequence items). To this end, we classified each transition as follows: forward (from Event 1 to Event 2), backward (from

Event 2 to Event 1), repetitions of each sequence item, outwards (from sequence items to any non-sequence item), inwards (from non-sequence items to sequence items), outside (among non-sequence items), and repetitions among non-sequence events (the full transition matrix is shown in Fig. 4e). We then compared the average proportion of forward transitions within the sequence (i.e., decoding Event 1 → Event 2) with the average proportions of (1) transitions from sequence items to items that were not part of the sequence (outwards transitions), and (2) transitions between events not part of the sequence (outside transitions) using paired two-sample t-tests with p values adjusted for four comparisons using Bonferroni correction (Fig. 4d).

*Analysis of sparse sequence events with lower SNR.* We only used resting-state data from the first study session before participants had any experience with the task (except a short training session outside the scanner). These resting-state data could not be successfully recorded in four participants. Therefore, the analyses were restricted to N = 32 of 36 participants. Participants were instructed to rest as calmly as possible with eyes opened while focusing on a white fixation cross that was presented centrally on the screen. For decoding on resting-state data, we used the union of all eight masks created for the functional task runs during the cross-validation procedure. Logistic regression classifiers were trained on masked data from slow trials of all eight functional runs and applied to all TRs of the resting-state data, similar to our sequence trial analysis. We assigned pseudo serial positions to each run randomly for every participant, assuming one fixed event ordering. We first characterized and compared the behavior of sequence detection metrics on resting-state and concatenated sequence trial data. For sequence trials, we only considered data from TRs within the expected forward and backward periods (TRs 2–13) and focused on the fastest (32 ms) and slowest (2048 ms) speed condition. Accordingly, we restricted the resting-state data to the first 180 TRs to match it to the length of concatenated sequence trial data (15 concatenated trials of 12 TRs each). For both fast and slow sequence trials and rest data, we then calculated the standard deviation of the probabilities (Fig. 5a) as well as the slope of a linear regression between serial position and their classification probabilities (Fig. 5b, c) at every TR. We then compared both the standard deviation of probabilities and the mean regression slopes over the entire rest period with the mean regression slopes in fast (32 ms) sequence trials using two-sided paired t-tests (Fig. 5a, b; ps adjusted for four comparisons using Bonferroni correction). The effect sizes (Cohen's d) were calculated as the difference between the means of the resting and sequence data, divided by the standard deviation of the differences[127]. Given the rhythmic fluctuations of the regression slope dynamics (Figs. 2e), we calculated the frequency spectra across the resting-state and concatenated sequence trial data using the Lomb-Scargle method (using the `lsp` function from the R package `lomb`, e.g.,[61] that is suitable for unevenly sampled data, and therefore accounts for potential artifacts due to data concatenation (Fig. 5d). The resulting frequency spectra were smoothed with a running average filter with width 0.005. Next, we extracted the mean power of the frequencies for fast and slow event sequences as predicted by Eq. (5) in both resting and sequence data. For example, for a 32 ms sequence with $\delta = 0.032 * 4 + 0.1 * 5 = 0.628$, one obtains the predicted frequency as $f_\delta = \frac{f}{1+f*0.628} = 0.17$, whereby $f$ equals the fitted single trial frequency $f = 1/5.24$. The mean power at the predicted frequencies were then compared between resting as well as fast and slow sequence data using two-sided paired t-tests with p values adjusted for multiple comparisons using FDR-correction[133].

We then inserted 1–6 sequence events into the pre-task resting-state period by blending TRs during resting state with TRs recorded during fast (32 ms) or slow (2048 ms) sequence trials. Specifically, we randomly selected six sequence trials for each speed condition, without replacement. Only TRs from the relevant time period (see above; 12 TRs for both speed conditions, respectively) were blended into the resting-state data. To investigate the effects of a reduced SNR, we systematically multiplied the probabilities of the inserted sequence TRs by a factor $\kappa$ of $\frac{4}{5}, \frac{1}{2}, \frac{1}{4}, \frac{1}{8}$, or 0, step-wise reducing the signal from 80% to 0% and added these scaled probabilities to the probability time courses of the resting-state data. The resting-state data used for blending were independently sampled from non-overlapping random locations within the resting-state data of the same participant. This ensured that even in the 0 SNR condition, potential artifacts due to data concatenation were present and would therefore not impact our comparisons between SNR levels. For each combination of the number of inserts and SNR levels, we then compared the mean standard deviation of the probabilities during sequence-inserted rest with sequence-free rest using a series of two-sided paired t-tests. p values were adjusted accordingly for 30 comparisons using FDR-correction[133] and log-transformed (base 20) to make them easier to visualize (here, a log-transformed p value of 1 corresponds to p < 0.05).

Finally, we calculated the frequency spectra of sequence-inserted rest data as before, separately for data with fast and slow sequence inserts. To achieve comparable resolution obtained in the above analyses, we over-sampled the frequency space by a factor of 2. Smoothing was then applied again as before. We then calculated the relative power of each frequency compared to sequence-free rest and averaged the relative frequency spectra across participants (Fig. 5h). As before, we extracted the mean power within the predicted fast and slow frequency range (± 0.01 Hz, given the smoothing) and compared them between fast and slow sequence-inserted rest and for different numbers of inserts and SNR levels. We

then compared the relative power for each sequence-inserted rest dataset, number of inserts, and SNR level against zero (no difference from sequence-free rest) using a series of two-sided one-sample $t$-tests ($p$ values uncorrected).

*Analysis of task-related reactivations in post-task resting-state data.* We investigated whether the frequency spectrum analyses described above could be used to detect task-related reactivations of stimulus sequences in post-task resting-state data in Session 1 (i.e., after participants performed four runs of the task). As the pre-task resting-state acquisition, post-task resting-state data consisted of a 5-min fMRI run during which participants rested calmly with eyes open but without any additional task. We calculated the frequency spectra (using the Lomb-Scargle method) across the pre- and post-task resting-state data as described above (see Fig. 6a). To this end, we calculated the slopes and frequency spectra in the two resting-state runs considering all permutations of possible sequential orderings of classification probabilities at every TR, rather than assuming a random ordering (as for the sequence-inserted rest analyses described above), then averaging across all data from all permutations. We then compared pre- and post-task rest directly by calculating the relative power of the frequency spectra as the difference between pre- and post-task rest (Fig. 6b). Finally, we assessed if the power difference in the fast frequency range (0.17 Hz), indicative of fast sequential neural events, between pre- and post-task rest was specific to the sequential combinations of stimuli that participants experienced during the task. To this end, we split the data depending on whether they were created based on sequences the participants experienced more or less frequently during the task. As described above, the 15 sequences that were selected for the sequence trials for each participant were considered more frequent compared to all other sequential permutations that participants experience during the slow trials. Lastly, to examine if the increases in power in the fast frequency range were specific to the more frequent sequences, we conducted a LME model with the resting-state run (pre- vs. post-task) and the sequence frequency (less vs. more frequent) as the main fixed effects of interest, and by-participant random intercepts and slopes (Fig. 6c). Post hoc multiple comparisons among the interacting factors were performed using Tukey's HSD test.

*Statistical analysis.* Main statistical analyses were conducted using LME models employing the `lmer` function of the `lme4` package (version 1.1.21[134]) in R (version 3.6.1[136]). If not stated otherwise, all models were fit with participants considered as a random effect on both the intercept and slopes of the fixed effects, in accordance with results from Barr et al.[137] who recommend to fit the most complex model consistent with the experimental design[137]. If applicable, explanatory variables were standardized to a mean of 0 and a standard deviation of 1 before they entered the models. If necessary, we removed by-participant slopes from the random effects structure to achieve a non-singular fit of the model[137]. Models were fitted using the `BOBYQA` (Bound Optimization BY Quadratic Approximation) optimizer[138,139] with a maximum of 500,000 function evaluations and no calculation of gradient and Hessian of nonlinear optimization solution. The likelihoods of the fitted models were assessed using Type III analysis of variance (ANOVA) with Satterthwaite's method. A single-step multiple comparison procedure between the means of the relevant factor levels was conducted using Tukey's HSD test[135], as implemented in the `emmeans` package in R (version 1.3.4[136,140]). In all other analyses, we used one-sample $t$-tests if group data were compared to, e.g., a baseline, or paired $t$-tests if two samples from the same population were compared. If applicable, correction for multiple hypothesis testing was performed using the FDR-correction method[133]. If not stated otherwise, $t$-tests were two-sided and the $\alpha$ level set to 0.05.

*Analysis of behavioral data.* The main goal of the current study was to investigate the statistical properties of BOLD activation patterns following the presentation of fast visual object sequences. Therefore, attentive processing of all visual stimuli was a prerequisite to ensure that we would be able to decode neural representations of the stimuli from occipito-temporal fMRI data. If behavioral performance was low, we could expect that participants did not attend well to the stimuli. We thus calculated the mean behavioral accuracy on sequence and repetition trials and excluded all participants that had a mean behavioral accuracy below the 50% chance level (Supplementary Fig. 1a). Mean behavioral accuracy scores of the remaining participants in the final sample are reported in the main text (Fig. 1e, f) and the SI (Supplementary Fig. 1). In order to assess how well participants detected upside-down stimuli on slow trials, we conducted a one-sided one-sample $t$-test against the 50% chance level, testing the a priori hypothesis that mean behavioral accuracy would be higher than chance. Cohen's $d$ quantified the effect size and was calculated as the difference between the mean of the data and the chance level, divided by the standard deviation of the data[127]. As low performance in this task condition could be indicated by both false alarms (incorrect response to upright stimuli) and misses (missed response to upside-down stimuli), we also checked whether the frequency of false alarms and misses differed (Supplementary Fig. 1b). Furthermore, we assessed if behavioral accuracy on slow trials used for classifier training was stable across task runs (Supplementary Fig. 1c). In order to examine the effect of sequence speed on behavioral accuracy in sequence trials, we conducted a LME model including the sequence speed condition as the main fixed effect of interest, and by-participant random intercepts and slopes (Fig. 1e). We then examined whether performance was above chance for all five speed conditions

and conducted five separate one-sided one-sample $t$-tests testing the a priori hypothesis that mean behavioral accuracy would be higher than a 50% chance level. All $p$ values were adjusted for multiple comparisons using FDR-correction[133]. The effect of serial position of the cued target image on behavioral accuracy is reported in the SI (Supplementary Fig. 1d). For repetition trials with forward and backward interference we conducted separate one-sided one-sample $t$-test for each repetition condition to test the a priori hypothesis that behavioral accuracy would be higher than the 50% chance level (Fig. 1f). Results for all repetition conditions are reported in the SI (Supplementary Fig. 1e). The effect sizes (Cohen's $d$) were calculated as for slow trials.

**Reporting summary.** Further information on research design is available in the Nature Research Reporting Summary linked to this article.

## Data availability

We publicly share all data used in this study. Data and code management was realized using DataLad [version 0.13.0[142], for details, see https://www.datalad.org/]. An overview of all the resources is publicly available on our project website: https://wittkuhn.mpib.berlin/highspeed/. All individual datasets can be found at https://gin.g-node.org/lnnrtwttkhn. Please note that each dataset is associated with a unique URL and Digital Object Identifier (DOI). We share all MRI and behavioral data adhering to the BIDS standard (cf.[101]) (https://github.com/lnnrtwttkhn/highspeed-bids; https://gin.g-node.org/lnnrtwttkhn/highspeed-bids; https://doi.org/10.12751/g-node.4ivuv8), all MRI quality metrics and reports based on MRIQC (cf.[105]) (https://github.com/lnnrtwttkhn/highspeed-mriqc; https://gin.g-node.org/lnnrtwttkhn/highspeed-mriqc; https://doi.org/10.12751/g-node.0vmyuh), all preprocessed MRI data using fMRIPrep (cf.[96,143]) (https://github.com/lnnrtwttkhn/highspeed-fmriprep; https://gin.g-node.org/lnnrtwttkhn/highspeed-fmriprep; https://doi.org/10.12751/g-node.0ft06t), all binarized anatomical masks used for feature selection (https://github.com/lnnrtwttkhn/highspeed-masks; https://gin.g-node.org/lnnrtwttkhn/highspeed-masks; https://doi.org/10.12751/g-node.omirok), all first-level GLM results used for feature selection (https://github.com/lnnrtwttkhn/highspeed-glm; https://gin.g-node.org/lnnrtwttkhn/highspeed-glm; https://doi.org/10.12751/g-node.d21zpv), all results of the multivariate decoding approach (https://github.com/lnnrtwttkhn/highspeed-decoding; https://gin.g-node.org/lnnrtwttkhn/highspeed-decoding; https://doi.org/10.12751/g-node.9zft1r), and the unprocessed data of the behavioral task acquired during MRI acquisition (https://github.com/lnnrtwttkhn/highspeed-data-behavior; https://gin.g-node.org/lnnrtwttkhn/highspeed-data-behavior; https://doi.org/10.12751/g-node.p7dabb). Bird sounds used as stimuli can be downloaded from https://audiojungle.net/item/british-bird-song-dawn-chorus/98074. The visual stimulus material is freely available from http://data.pymvpa.org/datasets/haxby2001/. The original authors of[55] hold the copyright of this dataset and made it available under the terms of the Creative Commons Attribution-Share Alike 3.0 license (see http://creativecommons.org/licenses/by-sa/3.0/for details). The images selected for the task were not modified. Source Data to reproduce the main parts of all figures are provided with this paper.

## Code availability

We share all code used in this study. An overview of all the resources is publicly available on our project website: https://wittkuhn.mpib.berlin/highspeed/. All code for the main statistical analyses can be found at https://github.com/lnnrtwttkhn/highspeed-analysis; https://gin.g-node.org/lnnrtwttkhn/highspeed-analysis; https://doi.org/10.12751/g-node.eqqdtg). All code to run the behavioral task can be found at (https://github.com/lnnrtwttkhn/highspeed-task; https://doi.org/10.5281/zenodo.4305888), Please note that we share all data listed in the Data availability section in modularized units alongside the code that created the data, usually in a dedicated `code` directory in each dataset, instead of separate data and code repositories. This approach allows to better establish the provenance of data (i.e., a better understanding which code and input data produced which output data), loosely following the DataLad YODA principles (for details, see the chapter "YODA: Best practices for data analyses in a dataset" in the DataLad handbook (version 0.13[144]), available at https://handbook.datalad.org/).

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

## Acknowledgements

This work was funded by an Independent Max Planck Research Group grant awarded to N.W.S by the Max Planck Society (M.TN.A.BILD0004) and a Starting Grant awarded to N.W.S by the European Union (ERC-2019-StG REPLAY-852669). We also acknowledge financial support by the Max Planck Institute for Human Development. We thank Eran Eldar, Sam Hall-McMaster, and Ondrej Zíka for helpful comments on a previous version of this manuscript, Gregor Caregnato for help with participant recruitment and data collection, Sonali Beckmann and Nadine Taube for assistance with MRI data acquisition, Anika Löwe for assistance with data collection, Lion Schulz for help with behavioral data analysis, Michael Krause for support with cluster computing, members of the Max Planck Research Group NeuroCode for helpful feedback throughout the project, and all participants for their participation. L.W. is a pre-doctoral fellow of the International Max Planck Research School on Computational Methods in Psychiatry and Ageing Research (IMPRS COMP2PSYCH). The participating institutions are the Max Planck Institute for Human Development, Berlin, Germany, and University College London, London, UK. For more information, see https://www.mps-ucl-centre.mpg.de/en/comp2psych.

## Author contributions

The following list of author contributions is based on the CRediT taxonomy[145]. For details on each type of author contribution, please see Brand et al.[145]. Conceptualization: L.W., N.W.S; Data curation: L.W.; Formal analysis: L.W., N.W.S; Funding acquisition: N.W.S; Investigation: L.W.; Methodology: L.W., N.W.S; Project administration: L.W., N.W.S; Resources: N.W.S; Software: L.W., N.W.S; Supervision: N.W.S; Validation: L.W., N.W.S; Visualization: L.W., N.W.S; Writing - original draft: L.W., N.W.S; Writing - review & editing: L.W., N.W.S.

## Funding

## Competing interests

The authors declare no competing interests.

## Additional information

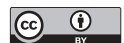

