## [Peer Review File · Nature Communications]

Reviewer #1 (Remarks to the Author):

Summary:

In this manuscript, Wittkuhn and Schuck examined whether functional magnetic resonance imaging (fMRI) can be used to measure subsecond sequences of neural activation patterns in humans, akin to hippocampal replay. The problem the authors set out to tackle is critical in advancing the study of cognitive neuroscience in humans, as we are largely limited to non-invasive measures of brain activity that force us to choose between spatial and temporal resolution. By finding ways to circumvent this trade-off, we can begin to measure psychological and neural phenomena that have previously been limited to study in patients or animal models.

Using a clever combination of experimental design, modelling and statistical analysis, Wittkuhn & Schuck demonstrate that multivariate classifiers applied to fMRI data can be used to detect the sequential activation of discrete images presented less than a second apart.

The main strengths of this paper are:

First, the authors present very thorough, convincing, and quantitative analysis;

Second, the results have broad impact, as they provide insight into (I) the temporal resolutions at which we can non-invasively classify discrete visual events and their ordering in a sequence, (II) how classification can be expected to work at different phases within a sequence of items, allowing for interference, and (III) and how these classifiers can apply to more unconstrained contexts, such as resting state, where the signal to noise ratio is considerably lower.

Although this paper presents very solid science, it does have some weaknesses which include: insufficient examination of how classification changes as a function of the human participants task state; somewhat confusing “forward” / “backward” jargon; some disconnects between the stimulus properties and the physiological and practical realities of replay.

Overall, we are confident that this manuscript will make an important contribution to the field. With that in mind, we offer the following points, which we hope can further strengthen this contribution.

Major Comments:

** Discrepancy between the duration of stimulus sequences and the motivating neural phenomena:

In the introduction, the authors refer to neural phenomena that occur on the subsecond scale (e.g., hippocampal replay: 200-300ms, orbitofrontal value representations: < 100ms) and use these as a rationale to attempt to decode rapid sequences of stimulus representations from fMRI. However, there is little discussion about the discrepancy between the duration of these neural events and the duration of the experimental sequences in this paradigm. For example, despite the ISI being as low as 32ms in the Sequence of Repetition trials, each stimulus was onscreen for 100ms, amounting to sequences that were at least 628 ms in length. To be clear, it makes sense that the authors chose to have each stimulus onscreen for 100ms, as participants may have found the task difficult to perform at shorter presentation times. However, some discussion is necessary about the marked discrepancy between the duration of the experimental sequences and the neural phenomena that motivated this

study. This is especially pertinent as the discussion explicitly mentions applying the classification approach from this manuscript to the measurement of these subsecond neural phenomena (see paragraph 2 of the Discussion).

Related to this same point: labeling the “speed” of the sequences in Figure 2E (and throughout Figure 3) as 32ms, 64ms, etc., is somewhat inaccurate, because the items in the sequence are not actually arriving at 32ms/item, 64ms/item, etc.

**** Classifying the cued vs. uncued stimulus category:**

In the Sequence and Repetition trials of this experiment, participants were cued to look for a certain stimulus in each trial (e.g., look for and remember the serial position of the shoe in a Sequence trial). However, the classification analyses on Sequence and Repetition trials were always tested on all volumes of a given trial, and averaged across trials irrespective of the category being cued. How does the classifier perform on the cued item’s position relative to positions of the uncued items? Does this change with different ISIs in the Sequence trials?

Furthermore, attending to the stimulus likely involves holding the image online (in “working memory”, perhaps) for the duration of the trial. Having the cued category in working memory could potentially boost classifier evidence for the cued stimulus category for the whole trial, or up until the cued stimulus has been presented. One potential way to control for the cued item, beyond counterbalancing its position, would be to re-run the classifier while excluding the cued-category stimulus on each trial. This way, classification cannot benefit from the explicit allocation of attention to a specific stimulus category per trial. More generally, please indicate (i) whether the cued category is decodable from the pre-stimulus period; and (ii) whether the overall effects shown in Figure 3 are different for trials when the last image in the sequence is cued or uncued.

In any case, the authors should clarify the contributions (or lack thereof) of explicit cueing to classification accuracy.

**** Use of terminology:**

The section of the results looking at the early and late halves of the forward and backwards periods was quite difficult to parse (“This sequential progression...”, In 216, p 9). Not only are the forward and backwards periods now divided into early and late phases, but the reader also has to keep the relationship between step direction (i.e., positive or negative) and where in the 5-element sequence these steps refer to. This becomes even more complicated as the reader encounters terms such as “fwd condition”, “bwd condition”, “forward-interference”, “backward-interference”, “forward transitions”, “backward transitions”, “inwards transitions”, “outwards transitions” and finally “outside transitions”. Also, the use of asymmetric language in describing the results further complicates reading (e.g., “...forward directed in the forward as compared to backward period...” as compared to “...negative directed in the forward period and vice versa in the backward period...”; see In 227-232 on p 9).

Here are some potential solutions that the authors should consider to help orient the reader through these analyses:

Add the “early” and “late” phase labels to Figure 2d and refer to this visual depiction in the results

section where these labels are introduced.

Add arrows to Figure 3h that indicate negative steps refer to earlier in the sequence, while positive steps are later in the sequence. This could look just like the gray arrows on Figure 3f.

Alternatively, might it be clearer if Figure 3h was reorganized to be plotted in time? Each speed could be a column/facet, where the X-axis for each facet has the following results in the following order: fwd-early, fwd-late, bwd-early, bwd-late. This way, the periods/phases are presented as they unfold in time and the reader doesn't have to do multi-step mental rotation to map Figure 3h back to any of the plots where the X-axis is time in TRs.

Try, whenever possible, to use symmetric language whenever possible (e.g., forward directed and backwards directed as opposed to forward directed and negative directed).

In relation to the transitions, is it possible to move the descriptions from the methods (ln 995-999, p 34-35) to the results section? The description in the results section is too brief to make sense of the various transition labels, and consequently limit how easily one can interpret, Figures 4d and e.

In relation to Figure 4e, could the diagonal simply be black and labelled "repetition", rather than being divided into 3 additional transition types? The colours are quite visually similar to one another (e.g., see inward, outward, outside, repetition out) and consolidating those three transition types into one might help reduce the figure's complexity a (meaningful) bit.

**** A gradual progression through all sequence events?:**

On page 9 (ln 232-234) the authors state: "This analysis suggests that transitions between decoded items reflect the gradual progression through all sequence events...". Is this true? The step size analysis indicates a difference in the direction of the predicted element relative to the midpoint of the sequence, which changes over time. However, considering these effects are strongly affected by the first and last element of the sequence, especially for trials with short ISIs (e.g., see Figure 3a) it seems to be a bit of an overstatement to imply strong evidence for a gradual progression through all elements in a sequence.

**** Implications for replay:**

As mentioned in our first point, the authors frame this paper in relation to fast neural sequences, such as hippocampal replay. However, it wasn't always easy to follow the implications of various insights from the authors model and experimental results on attempts to measure replay events using fMRI. For example, the authors state that "...forward replay can look like backward replay and vice versa" (ln 397-398, p 18). We agree that this is a critical insight - but further discussion of this point is warranted. Are we unable to dissociate between these two types of replay due to the properties of the hemodynamic response function we rely on in fMRI? Rather than only leaving this crucial point open to the speculation of the reader, it would be helpful if the authors provided a sentence or two that discuss this point more concretely.

Furthermore, it's not obvious how the interference results, from the repetition trials, relate to hippocampal replay events. We understand that there is a practical utility to these results, because they inform us as to the kinds of interference that arise in decoding one stimulus that is preceded (or followed) closely by another... but it is difficult to follow the practical relevance of all of this work for sequence replay (which is how the manuscript is framed as a whole).

**** Pre-task vs. post-task rest:**

From my understanding, it appears as if the resting state classification analyses were exclusively

conducted on the 5-minutes of resting state that were collected before task performance. Why was post-task rest not analyzed? It would be interesting if the power spectra for post-task rest was different from that of pre-task rest. Wouldn't the post-task resting state potentially contain task-related replay events and provide a naturalistic counterpart to the objective "sequence insertion" approach used on the pre-task rest?

Asymmetries between forward and backward interference:

Are there any electrophysiological studies that support the asymmetric interference findings reported in this paper? There is very little reference in the paper to whether any of the findings (e.g., early vs late item contributions to decoding) are consistent with the interference effects observed in electrophysiological studies of population firing rates.

Minor comments:

** Testing set:

Was the model only tested on accurate (in terms of a participant's behavioural response) Sequence and Repetition trials? It's clear that the model was only trained on accurate trials, but I might have missed if this was explicitly stated for the trials the classifier was tested on.

** Repetition trials:

I am having difficulties wrapping my head around the repetition trials. In the Methods section, the authors state:

A: "All sequences had a fixed length of nine stimuli in total." (ln 574, p 23)

B: "For each of these eight sequences we systematically varied the point of the switch to the second stimulus category from serial position 2 to 9. " (ln 575-577, p 23).

C: "Specifically, sequences with forward interference entailed a brief presentation of a single image that was followed by eight repetitions of a second image; whereas backward interference was characterized by a condition where eight image repetitions were followed by a single briefly presented item. " (ln 968-970, p 34)

How can all three of these points be true? If I understand the procedure correctly, (A) and (C) preclude the ability to have the switch between stimulus categories occur at any serial positions other than 1 or 9, right? Is it the case that the analysis in (C) only involves a small subset of the trials that are described by (A) and (B)?

Page 4 (Line 84):

"...quantifies to what extent fMRI allows to detect which elements were..."

Replace with "...allows one to detect..." or "...affords detection of the elements that..."

Page 4 (Line 97):

"with up to only 32 ms between stimuli."

Consider changing to:

"As little as 32 ms between stimuli"

Also, the wording here is somewhat misleading as the inter-onset interval is never as low as 32 milliseconds; it is only the offset-to-onset time that is as low as this.

Page 4 (Line 122):

“leave-one-run-out”

At this point in the manuscript, the reader has not yet been told what a “run” is.

Page 5 (Line 136):

“we approximated expectations for signals during sequential events”

Can this be made more concrete / intelligible?

Page 7 (Line 149):

“proportional to the ordering of events during the sequential process”

Can this be:

“proportional to the stimulus order”

?

Page 8 (Line 173):

“whether sequence order detection was evident in...”

Can this be:

“Whether sequence order was detectable from...”

?

Page 12 (Line 283):

No closing brace is required after “results”.

Page 12 (Line 297):

“...and (3) sequence item detection leads to within sequence pattern transitions.”

The language here is somewhat confusing. “Sequence item detection” is something that is done by a neuroscientist who is analyzing experimental data... it is not a neural process. How can this item detection lead to pattern transitions, which are presumably a neural process? There is something confusing about the wording here.

Page 13 (Figure 4b,c):

Couldn't these plots be simplified by replacing the bar with a single (black) dot with error bars that is placed on top of the points on the left? It's not clear what the bar plot buys you in this situation and simplifying the figure would be helpful.

Page 14:

“To account for possible SNR reductions, the inserted probability time courses were multiplied by a factor κ of $4/5$, $1/2$, $1/4$, $1/8$ or 0 and added to the probability time courses of the inversely scaled $(1 - \kappa)$ resting state data. Effectively, this led to a step-wise reduction of the inserted sequence signal from 80% to 0%, relative to the SNR obtained in the experimental conditions reported above.”

My understanding is that SNR is typically expressed as a ratio of signal-variance to noise-variance. If a raw signal is multiplied by a factor κ , then usually this would lead to a change in signal of

kappa-squared (rather than kappa)? If that is right then the kappa=0.8 setting should produce a change in SNR to 64% of its original value. It is possible I have the wrong idea, and this is a small point, but it could help to be clearer what is meant here by SNR.

Page 17:

“We did not mask stimulus sequences to establish a realistic scenario for the detection of replay events which occur spontaneously during rest with sufficient inter-event intervals that allow separation in time”

Perhaps this could be clearer as:

“In order to establish a realistic scenario for the detection of temporally separated spontaneous replay events, we did not mask stimulus sequences.”

?

Page 17:

“Second, we show that probabilistic classifiers provide major statistical improvements compared to analyses focused on the decoded category with the highest classification probability.”

I think the wording here could be improved. The key advantage seems to arise from using a continuous “strength” of evidence versus a thresholded category signal -- the fact that the strength score is expressed as a probability is (I think) inessential. The key distinction, I think, is whether or not a take-the-max threshold is applied to the classifier evidence?

Page 22:

“ During the waiting period participants were listening to bird sounds...”

Replace with “During this...” -- which will clarify that the bird sounds refer to the 16s delay period with the gray fixation.

Page 27:

“This procedure was repeated eight times so that each task run served as the training set once.”

Replace “training” with “testing”?

Page 32:

“The main question we asked for this analysis was to what extend...”

Replace “extend” with “extent”?

Figure S4:

There appears to be a mismatch between the caption text and the panel labels.

Reviewer #2 (Remarks to the Author):

In this manuscript, a multivariate approach allowing for the detection of sequentially activated neuronal patterns at a temporal spacing of less than 100ms is presented. This is a novel and exciting advancement in fMRI methodology, as classical univariate detection fails even at much slower rates. This paper opens up a new avenue in fMRI research that may rival the approaches used in EEG and MEG. The precise methodology and results should be described more clearly. In addition, more insight or perhaps thoughtful speculation into why the results were obtained - grounded perhaps in aspects unique to the HRF - should be provided throughout. Overall, it was an enjoyable paper to read with exciting results having potentially far reaching implications.

Specific Comments:

1. p. 3. An early paper for sub-TR fast event multivariate decoding likely should be mentioned for context: "Accurate decoding of sub-TR timing differences in stimulations of sub-voxel regions from multi-voxel response patterns," Misaki et al, NeuroImage, 66, 623-633, 2013.
2. p. 3, line 50: An odd recurring grammatical error should be corrected: "...have properties that allow to detect them more accurately." This perhaps should be changed to "have properties that allow more easy detection." ..or something like that, depending on the intended meaning of the sentence (which is hard to interpret). Basically "to detect" is not correct.
3. p. 4, line 65: "identified" should be "identify."
4. p. 4, line 64-66: It might be useful to mention the role that higher spatial resolution would play in improving detection. (i.e. less partial volume averaging of non-activation related signals).
5. p. 4, line 70: "allows to" is incorrect. "Allows inference of the speed.." is perhaps more correct.
6. p. 4, line 78: "to use" is incorrect. "Of using" is more correct.
7. p. 4, line 84: "to detect" should be "detection of"
8. p. 5, line 111-112: A concern in this approach is that classifiers were obtained to differentiate each out of five images - as even with re-ordering, there is likely a signature pattern for each that is a weighted composite of the five. Would it be possible to train classifiers that are independent of the set with which they are trained on and created with?
9. Figure 2. It's not clear what you mean by "forward" and "backward" period. Is it simply time before and after peak? Why call it forward and backward? Even from reading the text below, it's not clear why you chose those terms.
10. p. 8. It would be useful at this point to show an example of the classifier patterns for each image to give a sense of how different they were. Further, it would be helpful to calculate the spatial correlation of the classifiers, just as a reality check and to provide a sense of the sensitivity from a different perspective. Were there specific features within the multivariate pattern that helped to drive the discernibility?
11. p. 10, Figure 3: It would be useful to show in the figure what the TR was rather than have the x axis in units of TR without a reference to absolute time.
12. P 11. A clear difference in forward vs backward interference and detection is reported but, at least in this section and elsewhere I don't see any insight as to why this is. I wonder if backwards masking plays a role. Another possibility is the variability of the hemodynamic response as a function of time. Many studies have shown that the HRF shows significantly less variability during the rising phase and more variability during the falling phase. Another question related to sensitivity is the number of voxels used in the classifiers.
13. In the paper, it is difficult to relate the modulations in SNR to SNR in real data. Typically, SNR is

calculated as the average time course signal divided by the standard deviation over time. How was your use of SNR calculated? It should be mentioned here so that tying into real-world situations would be more direct.

14. p. 17: The word “localize” in line 356 usually implies spatial localization. Rather, it seems you mean decode or detect. This statement is also incorrect in that for two decades at least it has been shown that BOLD can decode event-related responses that are as brief as 16 ms. The key difference here is that they are among sequences of other interfering events spaced closely together. This point should be made more clear.

15. p. 17, line 390: grammatical error - “...because it allows to unravel.” This should be “allows the unravelling of”

16. p. 18, lines 395-424: At this point it would be helpful to speculate on why the falling slope of the HRF drives the sequentiality and contributes to re-ordering. Is it due in some way to the nature of the HRF? What does this imply about the HRF?

17. p. 19, line 438-440. The statement on the increasing sampling rate only partially increasing power is too vague. In fact, it’s known that increased sampling rate always increases power. The question is more along the lines of what is the relationship between sampling rate and power. I would suggest addressing this a bit more carefully as the reader is left with no real guidance on what sampling rate would be optimal. Do you think that the TR you use (1250 ms) is optimal? What TR would be optimal?

18. p. 27. Why was spatial smoothing used? Wouldn’t this reduce decoding power? Detailed decoding information is known to be eliminated by spatial smoothing.

Reviewer #3 (Remarks to the Author):

In this manuscript, Wittkuhn and Schuck examine with novel multivariate analyses methods if fast sequential events can be detected with fMRI. Due to the sluggishness of the BOLD signal, the non-invasive investigation in humans of fast events has been so far implemented with MEG and EEG. The authors had two main aims related to the investigation in fMRI of signals resulting from fast events (1) order detection and (2) element detection (to be able to determine if a sequence item is being reactivated).

In order to answer these questions, the authors trained probabilistic pattern classifiers for five objects that were presented multiple times in slow sequences five-items long, each with a different order permutation. The decoding probability for each item was assessed at these slow trials, and the authors calculated a fitted response model using averaged parameters for each object (class) decoded. Based on the response model mentioned, the authors were able to approximate expectations for signals during sequential events, by computing differences in probabilities between time-shifted events. The idea is that, even if the BOLD response is slow, the decoding probabilities for the different objects should be able to inform about sequence order. In the same task, there were sequence trials, consisting of permutations of the same five objects, but this time presented at different speeds. The authors tested the probabilistic classifiers on these sequence events in a series of clever analyses (figure 3), finding that the probabilities of the decoded objects reflected the real sequence order. These analyses combined decoding probabilities for each object within each TR, as well as across-TRs. The results nicely fit with the predicted model. Importantly, for earlier TRs not only the earliest objects in the sequence show a higher decoding probability, as well as the opposed pattern for later TRs, but also the time-course of the regression slopes between event position and probability reflect the correct sequence order. Next, the authors tackled the problem of element detection. For this, another condition had been introduced in the task design: fast sequences 9-

items long that consisted of repetitions of two of the five objects, one of the two objects, the first or the second appearing, was presented eight times in a row. The authors found that, despite interference from the repeated object, the other object that had been presented was still detected. Moreover, they found that the item at the end of the sequence was easier to decode. Finally, the authors simulated a lower signal to noise ratio to assess if sequence events can still be decoded in such circumstances, similar to those of events like hippocampal replay. The authors inserted in the rest state recordings fast or slow sequences and found that even in conditions of higher noise and the lowest number of sequence inserts, they could still detect fast and slow sequences.

Overall, I believe this manuscript was exceptionally well written and it was very clear. They have developed clever analyses that have allowed them to be able to detect fast sequence events in fMRI. All the analyses were thoughtful and in general were easy to follow. Overall, it is convincing and I would recommend it for publication.

Comments:

(1) The authors should make more clear how this manuscript fits with their previous paper (Schuck and Niv, 2019).

(2) On Line 124: "This analysis confirmed delayed and distinct increases in the estimated probability of the true stimulus class given the data, peaking at the fourth TR after stimulus onset, as expected (Fig. 2b)." In figure 2b, it is unclear why it was expected the peak on the fourth TR and would be informative to explain it more in the main manuscript.

(3) Regarding the repetition trials, if one reads first the main manuscript, then the methods and finally the supplementary methods, it is easy to get confused about the structure of the repetition trials. It is unclear in the main manuscript how were the sequences constructed. In figure 1c, it is said that "Repetition trials were always fast (32 ms ISI) and contained two visual images of which either the first or second was repeated eight times". The same structure is described on Line 237 'we investigated classification time courses in repetition trials, in which only two out of the five possible images were shown. Crucially, one image was repeated, while the other one was shown only once'. Both figure 1c, and comments in the main manuscript, make the reader understand that one of the categories was repeated only once and that happened either at the start or at the end of the nine items long repetitions sequence. There is a mention in line 260: "Additional conditions with intermediate levels of repetitions are reported in the SI (Fig. S1e)". I would suggest a bit more explanation regarding the different 'levels of repetitions' in this line of the main manuscript, as well as making explicit in the methods (line 574) that only two repetition patterns were presented in the main manuscript among all the ones described in the methods (line 574). It is not mentioned in the Methods and it becomes confusing. Only by reading Fig S1.e can one understand well.

(4) I understand that the 13 TR reported in many figures (e.g, fig 3a) includes all the time elapsed between the start of each sequence until the test starts, right?

(5) While this manuscript is a nice methodological paper, it might have potential questions about how the brain works, which would in fact shed light on the last claim of the authors that this method could might detect neural replay. To investigate if their method can be applied to investigate neural

mechanisms like hippocampal replay, the authors construct artificial fast events with different levels of signal-to-noise ratio, by embedding different numbers of sequence events into resting-state recordings. While this analysis was convincing to me, I suggest the authors perform a new analysis in their current data that could shed light onto this question: the authors could perform the same analyses, but training and testing the probabilistic classifiers on a hippocampus mask. After all, if this method should be able to detect replay after the stimulus has been presented, at least it should be able to detect hippocampal activity that responds to on-screen stimuli. Another question would be if there are learning effects. For instance, across event repetitions, stimuli will be learned and hence, the hippocampal activity might change. For instance, across repetitions, one would expect the hippocampus to anticipate the last items in a sequence.

(6) On Line 243, the authors state that “Finally, varying whether the second or first item is short allowed us to investigate if the ability to detect sequence elements is asymmetrical, and possibly favors the detection of late over early events.”. They write this later hypothesis but it is not explained why.

(7) In figure 4e, it is unclear what are “repetition 1” and “repetition 2”.

(8) Figure 5b and 5c of the main manuscript do not fit well. In figure 5c, I see higher absolute slopes for the slow condition, followed by the fast condition, and then the rest condition. But this is not what figure b shows. It is not well explained, neither in the text nor in the figure explanation below.

(9) On Line 473, “Auditory feedback was used to anatomically separate the expected neural activation patterns of visual stimuli and auditory feedback” is a bit unclear to me.

(10) On Line 743 of the main manuscript, the authors state that “In order to ensure that the classifier estimates were not biased by relative differences in class frequency in the training set, the weights associated with each class were adjusted inversely proportional to the class frequencies in each training fold.” I understand that, given that there were 5 classes to decode, the frequencies used to adjust were 1/5 for the class of interest, and 4/5 for the ‘other’ class, comprising any other classes?

(11) As I understand it, there were seven runs in this experiment. The probabilistic classifiers applied in each run were different from those for the other runs, right? Because you applied a leave-one-run-out cross-validation procedure. Line 799 of the main manuscript: “This procedure resulted in fold-specific maps of t-values that were used to select voxels from the left-out run of the cross-validation procedure. Note, that this approach avoids circularity (or so-called double-dipping) as the selective analysis (here, fitting of the GLMs to the training set) is based on data that is fully independent from the data that voxels are later selected from”.

Were there significant differences in decoding probability across-folds?

(12) typos: On Line 268, the word ‘the’ should be removed. On Line 345 of the main manuscript, the authors refer to fig 5f, when they are talking about figure 5h. On Line 348, they again mention fig 5f when they are referring to figure 5i.

**Response to reviews**

**Manuscript NCOMMS-20-12009A-Z**

*Faster than thought: Detecting sub-second activation*
*sequences with sequential fMRI pattern analysis*

Lennart Wittkuhn^{1,2*} and Nicolas W. Schuck^{1,2*}

¹Max Planck Research Group NeuroCode, Max Planck Institute for Human Development, Berlin, Germany

²Max Planck UCL Centre for Computational Psychiatry and Ageing Research, Berlin, Germany

*Correspondence to {schuck, wittkuhn}@mpib-berlin.mpg.de

Overview

We first reproduce each comment in blue. Our answers will follow immediately in black. Quotes
from the manuscript are indented. Changes applied to the manuscript are highlighted in yellow
with text removals struck through, both here and in the manuscript.

Answers to Reviewer 1

Remarks to the Author

*“In this manuscript, Wittkuhn and Schuck examined whether functional magnetic resonance*
*imaging (fMRI) can be used to measure subsecond sequences of neural activation patterns in*
*humans, akin to hippocampal replay. The problem the authors set out to tackle is critical in ad-*
*vancing the study of cognitive neuroscience in humans, as we are largely limited to non-invasive*
*measures of brain activity that force us to choose between spatial and temporal resolution. By*
*finding ways to circumvent this trade-off, we can begin to measure psychological and neural*
*phenomena that have previously been limited to study in patients or animal models.*

*Using a clever combination of experimental design, modelling and statistical analysis, Wittkuhn*
*& Schuck demonstrate that multivariate classifiers applied to fMRI data can be used to detect*
*the sequential activation of discrete images presented less than a second apart.*

*The main strengths of this paper are:*

*First, the authors present very thorough, convincing, and quantitative analysis;*

*Second, the results have broad impact, as they provide insight into (I) the temporal resolutions*
*at which we can non-invasively classify discrete visual events and their ordering in a sequence,*
*(II) how classification can be expected to work at different phases within a sequence of items,*
*allowing for interference, and (III) and how these classifiers can apply to more unconstrained*
*contexts, such as resting state, where the signal to noise ratio is considerably lower.*

*Although this paper presents very solid science, it does have some weaknesses which include:*
*insufficient examination of how classification changes as a function of the human participants*
*task state; somewhat confusing “forward” / “backward” jargon; some disconnects between the*
*stimulus properties and the physiological and practical realities of replay.*

*Overall, we are confident that this manuscript will make an important contribution to the field.*
*With that in mind, we offer the following points, which we hope can further strengthen this*
*contribution.”*

We are grateful to Reviewer 1 for their helpful and thorough review and for being so posi-
tive about our manuscript. Indeed, your question Q6 prompted an analysis that significantly
strengthens our paper and we would like to thank you for your input.

Major comments

**Q1: Discrepancy between the duration of stimulus sequences and the motivating**
**neural phenomena:** *In the introduction, the authors refer to neural phenomena that occur on*
*the subsecond scale (e.g., hippocampal replay: 200-300ms, orbitofrontal value representations:*
*< 100ms) and use these as a rationale to attempt to decode rapid sequences of stimulus rep-*
*resentations from fMRI. However, there is little discussion about the discrepancy between the*
*duration of these neural events and the duration of the experimental sequences in this paradigm.*
*For example, despite the ISI being as low as 32ms in the Sequence of Repetition trials, each*
*stimulus was onscreen for 100ms, amounting to sequences that were at least 628 ms in length.*
*To be clear, it makes sense that the authors chose to have each stimulus onscreen for 100ms, as*
*participants may have found the task difficult to perform at shorter presentation times. How-*
*ever, some discussion is necessary about the marked discrepancy between the duration of the*
*experimental sequences and the neural phenomena that motivated this study. This is especially*
*pertinent as the discussion explicitly mentions applying the classification approach from this*
*manuscript to the measurement of these subsecond neural phenomena (see paragraph 2 of the*
*Discussion). Related to this same point: labeling the “speed” of the sequences in Figure 2E*
*(and throughout Figure 3) as 32ms, 64ms, etc., is somewhat inaccurate, because the items in*
*the sequence are not actually arriving at 32ms/item, 64ms/item, etc.*

**A1:** We agree that the relationship between the temporal scale of our method and the neural
phenomena that we propose to study with it should be discussed in more detail.

In general, we also agree that the duration of our fastest sequences (628 ms) is a bit longer than
the speed of average replay events in the hippocampus. Kaefer et al. [1, their Fig. 3E], for
instance, report a median duration of replay events of 330 ms, in line with other studies [e.g.,
2, their Fig. 1e]. Notably, however, replay events can be significantly longer. In the mentioned
study by Kaefer et al. [1] it appears that approximately 20% of hippocampus events were longer
than 500 ms and Davidson et al. [3] reported events of up to 1000 ms. The median duration of
events in medial prefrontal cortex (PFC) reported by Kaefer et al. [1] was 740 ms.

A separate question is whether our method would be sensitive to events faster than our fastest
condition. Although this remains arguably an empirical question, we believe that our results
give reason for optimism. Specifically, while the effect sizes dramatically drop when vastly
different speed conditions are considered (2048 ms vs. 512 ms and 512 ms vs. 128 ms), the
three faster conditions of 128 vs. 64 vs. 32 ms do not seem to be characterized by a marked
reduction in effect sizes (see Fig. 3c in the main manuscript). It therefore does seem likely
to us that the reported effects would generalize to even faster events without a catastrophic
reduction in the sensitivity of our analysis.

Finally, as the reviewer points out, the sequence duration was to a large extent caused by
stimulus duration. Evidence from previous work using electroencephalography (EEG) suggests
that the neural response to successive visual stimuli is more strongly influenced by the ISI than

the stimulus duration [4, 5]. Translating these results to the current study, we speculate that
our methods may also work in cases with shorter pattern activations and thereby overall shorter
sequences.

We now discuss these issues in the Discussion on pages 22–23:

*“The fastest sequences studied in our experiments lasted 628 ms and were there-*
*fore longer than the average hippocampal replay event of about 300 ms [e.g., 17].*
*Yet, several factors support the idea that our method is still relevant for the study*
*of replay. First, previous studies have shown that a significant proportion of replay*
*events indeed lasts much longer. Davidson et al. [7] report sequence lengths of up*
*to 1000 ms and the data published by Kaefer et al. [17] indicates that about 20%*
*of events in the hippocampus are longer than 500 ms. In addition, the median du-*
*ration of replay events in medial PFC reported in Kaefer et al. [17] was 740 ms.*
*This indicates that a significant proportion of replay events will be covered by our*
*method. Second, our ISI was as fast as 32 ms, which corresponds to the time lag*
*between activations reported in several magnetoencephalography (MEG) studies [46]*
*and therefore might capture the important aspect of temporal separation between ac-*
*tivation patterns well. Third, while effect sizes showed a pronounced decrease when*
*comparing the slower conditions (2048 ms: 3.04; 512 ms: 1.36; 128 ms: 0.78, for*
*the backwards effect of regression slopes, Fig. 3c), accelerating sequence speeds be-*
*yond 128 ms seemed not to be associated with a comparable decrease in effect sizes*
*(64 ms: 0.66, 32 ms: 1.02). This indicates that the sensitivity of our methods for*
*even faster events sequences might not be catastrophically diminished. Fourth, the*
*sequence duration of 628 ms was to a large extent due to the stimulus duration of*
*100 ms. Evidence from previous work using EEG suggests that the neural response*
*to successive visual stimuli is more strongly influenced by the ISI than the stimulus*
*duration [73, 74]. Hence, we speculate that our methods may also work in cases with*
*shorter pattern activations and thereby overall shorter sequences.”*

We have also adapted Fig. 1d, in which we now include a clear indication that the sequence
speed conditions refer to the ISI, mention the extra 100 ms for stimulus presentation and
prominently illustrate the total sequence length of the shorter conditions. We hope that this
will reduce any confusion. We also state this very explicitly in the manuscript:

*“Sequence speed was manipulated by leaving either 32, 64, 128, 512 or 2048 ms*
*between pictures, while images were always presented briefly (100 ms per image,*
*total sequence duration 0.628–8.692 s). Note, that we refer to the ISI as “sequence*
*speed”, see Fig. 1d.”*

For convenience, we also reproduce the updated Fig. 1d below (see Fig. R1):

Figure R1: Updated Fig. 1d with an illustration of sequence speed conditions. Illustration of the three fastest sequence speed conditions of 32, 64, and 128 milliseconds (ms) inter-stimulus interval (ISI) between images. Mean behavioral accuracy (in %; y axis) in upside-down slow trials.

**Q2: *Classifying the cued vs. uncued stimulus category:*** *In the Sequence and*
*Repetition trials of this experiment, participants were cued to look for a certain stimulus in*
*each trial (e.g., look for and remember the serial position of the shoe in a Sequence trial).*
*However, the classification analyses on Sequence and Repetition trials were always tested on*
*all volumes of a given trial, and averaged across trials irrespective of the category being cued.*
*How does the classifier perform on the cued item's position relative to positions of the uncued*
*items? Does this change with different ISIs in the Sequence trials? Furthermore, attending*
*to the stimulus likely involves holding the image online (in "working memory", perhaps) for*
*the duration of the trial. Having the cued category in working memory could potentially boost*
*classifier evidence for the cued stimulus category for the whole trial, or up until the cued stimulus*
*has been presented. One potential way to control for the cued item, beyond counterbalancing*
*its position, would be to re-run the classifier while excluding the cued-category stimulus on*
*each trial. This way, classification cannot benefit from the explicit allocation of attention to a*
*specific stimulus category per trial. More generally, please indicate (i) whether the cued category*
*is decodable from the pre-stimulus period; and (ii) whether the overall effects shown in Figure*
*3 are different for trials when the last image in the sequence is cued or uncued. In any case,*
*the authors should clarify the contributions (or lack thereof) of explicit cueing to classification*
*accuracy.*

**A2:** We thank Reviewer 1 for raising these questions regarding the influence of the target
cue on decoding performance in sequence and repetition trials. The analyses of sequence trials
were averaged across all trials to ensure sufficient power given 15 trials per sequence speed / ISI
condition (75 trials in total per participant). As suggested by the reviewer, we split the trials by
the target cue's position and analyze the corresponding classification time courses separately.
Please note, that the cued target category appeared more often at the later compared to earlier
serial positions. Hence, trials were binned for target positions 1–3 and compared to trials with
target position 4 and 5. This ensured sufficient power and balanced the number of trials in each
bin (5 trials per target cue position for each participant and speed condition). As can be seen
in Fig. R2, no influence of target cue position on the sequentiality metrics became apparent.
In the 32 ms condition, for instance, sequentiality in the backward period was significant for
all tested cue positions (all p 's < .05, all d 's > 0.41). The remarkable similarity of these
results that included only a third of the data in each bin of target cue trials confirms the
robustness of our findings. Next, we followed the reviewer's helpful suggestion and investigated
if "*the cued category is decodable from the pre-stimulus period*". To this end, we applied the
classifiers to five TRs from cue onset (spanning the cue period of 1000 ms, the following blank
screen of 3850 ms and the fixation dot of 300 ms shortly before the stimulus sequence onset).
As for slow trials (see Fig. 2b in the main text), we then calculated the mean probabilistic
classification evidence for all five stimulus classes depending on the cued target category (see
Fig. R3). These results indicated that the cued target category could not be decoded from
the pre-stimulus period. Regarding the repetition trials, we would like to note that these trials
were already analyzed depending on the serial position of the cued stimulus category. The

main purpose of the repetition trials was to investigate classification probabilities depending
on the number of repetitions of only two image categories. In the main manuscript, we only
report the two most extreme conditions, that is, when the first / second item is presented only
once and followed / preceded by eight repetitions of the other item, respectively. Please see
the “Additional results for repetition trials” in the supplementary information (SI) for details
on all intermediate task conditions. Together, these additional analyses suggest that neither
the cued target category’s serial position within the stimulus sequence nor the target cue itself
influenced the sequentiality metric on sequence and repetition trials.

Figure R2: Effects of the cued target's serial position on classification time courses. (a) Time courses (repetition times (TRs) from sequence onset; x-axis) of classifier probabilities (%) (y-axis) per event (colors), sequence speed (vertical panels) and serial position of the cued event (horizontal panels). (b) Time courses (TRs from sequence onset; x-axis) of mean regression slopes between event position and probability (y-axis) for each speed (colors) and serial position of the cued event (panels). Positive / negative values indicate forward / backward sequentiality. (c) Mean slope coefficients for each speed (colors / vertical panels), period (forward vs. backward; x-axis), and serial position of the cued event (horizontal panels). Stars indicate significant differences from baseline. Errorbars / shaded areas represent ± 1 standard error of the mean (SEM). Effect sizes are indicated by Cohen's d . Stars indicate $p < .05$, false discovery rate (FDR)-corrected. 1 TR = 1.25 seconds (s).

Figure R3: Decoding the cued category from the pre-stimulus period. Time courses (in TRs from cue onset; x-axis) of probabilistic classification evidence (in %; y-axis) for all five cued target categories (panels).

**Q3: Use of terminology:** *The section of the results looking at the early and late halves of the*
*forward and backwards periods was quite difficult to parse (“This sequential progression . . .”, ln*
*216, p 9). Not only are the forward and backwards periods now divided into early and late phases,*
*but the reader also has to keep the relationship between step direction (i.e., positive or negative)*
*and where in the 5-element sequence these steps refer to. This becomes even more complicated as*
*the reader encounters terms such as “fwd condition”, “bwd condition”, “forward-interference”,*
*“backward-interference”, “forward transitions”, “backward transitions”, “inwards transitions”,*
*“outwards transitions” and finally “outside transitions”. Also, the use of asymmetric language*
*in describing the results further complicates reading (e.g., “. . . forward directed in the forward*
*as compared to backward period” as compared to “. . . negative directed in the forward period*
*and vice versa in the backward period . . .”; see ln 227-232 on p 9). Here are some potential*
*solutions that the authors should consider to help orient the reader through these analyses: Add*
*the “early” and “late” phase labels to Figure 2d and refer to this visual depiction in the results*
*section where these labels are introduced. Add arrows to Figure 3h that indicate negative steps*
*refer to earlier in the sequence, while positive steps are later in the sequence. This could look just*
*like the gray arrows on Figure 3f. Alternatively, might it be clearer if Figure 3h was reorganized*
*to be plotted in time? Each speed could be a column/facet, where the X-axis for each facet has*
*the following results in the following order: fwd-early, fwd-late, bwd-early, bwd-late. This way,*
*the periods/phases are presented as they unfold in time and the reader doesn’t have to do multi-*
*step mental rotation to map Figure 3h back to any of the plots where the X-axis is time in TRs.*
*Try, whenever possible, to use symmetric language whenever possible (e.g., forward directed*
*and backwards directed as opposed to forward directed and negative directed). In relation to*
*the transitions, is it possible to move the descriptions from the methods (ln 995-999, p 34-35)*
*to the results section? The description in the results section is too brief to make sense of the*
*various transition labels, and consequently limit how easily one can interpret, Figures 4d and*
*e. In relation to Figure 4e, could the diagonal simply be black and labelled “repetition”, rather*
*than being divided into 3 additional transition types? The colours are quite visually similar to*
*one another (e.g., see inward, outward, outside, repetition out) and consolidating those three*
*transition types into one might help reduce the figures complexity a (meaningful) bit.*

**A3:** We thank Reviewer 1 for these helpful suggestions and have now substantially revised
the relevant sections in our manuscript. We have removed the use of asymmetric language in
describing the results of the early versus late phases of sequence trials:

*“This sequential progression through the involved sequence elements had implications*
*for transitions between consecutively decoded events: ~~Initially, when early elements~~*
*~~begin to dominate the signal in the first half of the forward period (henceforth early),~~*
*~~the position of decoded sequence items decreased relative to baseline. During the first~~*
*~~half of the backward period, however, the decoded serial positions increased, reflect-~~*
*~~ing the ongoing progression through all sequence elements from first to last. The~~*
*~~reverse was true during the second half of both periods (henceforth late): positions~~*

*began to increase in the forward period, but during the second half of the backward*
*period, the decoded positions were about to return back to baseline from the last de-*
*coded item, thus decreasing again. the transitions will be a direct function of the*
*slope of the average decoded position shown in Fig. 3f. When the slope is negative,*
*the steps between successive sequence items are backward and reflect the transition*
*from a later position to an earlier position. When the slope is positive, the steps are*
*forward, reflecting a progression from an earlier event position to a later event po-*
*sition. To verify this effect, we computed This can be verified by computing the step*
*sizes between consecutively decoded serial events as in [53]. For example, observing a*
*2→4 transition of decoded events in consecutive TRs would correspond to a forward*
*step of size +2, while a 3→2 transition would reflect a backward step of size -1.*
*As can be seen from Fig. 3f, both the early and late phase of the response included*
*periods with a negative and a positive slope, in line with our predictions (formally,*
*the prediction can be obtained by taking the derivative with respect to time of Equa-*
*tion 6 (see Methods), i.e., the function shown in Fig. 2e). We therefore considered*
*the periods with a positive and negative position slope separately for the early and*
*late phase. As expected, the early transitions were mainly forward during the period*
*of a positive slope as compared to the negative slope periods for speed conditions of*
*512 and 2048 ms ($p \leq .005$, Fig. 3h). Average step sizes of late transitions, in*
*contrast, were negative directed in the forward period and vice versa in the backward*
*period In line with the above mentioned predictions, the step sizes of early transitions*
*were significantly more forward directed in the forward as compared to the backward*
*period Similarly, the late transitions were also forward and backward during the pos-*
*itive and negative slope periods, respectively, and differed in all speed conditions (p*
*$\leq .05$, Fig. 3h), except the 64 ms condition ($p = .19$). This analysis suggests that*
*transitions between decoded items reflect the ordered gradual progression through all*
*from early to late and then from late to early sequence events, even when events*
*were separated only by tens of milliseconds. ”*

We have also changed the wording in repetition trials:

“We next turned to our second main question, asking whether we can detect which
patterns were part of a fast sequence and which were not. One important reason
why detecting which patterns were activated during sequence events might be more
difficult than in a standard setting is that co-activation of multiple patterns close in
time could lead to interference. We therefore investigate such interference in detail
below. We analyzed classification time courses in repetition trials, in which only two
out of the five possible images were shown. One of the two images was repeated,
while the other one was shown only once. This setup allowed us to study to what
extent another activation (the repeated image) can interfere with the detection of
a brief activation pattern of interest (the image shown only once). The repeating

*image was shown eight times, which created maximally adverse effects for the de-*
*tection of the single image. To ask if detection of brief activations is differently*
*affected by events occurring before versus after the single event, we varied whether*
*the second or first item is short. We pose this question because the backward effects*
*were consistently larger than forward effects in our sequentiality analyses reported*
*above (see Fig. 3c). This implies that one briefly presented item at the end of a*
*sequence will be easier to detect than a briefly presented item at the beginning of a*
*sequence, even though both were equally close in time to another strong activation*
*signal. In consequence, short items followed by a longer item during repetition trials*
*should be more difficult to detect than those preceded by a long item. To test this*
*idea, we considered the above described two order conditions. We will term the case*
*in which the first image was shown briefly once and followed immediately by eight*
*repetitions of a second image the forward interference condition, because the forward*
*phase of the sequential responses suffers from interference. Correspondingly, trials*
*in which the first image was repeated eight times and the second image was shown*
*once will be termed the backward interference condition. In all cases, images were*
*separated by only 32 ms. Comparing the forward and backward conditions allowed*
*us to assess the asymmetries, which had become apparent in the results presented*
*above (Fig. 3). As before, we applied the classifiers trained on slow trials to the data*
*acquired in repetition trials to obtain and obtained the estimated probability of every*
*class given the data for each TR (Figs. 4a, S7). The expected relevant time period*
*was determined to be from TRs 2 to 7 and used in all analyses (see rectangular areas*
*in Fig. 4a)."*

Finally, we have updated the Figures as suggested. In detail, regarding the individual sugges-
tions:

*"[...] Add the "early" and "late" phase labels to Figure 2d and refer to this visual*
*depiction in the results section where these labels are introduced."*

Figure 2d now shows labels for the early and late phases of the forward and backward period,
respectively, and we refer to them in the Results section, when the labels are introduced, as
suggested.

*"[...] Add arrows to Figure 3h that indicate negative steps refer to earlier in the*
*sequence, while positive steps are later in the sequence. This could look just like*
*the gray arrows on Figure 3f. Alternatively, might it be clearer if Figure 3h was*
*reorganized to be plotted in time? Each speed could be a column/facet, where the*
*X-axis for each facet has the following results in the following order: fwd-early, fwd-*
*late, bwd-early, bwd-late. This way, the periods/phases are presented as they unfold*
*in time and the reader doesn't have to do multi-step mental rotation to map Figure*
*3h back to any of the plots where the X-axis is time in TRs."*

We thank the reviewer for this detailed suggestion. Unfortunately, it is not possible to follow
the reviewer’s suggestion in a way that would not misrepresent the data. First, it is not correct
that negative and positive steps always refer to earlier and later in the sequence, respectively.
Rather, step sizes are negative for the early phase of the forward and late phase of the backward
period, while step sizes are positive for the late phase of the forward period and the early phase
of the backward period. Thus, adding arrows to Figure 3h as in Figure 3f would not be correct.
It is also not possible to implement the alternative suggestion of the reviewer. Figure 3h shows
a statistical comparison of the early and late phases of the forward and backward periods,
respectively. Plotting these data in time would show bars next to each other that are not
compared thus obscuring the statistical test that was conducted. We hope that the reviewer
understands that we choose to leave Figure 3h as it is and hope that the changes applied to
Figure 2d support the understanding of Figure 3h as well.

*“[...] In relation to Figure 4e, could the diagonal simply be black and labelled “rep-*
*etition”, rather than being divided into 3 additional transition types? The colors are*
*quite visually similar to one another (e.g., see inward, outward, outside, repetition*
*out) and consolidating those three transition types into one might help reduce the*
*figures complexity a (meaningful) bit.”*

We have now combined all repetitions into one transition type label, called “repetition” and
have changed the color to white. We have refrained from coloring the “repetitions” in black as
we had the impression that the high contrast would draw too much attention to a transition
type that is not the main focus of the analyses. We hope that the reviewer approves of these
changes.

**Q4: A gradual progression through all sequence events?:** *On page 9 (ln 232-234)*
*the authors state:*

*“This analysis suggests that transitions between decoded items reflect the gradual*
*progression through all sequence events ...”*

*Is this true? The step size analysis indicates a difference in the direction of the predicted element*
*relative to the midpoint of the sequence, which changes over time. However, considering these*
*effects are strongly affected by the first and last element of the sequence, especially for trials*
*with short ISIs (e.g., see Figure 3a) it seems to be a bit of an overstatement to imply strong*
*evidence for a gradual progression through all elements in a sequence.*

**A4:** We thank Reviewer 1 for this comment. We would like to clarify an apparent confusion
about the analysis of serial position and the analysis of step-sizes. The reviewer’s description of
the analysis that “indicates a difference in the direction of the predicted element relative to the
midpoint of the sequence, which changes over time”, nicely summarizes the analysis of average
serial position. This is different from the analysis of step-sizes however. The analysis of serial
position simply calculates the mean serial position at every timepoint. The analysis of step-sizes

investigates the directionality of steps *between consecutive TRs*. As the reviewer points out,
this analysis does not explicitly analyze transitions between specific sequence elements (e.g.,
in their true order) but agnostically calculates the step-sizes between any two consecutively
decoded events. The reviewer is right that this could also reflect transitions from early to late
sequence events more generally, rather than short 1-step transitions from element to element
through the entire sequence. Thus, we changed the relevant statement in the main manuscript
accordingly:

*“This analysis suggests that transitions between decoded items reflect the **ordered***
***gradual progression through all from early to late and then from late to early** sequence*
*events, even when events were separated only by tens of milliseconds.”*

**Q5: Implications for replay:** *As mentioned in our first point, the authors frame this paper*
*in relation to fast neural sequences, such as hippocampal replay. However, it wasn’t always easy*
*to follow the implications of various insights from the authors model and experimental results*
*on attempts to measure replay events using fMRI. For example, the authors state that “...*
*forward replay can look like backward replay and vice versa” (ln 397-398, p 18). We agree that*
*this is a critical insight - but further discussion of this point is warranted. Are we unable to*
*dissociate between these two types of replay due to the properties of the hemodynamic response*
*function we rely on in fMRI? Rather than only leaving this crucial point open to the speculation*
*of the reader, it would be helpful if the authors provided a sentence or two that discuss this point*
*more concretely. Furthermore, it’s not obvious how the interference results, from the repetition*
*trials, relate to hippocampal replay events. We understand that there is a practical utility to*
*these results, because they inform us as to the kinds of interference that arise in decoding*
*one stimulus that is preceded (or followed) closely by another ... but it is difficult to follow the*
*practical relevance of all of this work for sequence replay (which is how the manuscript is framed*
*as a whole).*

**A5:** We thank Reviewer 1 for this comment. We agree that the implications of our findings
for the study of replay should not be left solely to the interpretation of the reader but rather be
described more precisely in the manuscript. Regarding the implications for the finding forward
versus backward replay events, we added the following sentence to the manuscript:

*“This represents a crucial insight, given the different functional roles assigned to*
*forward and backward replay [...]. **We note that future research should be careful***
***when interpreting directionality, as the relationship between decoded and true direc-***
***tionality is not straightforward. One approach in this context could be to investigate***
***the order of sequence direction itself. If items appear to be ordered first in direc-***
***tion A, and a few TRs later in direction B, then direction A seems to be the true***
***one. Probabilistic classifiers might prove particularly useful for such analyses as they***
***make it possible to characterize sequential ordering within a single measurement.”***

Regarding the implications of our findings for the interference effects demonstrated in the data

from the repetition trials, we added the following additional sentences to the manuscript:

*‘Third, we have shown that the interference of activation patterns of fast sequen-*
*tial neural events is stronger for early compared to late events. Importantly, early*
*events remained detectable despite this interference, demonstrating that our method*
*can detect the elements of a replay event with fMRI. The prominence of the last se-*
*quence item implies that any apparent over-occurrence of one particular item might*
*imply that this item was a frequent start or end point of replayed trajectories. Past*
*research has shown that task aspects, such as goals, heavily influence which items*
*sequences start or end with [76].’*

**Q6: Pre-task vs. post-task rest:** *From my understanding, it appears as if the resting state*
*classification analyses were exclusively conducted on the 5-minutes of resting state that were*
*collected before task performance. Why was post-task rest not analyzed? It would be interesting*
*if the power spectra for post-task rest was different from that of pre-task rest. Wouldn't the*
*post-task resting state potentially contain task-related replay events and provide a naturalistic*
*counterpart to the objective “sequence insertion” approach used on the pre-task rest?*

**A6:** We had initially not analyzed post-task resting-state data because our task had no
particular mnemonic component or sequential structure that we expected to elicit replay in
post-task rest. But we are glad the reviewer brought this up and investigated the post-task
resting-state data as suggested. As you will see, the results are stunning and significantly
enhance our manuscript. In fact, we found exactly what the reviewer suggested: evidence
for spontaneous reactivation of sequences of the shown stimuli in the post-task rest condition,
relative to the pre-task rest condition. The results are shown in Fig. R4 (and Fig. 6 in the
main manuscript). We believe that these additional results are exciting and, as the reviewer
suggested, “provide a naturalistic counterpart to the objective “sequence insertion” approach
used on the pre-task rest”. We now include these findings in an additional paragraph in the
Results section, as well as in Figure 6 (reproduced for convenience in in Fig. R4):

[revised manuscript text omitted]

Finally, we now also mention this additional finding in the Abstract:

*“Moreover, the frequency spectrum of our sequentiality metric distinguished between*
 *sub- versus supra-second sequence speeds. Lastly, we applied our method to post-task*
 *rest data and found evidence for fast replay of task-related stimuli. This indicates*
 *that even simple tasks without memory requirements might elicit sequential replay*
 *in sensory brain areas, and shows that our method can be used to detect such spon-*
 *taneously occurring replay. Our method paves the way for novel investigations of*
 *fast computations in the human brain, like hippocampal replay.”*

Figure R4: Detecting fast task-related reactivations in post-task resting-state data. (a) Normalized frequency spectra of regression slopes in pre- and post-task resting-state data. Annotations indicate predicted frequencies based on Eqn. 5. Shaded areas represent ± 1 SEM. (b) Relative power (difference between pre- and post-task rest) of normalized frequency spectra shown in (a). (c) Mean power at predicted fast frequency (0.17) in pre- and post-task resting-state data (x-axis) for less and more frequent stimulus sequences. Each dot corresponds to averaged data from one participant. Errorbars represent ± 1 SEM.

**Q7: Asymmetries between forward and backward interference:** *Are there any*
 *electrophysiological studies that support the asymmetric interference findings reported in this*
 *paper? There is very little reference in the paper to whether any of the findings (e.g., early*
 *vs late item contributions to decoding) are consistent with the interference effects observed in*
 *electrophysiological studies of population firing rates.*

**A7:** This is an interesting question. We have assumed that it is an issue that arises only
 in fMRI. More specifically, we believe that the sustained BOLD signal that follows even brief
 events is the source of measurement interference. The last item in a sequence has the particular
 advantage that it is not followed by another item. It may also be the case that the slightly

asymmetric HRF, which features a longer tail than rising slope, provides benefits. More spec-
ulatively, it is known that in some cases postsynaptic neurons can produce neurotransmitters
that diffuse back to presynaptic neurons [e.g., 6]. Such retrograde pathways can be inhibitory.
The question whether this would have any relations to replay is difficult as we are studying
stimulus evoked responses. We are not aware of any electrophysiological findings from the re-
play literature that would indicate such an asymmetry. If this were the case it could indeed
hint at either shared technical aspects of decoding or be a sign of the mentioned retrograde
inhibition that occurred in a replay setting but also in the stimulus evoked responses studies
here. We now write in the discussion:

*“The origins of this asymmetry are not entirely clear. It seems possible that they*
*reflect the benefits of the last item not being followed by another activation that could*
*impede its detection. A relation to the asymmetric shape of the HRF, to changing*
*HRF variability with time and even to inhibitory retrograde neurotransmitters [e.g.,*
*75] cannot be ruled out.”*

Minor comments

**Q8: Testing set:** *Was the model only tested on accurate (in terms of a participant’s*
*behavioural response) Sequence and Repetition trials? It’s clear that the model was only trained*
*on accurate trials, but I might have missed if this was explicitly stated for the trials the classifier*
*was tested on.*

**A8:** We thank the reviewer for asking for this information, which was indeed not explicitly
stated in the manuscript. The classifiers were applied to both accurate and inaccurate sequence
trials. We now added the following sentence to the Methods section:

*“When the classifiers were applied to sequence and repetition trials, data from both*
*accurate and inaccurate trials were used to allow for an equal number of test trials*
*across participants and maximize statistical power within the current study design.*
*As shown in Fig. 1e-f, behavioral performance on sequence and repetition trials was*
*high and significantly above chance.”*

**Q9: Repetition trials** *I am having difficulties wrapping my head around the repetition*
*trials. In the Methods section, the authors state: A: “All sequences had a fixed length of nine*
*stimuli in total.” (ln 574, p 23) B: “For each of these eight sequences we systematically varied*
*the point of the switch to the second stimulus category from serial position 2 to 9. ” (ln 575-*
*577, p 23). C: “Specifically, sequences with forward interference entailed a brief presentation*
*of a single image that was followed by eight repetitions of a second image; whereas backward*
*interference was characterized by a condition where eight image repetitions were followed by a*
*single briefly presented item. ” (ln 968-970, p 34) How can all three of these points be true? If*
*I understand the procedure correctly, (A) and (C) preclude the ability to have the switch between*
*stimulus categories occur at any serial positions other than 1 or 9, right? Is it the case that*

*the analysis in (C) only involves a small subset of the trials that are described by (A) and (B)?*

**A9:** We thank the reviewer for this comment, which was also raised by Reviewer 3 (see Q3
 of Reviewer 3) and apologize for the confusion about the design of the repetition trials. The
 reviewer is correct that the analysis of repetition trials reported in the main manuscript only
 involved a subset of all repetition trials that were shown to the participant. We focused on the
 two most extreme cases of repetition trials with a first event presented once followed by eight
 repetitions of a second event, or eight repetitions of the first event followed by a second event
 presented once. As stated in the manuscript, “additional conditions with intermediate levels
 of repetitions are reported in the SI”. In order to prevent confusion of the reader, we added
 the following sentences at the end of the first paragraph about the results of the repetition
 trials

*“Note, that our analyses focused on the two extreme cases of repetition trials with*
 *one versus eight repetitions of the first versus second item (or vice versa) while the*
 *experiment also included repetition trials with intermediate levels of repetitions (see*
 *SI). Other repetition trials included cases in which the second item began to appear*
 *at each possible position from 2 to 9. The other repetition trials could therefore in-*
 *clude, for instance, three repetitions of the first and six repetitions of the second*
 *image, or four repetitions of the first and five repetitions of the second item, etc.*
 *The results reported in the SI indicate that effects in these trials smoothly transition*
 *between the extremes shown in the main manuscript.”*

Further, we added the following sentence to the beginning of the Methods section describing
 the analysis of repetition trials:

*“Repetition trials included varying repetitions of two images in a sequence of nine*
 *items in total. All analyses reported in the Results section focused on the two most*
 *extreme cases, (1) the first image shown once followed by eight repetitions of the*
 *second image, and (2) eight repetitions of the first image followed by the second*
 *image shown once. Analyses of all intermediate levels of repetitions are reported in*
 *the SI.”*

**Q10: Page 4 (Line 84):** “... quantifies to what extent fMRI allows to detect which
 elements were ...” Replace with “... allows one to detect ...” or “... affords detection of the
 elements that ...”

**A10:** We thank the reviewer for this proposed change, which, combined with a similar
 suggestion by Reviewer 2 (see Q7 of Reviewer 2 below) resulted in the following update of the
 sentence:

*“The second effect, element detection, quantifies to what extent fMRI allows to de-*
 *tect which elements detection of elements that were part of a sequence and which*

*that were not.* ”

**Q11:** *Page 4 (Line 97): “with up to only 32 ms between stimuli.” Consider changing to:*
*“As little as 32 ms between stimuli”. Also, the wording here is somewhat misleading as the*
*inter-onset interval is never as low as 32 milliseconds; it is only the offset-to-onset time that is*
*as low as this.*

**A11:** We have changed the sentence as suggested and added information about the stimulus
duration:

*“Importantly, image presentation rate was greatly increased in sequence and repe-*
*tition trials, with ~~up to only~~ as little as 32 ms between stimuli and a presentation*
*time of 100 ms per stimulus.”*

**Q12:** *Page 4 (Line 122): “leave-one-run-out” At this point in the manuscript, the reader has*
*not yet been told what a “run” is.*

**A12:** We now include the information about task runs in the preceding paragraph on page
5:

*“The analyses included $N = 36$ human participants who underwent two fMRI ses-*
*sions with four task runs each, i.e. eight runs in total”*

Further, we add a short explanation of leave-one-run-out classification in the parentheses fol-
lowing the sentence quoted by the reviewer:

*“Cross-validated (leave-one-run-out) classification accuracy was on average 87.09%*
*($SD = 3.50\%$; $p < .001$, compared to chance; $d = 19.16$; Fig. 2a; eight folds in*
*total, as the experiment consisted of two sessions with four runs each.).”*

**Q13:** *Page 5 (Line 136): “we approximated expectations for signals during sequential*
*events” Can this be made more concrete / intelligible?*

**A13:** We thank Reviewer 1 for this remark and changed the relevant sentence as suggested:

*“Based on this fit to single events, we ~~approximated~~ derived expectations for signals*
*probabilistic time courses during sequential events.”*

**Q14:** *Page 7 (Line 149): “proportional to the ordering of events during the sequential*
*process” Can this be: “proportional to the stimulus order”?*

**A14:** We thank Reviewer 1 for this suggestion. The prediction that activation strengths will
be proportional to the *event order* during the sequential process is a hypothesis that we believe
extends beyond the current experiment. Emphasizing the broader application of our method
to any sequential process, not just sequences of stimuli as used in the current experiment, we

generally refer to *event* sequences. To clarify the relevant sentence further, we changed it in
 the manuscript as stated below:

“[...] proportional to the ~~ordering of events~~ **true event order** during the sequential
 process”

**Q15: Page 8 (Line 173):** “whether sequence order detection was evident in ...” Can this
 be: “Whether sequence order was detectable from ...”?

**A15:** We thank Reviewer 1 for this suggestion. The relevant sentence in the manuscript was
 changed accordingly:

“We investigated whether sequence order ~~detection was evident in~~ **was detectable**
 **from** the relative pattern activation strength within a single measurement.”

**Q16: Page 12 (Line 283):** No closing brace is required after “results”.

**A16:** Thanks, the closing brace was removed as suggested:

“Repeating all analyses using proportions of decoded classes (the class with the max-
 imum probability was considered decoded at every TR), or considering all repetition
 trial conditions, also revealed qualitatively similar results.”

**Q17: Page 12 (Line 297):** “. . . and (3) sequence item detection leads to within sequence
 pattern transitions.” The language here is somewhat confusing. “Sequence item detection” is
 something that is done by a neuroscientist who is analyzing experimental data . . . it is not a
 neural process. How can this item detection lead to pattern transitions, which are presumably
 a neural process? There is something confusing about the wording here.

**A17:** We thank the reviewer for raising this issue and agree that the wording might be
 confusing here. We simply meant that the detection of items that were part of the sequence
 makes it possible for us to detect transitions between decoded items that are part of the
 sequence. We now revised the sentence as follows:

“(3) ~~sequence item detection leads to within sequence pattern transitions~~ **the de-**
 **tection of sequence items made it possible to observe within-sequence transitions**
 **between decoded items.**”

**Q18: Page 13 (Figure 4b,c):** Couldn't these plots be simplified by replacing the bar with
 a single (black) dot with error bars that is placed on top of the points on the left? It's not clear
 what the bar plot buys you in this situation and simplifying the figure would be helpful.

**A18:** We agree with the reviewer that the visual complexity of Figure 4b and 4c could be
 reduced by removing the bar plots. We have now replaced the bar plots with diamond-shaped
 dots (including errorbars) that are visually distinct from the small circles showing averaged
 data of individual participants (see updated Figure 4 in the main manuscript). However, we

chose to still show the diamond-shapes next to the individual participants' data rather than on
 top of it, to avoid potential confusion and provide a clear indication of the sample mean (next
 to the sample median, plotted on top of the “raincloud” plot on the right).

**Q19: Page 14:**

*“To account for possible SNR reductions, the inserted probability time courses were*
 *multiplied by a factor kappa of 4/5 , 1/2 , 1/4 , 1/8 or 0 and added to the probability*
 *time courses of the inversely scaled (1 - kappa) resting state data. Effectively, this*
 *led to a step-wise reduction of the inserted sequence signal from 80% to 0%, relative*
 *to the SNR obtained in the experimental conditions reported above.”*

*My understanding is that SNR is typically expressed as a ratio of signal-variance to noise-*
 *variance. If a raw signal is multiplied by a factor kappa, then usually this would lead to a*
 *change in signal of kappa-squared (rather than kappa)? If that is right then the kappa=0.8*
 *setting should produce a change in SNR to 64% of its original value. It is possible I have the*
 *wrong idea, and this is a small point, but it could help to be clearer what is meant here by SNR.*

**A19:** We thank the reviewer for this comment, which was made in a similar way by Reviewer
 2 (see Q13 by Reviewer 2 below). We apologize that it has not become clear what we meant
 by SNR and clarify in the following (giving the same answer to both reviewers).

We use the term SNR to describe the mixing proportion of (a) data that contains signal about
 sequential events and (b) the data that does not contain any signal (the data which comes
 from the baseline resting-state session). For example, let's assume that the data from the
 sequence trials had an SNR of 1 and therefore was equally composed of signal (s) and noise
 (n): $D_{seq} = 1s + 1n$. In this example, we would regard the signal-to-noise ratio as 1. Now
 the sequence trial data is combined with data from the resting-state session that only contains
 noise $D_{rest} = n$. The mixing proportion is κ , such that the synthetically combined data, D_{κ} ,
 becomes $D_{\kappa} = \kappa D_{seq} + (1 - \kappa) D_{rest} = \kappa(1s + 1n) + (1 - \kappa)n$. The result will be that D_{κ}
 is composed of $\kappa s + 1n$, and hence has an SNR of κ instead of 1. In order to make the meaning
 of the term “SNR” in the current study context more clear, we added the following sentence
 to the Results section:

*“Thus, here we use the term SNR to describe the relative mixing proportion of (a)*
 *data from the task, which contains sequential signal, with (b) data from the pre-*
 *task resting-state session, which contains only noise. Note that this is different from*
 *the common definition of SNR in univariate fMRI as the ratio of average signal to*
 *standard deviation over time.”*

**Q20: Page 17:**

*“We did not mask stimulus sequences to establish a realistic scenario for the detec-*

*tion of replay events which occur spontaneously during rest with sufficient inter-event*
*intervals that allow separation in time.”*

*Perhaps this could be clearer as: “In order to establish a realistic scenario for the detection of*
*temporally separated spontaneous replay events, we did not mask stimulus sequences.”*

**A20:** Thanks, we adjusted the sentence following the reviewer’s suggestion:

*“In order to establish a realistic scenario for the detection of temporally separated*
*spontaneous replay events during rest, we did not mask stimulus sequences.”*

**Q21: Page 17:**

*“Second, we show that probabilistic classifiers provide major statistical improvements*
*compared to analyses focused on the decoded category with the highest classification*
*probability.”*

*I think the wording here could be improved. The key advantage seems to arise from using*
*a continuous “strength” of evidence versus a thresholded category signal – the fact that the*
*strength score is expressed as a probability is (I think) inessential. The key distinction, I think,*
*is whether or not a take-the-max threshold is applied to the classifier evidence?*

**A21:** We thank Reviewer 1 for this comment. The reviewer is right that the main improvement
of our method comes from analyzing the continuous ordering of classifier probabilities within a
single TR compared to the order of categorial labels with the highest classification probability
across TRs. In order to further clarify this aspect for the reader, we followed the reviewer’s
suggestion and improved the wording as follows:

*“Second, We show that ~~probabilistic classifiers~~ the analyses of classifier probabili-*
*ties provide major statistical improvements compared to analyses focused on the*
*decoded category with the highest classification probability (as in [53]). The key ad-*
*vantage is that probabilistic classifiers provide a continuous metric of classification*
*evidence and thereby allow detection of sequentiality within a single measurement*
*(i.e., within a single TR). This results in significant information gain compared to*
*the assessment of sequential ordering that considers only a single label per TR.”*

**Q22: Page 22:**

*“During the waiting period participants were listening to bird sounds ...”*

*Replace with “During this ...” – which will clarify that the bird sounds refer to the 16s delay*
*period with the gray fixation.*

**A22:** We thank the reviewer for this suggestion and have adjusted the sentence as pro-
posed:

*“During ~~the~~ this waiting period participants were listening to bird sounds (which*

*can be downloaded from [...] in order to keep them moderately entertained without*
*additional visual input.”*

**Q23: Page 27:**

*“This procedure was repeated eight times so that each task run served as the training*
*set once.”*

*Replace “training” with “testing”?*

**A23:** We thank the reviewer for reporting this mistake that we corrected as suggested:

*“This procedure was repeated eight times so that each task run served as the ~~training~~*
*testing set once.”*

**Q24: Page 32:**

*“The main question we asked for this analysis was to what extend . . . ”*

*Replace “extend” with “extent”?*

**A24:** We thank the reviewer for reporting this typo, which we fixed in the manuscript as
shown below:

*“The main question we asked for this analysis was to what extend~~t~~ we can infer the*
*serial order of image sequences from relative activation differences in fMRI pattern*
*strength within single measurements (a single TR).”*

**Q25: Figure S4:** *There appears to be a mismatch between the caption text and the panel*
*labels.*

**A25:** We thank the reviewer for reporting this error. We have corrected the Figure caption
accordingly.

Answers to Reviewer 2

Remarks to the Author

*“In this manuscript, a multivariate approach allowing for the detection of sequen-*
 *tially activated neuronal patterns at a temporal spacing of less than 100ms is pre-*
 *sented. This is a novel and exciting advancement in fMRI methodology, as classical*
 *univariate detection fails even at much slower rates. This paper opens up a new*
 *avenue in fMRI research that may rival the approaches used in EEG and MEG.*
 *The precise methodology and results should be described more clearly. In addition,*
 *more insight or perhaps thoughtful speculation into why the results were obtained*
 *- grounded perhaps in aspects unique to the HRF - should be provided throughout.*
 *Overall, it was an enjoyable paper to read with exciting results having potentially*
 *far reaching implications.”*

We thank Reviewer 2 for providing such thoughtful comments. We are very glad to hear that
 our manuscript was an “enjoyable paper to read with exciting results having potentially far
 reaching implications” and that the reviewer regards our approach as a “novel and exciting
 advancement in fMRI methodology”.

Specific comments

**Q1: p. 3.** *An early paper for sub-TR fast event multivariate decoding likely should be*
 *mentioned for context: “Accurate decoding of sub-TR timing differences in stimulations of sub-*
 *voxel regions from multi-voxel response patterns,” Misaki et al, NeuroImage, 66, 623-633, 2013.*

**A1:** We thank the reviewer for reminding us to include this paper, which is indeed very
 relevant to mention in this context. We are now citing the study by Misaki et al. [7] in the
 Introduction (page 3) as an example for how MVPA can be leveraged to study short timing
 differences between stimuli:

*“Moreover, Misaki et al. [31] were able to decode onset differences in visual stim-*
 *ulation of only 100 ms when two stimuli were shown to one eye before the other.*
 *Interestingly, Misaki et al. [31] indicated that timing differences become most ap-*
 *parent in peak activation strength, rather than temporal aspects of the hemodynamic*
 *response function (HRF).”*

**Q2: p. 3, line 50:** *An odd recurring grammatical error should be corrected: ... “have*
 *properties that allow to detect them more accurately.” This perhaps should be changed to “have*
 *properties that allow more easy detection.” ... or something like that, depending on the intended*
 *meaning of the sentence (which is hard to interpret). Basically “to detect” is not correct.*

**A2:** We thank the reviewer for this suggestion and have adjusted the sentence as proposed on

page 3.

“Second, some fast sequence events have properties that **make it easier to detect**
**them** ~~allow to detect them more easily.~~”

We have also corrected the phrasing “allow to” throughout the manuscript and the abstract.

**Q3: p. 4, line 65:** “identified” should be “identify.”

**A3:** We thank the reviewer for this comment. However, we believe that “identified” was
correct in this case as it was referring to the authors of Schuck and Niv [8] who “**identified**
fast sequential hippocampal pattern reactivation in resting humans using fMRI”. Anyway, we
take the reviewer’s comment as a helpful pointer to a potentially confusing syntax. We revised
the manuscript on page 4 as shown below:

“Recently, we have hypothesized that the properties of BOLD signals mentioned
above should enable the investigation of rapid neural dynamics. **Indeed, using fMRI,**
**we** ~~and~~ identified fast sequential hippocampal pattern reactivation in resting humans
~~using fMRI~~ [53].”

**Q4: p. 4, line 64-66:** It might be useful to mention the role that higher spatial resolution
would play in improving detection. (i.e. less partial volume averaging of non-activation related
signals).

**A4:** We thank the reviewer for this suggestion. Rather than mentioning the role of higher
spatial resolution in the Introduction, we have now included the following sentence in the
Discussion (page 24):

“**Further, increased spatial resolution might improve detection due to less partial**
**volume averaging of non-activation related signals.**”

**Q5: p. 4, line 70:** “allows to” is incorrect. “Allows inference of the speed ...” is perhaps
more correct.

**A5:** We thank the reviewer for this suggestion, which we implemented accordingly:

“Second, we developed a modelling approach of multivariate fMRI pattern classifica-
tion time courses that validates our experimental results and allows ~~to infer~~ **inference**
**of** the speed of fast sequential neural process from the frequency spectra of our fMRI
sequentiality metric.”

**Q6: p. 4, line 78:** “to use” is incorrect. “Of using” is more correct.

**A6:** Thanks, we changed the wording as follow:

“As discussed above, we investigated the possibility ~~to use~~ **whether fMRI can be used**

**to** address two cornerstones of understanding signals resulting from fast activation
sequences: order detection and element detection.”

**Q7: p. 4, line 84:** “to detect” should be “detection of”

**A7:** We thank the reviewer for this proposed change, which, combined with a similar
suggestion by Reviewer 1 (see Q10 of Reviewer 1 above) resulted in the following update of the
sentence:

“The second effect, element detection, quantifies to what extent fMRI allows ~~to de-~~
~~tect which elements~~ **detection of elements that** were part of a sequence and ~~which~~
**that** were not. ”

**Q8: p. 5, line 111-112:** A concern in this approach is that classifiers were obtained
to differentiate each out of five images – as even with re-ordering, there is likely a signature
pattern for each that is a weighted composite of the five. Would it be possible to train classifiers
that are independent of the set with which they are trained on and created with?

**A8:** We thank Reviewer 2 for this comment. As described in the main manuscript and
Methods section, the slow trials used for classifier training were set up in way that all the
stimulus presentations that yielded the training set were presented in each order permutation.
The rationale behind this experimental design was that every stimulus is preceded by all the
other four stimuli with equal frequency. Any “spilling” of activations across trials can therefore
bias decoding. The possibility that the reviewer is raising, could include a classifier training
set with stimuli that are not included in the sequence trials. This is an interesting idea,
but unfortunately not possible with the current dataset. Note, however, that repetition trials
only included a subset of image categories, so the reviewers suggestion was already partially
implemented in some analyses.

**Q9: Figure 2.** It’s not clear what you mean by “forward” and “backward” period. Is it
simply time before and after peak? Why call it forward and backward? Even from reading the
text below, it’s not clear why you chose those terms.

**A9:** We apologize that it has not become clear what we mean by the forward and backward
periods. The forward and backward periods reflect that the activation strength of items will first
be in the same order as the sequence order (i.e., forward ordered), while later the strength will be
in the reverse order relative to the sequence (hence, backward order). The time periods during
which these two types of ordering occur were called forward and backward periods. Note, that
we derived the existence and duration of these periods from a simple model used to predict the
expected time courses of classifier evidence during sequential events. Our modelling approach
indicated that following two successive events, the signal in earlier TRs will be dominated
by the first event and the activation strength of the two events will be proportional to the
event order during the sequential process, i.e., the early event has higher activation strength
compared to the second, later, event. After half a cycle of the assumed sine-wave response

function, this pattern reverses and the second event then has higher activation strength than
the first event. This caused the above mentioned forward and backward ordering. We have
adapted the manuscript to make this more clear as follows:

*“Simply put, this means that the strength of overlapping activations will initially be*
*ordered “forward”, in the same way as the sequence, i.e., earlier items will be acti-*
*ated stronger. In a later period, however, this will reverse and result in backwards*
*ordering, i.e., earlier items will be activated less. In summary, three predictions*
*therefore arise from this model: (1) the first event will dominate the signal in earlier*
*TRs and activation strengths will be proportional to the ~~ordering of events~~ true event*
*order during the sequential process; (2) in later TRs, the last sequence element will*
*dominate the signal, and the activation strengths will be ordered ~~in reverse~~ back-*
*wards; (3) the duration and strength of these two effects will depend on the fitted*
*response duration and the timing of the stimuli as specified above (Fig. 2e, Equations*
*4–6, see Methods). For sequences with more than two items (as in sequence trials,*
*see below), δ is defined as the interval between the onsets of the first and last sequence*
*item. To reflect the relation between the true order and the activation strength, we*
*henceforth term the above mentioned early and late TRs the forward and backward*
*periods, and consider all results below either separately for these phases, or for both*
*relevant periods combined (calculating periods depending on the timings of image*
*sequences and rounding TRs, see Methods).”*

**Q10:** *p. 8. It would be useful at this point to show an example of the classifier patterns for*
*each image to give a sense of how different they were. Further, it would be helpful to calculate*
*the spatial correlation of the classifiers, just as a reality check and to provide a sense of the*
*sensitivity from a different perspective. Were there specific features within the multivariate*
*patter that helped to drive the discernibility?*

**A10:** Example average activation patterns for the five different classes from one participant
are shown in Fig. R5 below. As can be seen, the five stimulus categories activate a mix of
overlapping and non-overlapping sets of voxels. Note that the “patchiness” of activation maps
reflects that our mask only included grey-matter voxels. Calculating the spatial correlations, as
the reviewer requests, indicates that the patterns were slightly negatively correlated, ranging
from $r = .02$ for the correlation between cat and face to $r = -.44$ for the correlation between
cat and house (see Table 1 below). Note that negative correlations can sometimes be caused
by standardization. We chose to report correlations that are most reflective of our decoding
results, in which normalized patterns were used (as common in MVPA).

Unfortunately, there is no straightforward way to determine which features helped the classifier
to discern the categories, as the weights cannot be easily interpreted [cf. 9]. We hope that the
average activation patterns shown here provide some insight and report these, along with the
correlations, in the SI of the revised manuscript and refer the reader to it in the Results:

Table 1: Average correlation between average spatial patterns associated with each image category

	Cat	Chair	Face	House	Shoe
Shoe	-.33	-.16	.29	-.15	1
House	-.44	-.20	-.37	1	
Face	.02	-.31	1		
Chair	-.23	1			
Cat	1				

Figure R5: Spatial distribution of mean voxel activations in one example participant for five stimuli Averaged patterns of voxel activations (colors) for the five decoded stimuli (horizontal panels) in one example participant (sub-01) plotted against the participant’s individual defaced structural scan.

*“Spatial patterns associated with image categories indicated a mix of overlapping*
 *and non-overlapping sets of voxels, and average correlations between the mean voxel*
 *patterns were negative (see SI).”*

**Q11:** *p. 10, Figure 3: It would be useful to show in the figure what the TR was rather*
 *than have the x axis in units of TR without a reference to absolute time.*

**A11:** We thank the reviewer for this suggestion. In order to maintain consistency throughout
 the manuscript, we have decided to keep the x-axes in TRs but now explicitly indicate in the
 Figures that **“1 TR = 1.25 s”**. We have not only added this additional label to the subpanels
 in Figure 3 but also to all other relevant plots in Figure 2 and 4. Furthermore, the equivalent
 of one TR in absolute time is also stated in each Figure caption.

**Q12:** *P 11. A clear difference in forward vs backward interference and detection is reported*
 *but, at least in this section and elsewhere I don't see any insight as to why this is. I wonder*
 *if backwards masking plays a role. Another possibility is the variability of the hemodynamic*
 *response as a function of time. Many studies have shown that the HRF shows significantly*
 *less variability during the rising phase and more variability during the falling phase. Another*
 *question related to sensitivity is the number of voxels used in the classifiers.*

**A12:** We thank the reviewer for raising this point and agree that the differences in forward
 versus backward interference should be discussed in more detail. A similar question was raised
 by Reviewer 1 (Q5) and this question also seems to relate to Q16 discussed below. Unfortu-
 nately, at this point we can only offer speculation about this effect. One potential explanation
 might be that the long tail of the falling slope of the HRF offers more time for measurement
 opportunities but it also seems possible that HRF variability is reduced as a function of time
 (as the reviewer suggests). Most speculatively, it is known that in some cases postsynaptic
 neurons can produce neurotransmitters that diffuse back to presynaptic neurons [e.g., 6]. Such
 retrograde pathways can be inhibitory. In this vein, we now write in the Discussion:

*“The origins of this asymmetry are not entirely clear. It seems possible that they*
 *reflect the benefits of the last item not being followed by another activation that could*
 *impede its detection. A relation to the asymmetric shape of the HRF, to changing*
 *HRF variability with time and even to inhibitory retrograde neurotransmitters [e.g.,*
 *75] cannot be ruled out.”*

The “number of voxels used in the classifiers” were already reported in the submitted manuscript
 on page 4 in the section on “Training fMRI pattern classifiers on slow events” and the relevant
 sentence reproduced below with the indication of the number of voxels highlighted in yellow
 for convenience:

*“fMRI data were masked by a grey-matter-restricted region of interest (ROI) of*
 *occipito-temporal cortex, known to be related to visual object processing [11162 vox-*
 *els in the masks on average; cf. 53, 56-58.]”*

**Q13:** *In the paper, it is difficult to relate the modulations in SNR to SNR in real data.*
 *Typically, SNR is calculated as the average time course signal divided by the standard deviation*
 *over time. How was your use of SNR calculated? It should be mentioned here so that tying into*
 *real-world situations would be more direct.*

**A13:** We thank the reviewer for asking this question, which was also raised by Reviewer 1
 (see Q19 by Reviewer 1 above). We apologize that it has not become clear what we meant by
 SNR and clarify in the following (giving the same answer to both reviewers):

We use the term SNR to describe the mixing proportion of (a) data that contains signal about
 sequential events and (b) the data that does not contain any signal (the data which comes
 from the baseline resting-state session). For example, let's assume that the data from the
 sequence trials had an SNR of 1 and therefore was equally composed of signal (s) and noise
 (n): $D_{seq} = 1s + 1n$. In this example, we would regard the signal-to-noise ratio as 1. Now
 the sequence trial data is combined with data from the resting-state session that only contains
 noise $D_{rest} = n$. The mixing proportion is κ , such that the synthetically combined data, D_{κ} ,
 becomes $D_{\kappa} = \kappa D_{seq} + (1 - \kappa) D_{rest} = \kappa(1s + 1n) + (1 - \kappa)n$. The result will be that D_{κ}
 is composed of $\kappa s + 1n$, and hence has an SNR of κ instead of 1. In order to make the meaning
 of the term "SNR" in the current study context more clear, we added the following sentence
 to the Results section:

*“Thus, here we use the term SNR to describe the relative mixing proportion of (a)*
 *data from the task, which contains sequential signal, with (b) data from the pre-*
 *task resting-state session, which contains only noise. Note that this is different from*
 *the common definition of SNR in univariate fMRI as the ratio of average signal to*
 *standard deviation over time.”*

**Q14: p. 17:** *The word “localize” in line 356 usually implies spatial localization. Rather, it*
 *seems you mean decode or detect. This statement is also incorrect in that for two decades at*
 *least it has been shown that BOLD can decode event-related responses that are as brief as 16*
 *ms. The key difference here is that they are among sequences of other interfering events spaced*
 *closely together. This point should be made more clear.*

**A14:** We agree with Reviewer 2 that “localize” might be misleading in this context. As
 suggested by the reviewer, we (1) replaced “localize” with “detect” as shown below, (2) made
 clear that it is the combination of BOLD fMRI and multivariate probabilistic decoding methods
 that enables the detection of fast neural event sequences, and (3) that this is also true for closely
 timed events.

*“We demonstrated that BOLD fMRI in combination with multivariate probabilistic*
 *decoding can be used to localize detect sub-second neural events sequences of closely*
 *timed neural events non-invasively in humans.”*

**Q15: p. 17, line 390:** *grammatical error - “...because it allows to unravel.” This should*

*be “allows the unravelling of”*

**A15:** Thanks, we corrected the grammatical error as suggested and updated the manuscript
(along with other changes):

*“The key advantage is that probabilistic classifiers provide a continuous metric of*
*classification evidence and thereby allows to unravel the detection of sequential or-*
*dering within a single measurement (i.e., within a single TR)”*

**Q16:** *p. 18, lines 395-424: At this point it would be helpful to speculate on why the falling*
*slope of the HRF drives the sequentiality and contributes to re-ordering. Is it due in some way*
*to the nature of the HRF? What does this imply about the HRF?*

**A16:** This question seems to relate to Q13 answered above. We can only speculate about
the role of the HRF in driving sequentiality and contributing to re-ordering. One potential
explanation might be that the long tail of the falling slope of the HRF offers more time for
measurement opportunities. Importantly, and independently of the HRF, the end points of the
sequences are followed by a longer pause (both in our experiment and in replay). This ensures
that the activation of the last element does not “compete” against the next occurring activation.
In line with this idea, the results from the repetition trials show that events preceded by other
events are easier to detect than events followed by other events. This may lead to a dominance
of the last event activation, during which the sequence appears to be reversely ordered (the last
element is activated most, the second to last element second to most, etc.).

**Q17:** *p. 19, line 438-440. The statement on the increasing sampling rate only partially*
*increasing power is too vague. In fact, it’s known that increased sampling rate always increases*
*power. The question is more along the lines of what is the relationship between sampling rate*
*and power. I would suggest addressing this a bit more carefully as the reader is left with no real*
*guidance on what sampling rate would be optimal. Do you think that the TR you use (1250 ms)*
*is optimal? What TR would be optimal?*

**A17:** Our statement may indeed have been somewhat vague. We agree that sampling
rate in general increases power. What we meant to say is that in the context of our specific
analysis, shorter TRs may lead to very little additional benefits. Our rationale was based on
two ideas: First, in MRI, longer TRs provide better SNR, as they allow more time for the
longitudinal magnetization to reach its maximum. Second, the main factor making detection of
fast event sequences difficult is the signal delay due to the HRF. As long as the underlying (slow)
HRF dynamics remain the same, an increased sampling rate will help only slightly to identify
temporal order of fast neural event sequences. Moreover, our method mainly capitalizes on
the ordering of activation patterns within individual TRs, rather than the transitions between
TRs.

Overall, we therefore believe that it eventually remains an empirical issue and unfortunately
we cannot answer confidently whether our TR was optimal. Interestingly, in a study by Misaki

et al. [7] aimed at detecting temporal differences in monocular visual stimulation of 100 ms,
downsampling the data from an original TR of 250 ms to 500, 1000, 2000, or 4000 ms did not
affect decoding accuracy scores [see 7, their Fig. 6].

In summary we think that researchers should carefully deliberate if the costs of shorter TRs (less
SNR, decreased spatial resolution, etc.) are really met by benefits for power in the temporal
domain. To make this aspect clearer to the reader, we included the following sentence in the
Discussion:

~~“It should be noted, however, that an increased sampling rate will only partially in-~~
~~crease power, since the extended HRF duration ensures measurement opportunities~~
~~up to 10 s after the sequence. Whether an increased sampling rate would be benefi-~~
~~cial for the detection of fast event sequences is difficult to predict. First, longer TRs~~
~~provide better signal-to-noise ratio as they allow more time for longitudinal mag-~~
~~netization. In addition, faster sampling will not affect the underlying (slow) HRF~~
~~dynamics that impede the identification of temporal order of fast neural event se-~~
~~quences. Sampling the activation time courses at a faster rate might not reveal more~~
~~information about the sequential process under investigation. Whether shorter TRs~~
~~can make up for the downsides in spatial resolution and SNR therefore seems an~~
~~empirical question.”~~

**Q18:** *p. 27. Why was spatial smoothing used? Wouldn't this reduce decoding power?*
*Detailed decoding information is known to be eliminated by spatial smoothing.*

**A18:** We thank the reviewer for this comment, which addresses a prevailing controversy about
the utility of spatial smoothing for multivariate pattern analysis [for a recent review includ-
ing this discussion, see e.g., 10]. Several previous studies suggest that minimal to moderate
spatial smoothing has, if anything, a positive effect on the distinctiveness of neural activation
patterns [see, e.g., 11–16] and thereby decoding accuracy, while a few studies also report disad-
vantageous effects [17, 18]. In an fMRI-study with the similar aim of decoding sub-TR timing
differences of monocular visual stimulation, the authors note that “smoothing did not affect
the decoding accuracies very much” [7]. Importantly, most studies seem to agree that little
or no spatial smoothing should be used when decoding fine-grained spatial information (e.g.,
columnar activation differences in V1), but that spatial smoothing can be beneficial especially
if the activation patterns of interest are anatomically more distributed (e.g., when decoding
affective states) [e.g., 10, 19, 20]. The large majority of authors also note that the effects and
degree of smoothing on decoding results are influenced by additional factors, like the scale of
neural organization, the amount of head movement and the cognitive task performed during
scanning, or other pre-processing steps like detrending. In the current study, we decoded five
visual object categories known to elicit neural activation patterns that are broadly distributed
across occipito-temporal brain regions [cf. 21]. In line with previous results, we found that
moderate levels of smoothing (FWHM of 4–6 mm) yielded higher decoding as compared to

weak (2 mm) and strong (8 mm) smoothing (see Fig. R6). Note, that these analyses were
 conducted in an independent pilot sample of $N = 3$ participants that was collected before main
 data collection conducted. Hence, we assume that in our case spatial smoothing has benefitted,
 rather than impaired, the decoding power.

Figure R6: Influence of the degree of smoothing, temporal shift, and classification approach on cross-validated decoding accuracies. Cross-validated classification accuracy in decoding the five unique visual objects in occipito-temporal data during task performance (y-axis) depending on the degree of smoothing (Full Width at Half Maximum (FWHM) in mm; x-axis), the temporal shift from stimulus onset (in s; colors), and different classification approaches (logistic regression and two variants of linear support vector machines (SVMs); panels). $N = 3$ participants from an independent pilot sample. Chance level is 0.2 (horizontal blue line).

**Answers to Reviewer 3**

**Remarks to the Author**

*“In this manuscript, Wittkuhn and Schuck examine with novel multivariate analyses methods if*
*fast sequential events can be detected with fMRI. Due to the sluggishness of the BOLD signal, the*
*non-invasive investigation in humans of fast events has been so far implemented with MEG and*
*EEG. The authors had two main aims related to the investigation in fMRI of signals resulting*
*from fast events (1) order detection and (2) element detection (to be able to determine if a*
*sequence item is being reactivated).*

*In order to answer these questions, the authors trained probabilistic pattern classifiers for five*
*objects that were presented multiple times in slow sequences five-items long, each with a differ-*
*ent order permutation. The decoding probability for each item was assessed at these slow trials,*
*and the authors calculated a fitted response model using averaged parameters for each object*
*(class) decoded. Based on the response model mentioned, the authors were able to approximate*
*expectations for signals during sequential events, by computing differences in probabilities be-*
*tween time-shifted events. The idea is that, even if the BOLD response is slow, the decoding*
*probabilities for the different objects should be able to inform about sequence order. In the*
*same task, there were sequence trials, consisting of permutations of the same five objects, but*
*this time presented at different speeds. The authors tested the probabilistic classifiers on these*
*sequence events in a series of clever analyses (figure 3), finding that the probabilities of the de-*
*coded objects reflected the real sequence order. These analyses combined decoding probabilities*
*for each object within each TR, as well as across-TRs. The results nicely fit with the predicted*
*model. Importantly, for earlier TRs not only the earliest objects in the sequence show a higher*
*decoding probability, as well as the opposed pattern for later TRs, but also the time-course of*
*the regression slopes between event position and probability reflect the correct sequence order.*
*Next, the authors tackled the problem of element detection. For this, another condition had*
*been introduced in the task design: fast sequences 9-items long that consisted of repetitions of*
*two of the five objects, one of the two objects, the first or the second appearing, was presented*
*eight times in a row. The authors found that, despite interference from the repeated object, the*
*other object that had been presented was still detected. Moreover, they found that the item at*
*the end of the sequence was easier to decode.*

*Finally, the authors simulated a lower signal to noise ratio to assess if sequence events can still*
*be decoded in such circumstances, similar to those of events like hippocampal replay. The authors*
*inserted in the rest state recordings fast or slow sequences and found that even in conditions*
*of higher noise and the lowest number of sequence inserts, they could still detect fast and slow*
*sequences.*

*Overall, I believe this manuscript was exceptionally well written and it was very clear. They*
*have developed clever analyses that have allowed them to be able to detect fast sequence events*
*in fMRI. All the analyses were thoughtful and in general were easy to follow. Overall, it is*

*convincing and I would recommend it for publication.”*

We are glad to hear that Reviewer 3 found our manuscript “exceptionally well written” and the
analyses clever and thoughtful and we are pleased that the reviewer recommends our manuscript
for publication.

**Comments**

**Q1:** *The authors should make more clear how this manuscript fits with their previous paper*
*(Schuck and Niv, 2019).*

**A1:** We thank the reviewer for this comment. Our previous submission included one para-
graph in the discussion related to this question but we apologize if our explanations were not
sufficiently clear. Generally, our current results support the finding by Schuck and Niv [8] that
fMRI can be used to study replay. The two most important aspects we verify here relate to
the potential speed and direction of the sequences reported in our previous paper.

First, the results reported in Schuck and Niv [8] clearly indicated sequential reactivation, but
we could not be sure that these reactivations were fast. We stated this in the discussion of our
original publication: “*Although we provide evidence that our results could reflect fast sequential*
*replay events with speeds similar to what was found in these reports, we cannot infer the speed*
*of the replay directly from our observations.”. With the present paper we show that fMRI can*
*be sensitive to sequential reactivation at fast timescales.*

Second, the Schuck and Niv [8] paper reported forward rather than reverse replay. We spec-
ulated in the Discussion that this may have several possible explanations (“*[this] may suggest*
*that in our experiment, replay was related more to memory function rather than planning. [...]*
*Alternatively, decoding may have been dominated by the falling slope of hemodynamic responses,*
*which could lead to order inversions.”). In the present paper we clarify the origins of forward*
*and backward ordering of fMRI activation patterns. We show that probabilistic classifier evi-*
*dence in earlier TRs reflects the forward order of the sequences while this pattern reverses in*
*later TRs. Importantly, we demonstrate an asymmetry in decoding early versus late sequential*
*events. This can therefore lead fMRI sequences to appear in the reverse order relative to the*
*underlying neural sequences.*

While these implications are exciting advancements, it is important to keep in mind that the
benefits of experimental setting came at the cost that they also introduced important differences
from a replay study in various regards (e.g., different cortical area). We now include all these
points in a revised paragraph:

*“Our results ~~offer methodological advancement and~~ deepen the understanding of our*
*previous findings [53] in two ways. First, we provide additional empirical evidence*
*that our sequentiality analyses based on multivariate fMRI pattern classification are*
*indeed sensitive to fast neural event sequences. ~~using an~~ To this end, we used an*

experimental setup where the order of sequential events is known – in contrast to
analyses of resting-state data in Schuck and Niv [53] where the order and speed of
event sequences can only be assumed. **Second, Schuck and Niv [53] observed forward**
**ordered replay. Our present study clarifies the origins of forward and backward or-**
**dering of fMRI activation patterns. We show that probabilistic classifier evidence in**
**earlier TRs reflects the forward order of the sequences while this pattern reverses in**
**later TRs. Importantly, we demonstrate an asymmetry in decoding early versus late**
**sequential events. This can therefore lead fMRI pattern sequences to appear in the**
**reverse order relative to the underlying neural sequences. This represents a crucial**
**insight, given the different functional roles assigned to forward and backward replay**
**[see e.g., 32].”**

**“In addition, our study introduces important methodological advancements that go**
**beyond our original publication. Second, We show that probabilistic classifiers the**
**analyses of classifier probabilities provide major statistical improvements compared**
**to analyses focused on the decoded category with the highest classification probability**
**(as in [53]). The key advantage is that probabilistic classifiers provide a continu-**
**ous metric of classification evidence and thereby allows to unravel the detection of**
**sequential ordering within a single measurement (i.e., within a single TR). This re-**
**sults in significant information gain compared to the assessment of sequential order-**
**ing that considers only a single label per TR. Third, Moreover, we leverage frequency**
**spectrum analysis in a novel approach to make inferences about the speed of the**
**sequential neural process. Although the sampling rate (i.e., the TR) of fMRI is usu-**
**ally less than the speed of replay events, frequency spectrum analyses can charac-**
**terize the speed of fast sequential events during rest. Together, these methodological**
**advances offer new insights into previous fMRI studies investigating hippocampal**
**replay in humans, including our own work [53]. In addition, some caveats have**
**to be noted. Second, Our results indicate that the sequentiality in fMRI analyses is**
**mainly influenced by the first and last element of a fast sequence. Given that replay**
**events are often structured by task-relevant features like the start and goal location in**
**a spatial environment [e.g., 76], analyzing the transitions between the corresponding**
**decoded events will offer insights into the content and functional role of fast replay**
**events. Moreover, it is important to keep in mind that the benefits of our experimen-**
**tal setting came at the cost that they also introduced important differences from a**
**replay study in various regards, including the focus on extra-hippocampal activations**
**and sensory stimulation.”**

**Q2: On Line 124:**

**“This analysis confirmed delayed and distinct increases in the estimated probability**
**of the true stimulus class given the data, peaking at the fourth TR after stimulus**
**onset, as expected (Fig. 2b).”**

*In figure 2b, it is unclear why it was expected the peak on the fourth TR and would be informative*
 *to explain it more in the main manuscript.*

**A2:** As stated in the main manuscript, the classifiers were trained on “fMRI data acquired
 3.75 to 5 s after stimulus onset (corresponding to the fourth TR, see Methods)”. This time
 window is roughly in line with the expected peak of the HRF around 6 seconds after stimulus
 onset, and was chosen based on piloting results that indicated that best decoding on slow trials
 could be achieved from this TR (see Fig. R7 below).

Figure R7: Influence of the degree of smoothing, temporal shift, and classification approach on cross-validated decoding accuracies. Cross-validated classification accuracy in decoding the five unique visual objects in occipito-temporal data during task performance (y-axis) depending on the degree of smoothing (FWHM in mm; x-axis), the temporal shift from stimulus onset (in s; colors), and different classification approaches (logistic regression and two variants of linear support vector machines (SVMs); panels). $N = 3$ participants from an independent pilot sample. Chance level is 0.2 (horizontal blue line).

Thus, when the classifiers are applied to seven TRs following stimulus onset, it can be expected
 that the highest classification probabilities will be obtained at the TR the classifier was trained
 on (i.e., the fourth TR as stated above). To make this even clearer to the reader, we added the
 following additional information to the main manuscript:

*“This analysis confirmed delayed and distinct increases in the estimated probability*
 *of the true stimulus class given the data, peaking at the fourth TR after stimulus*
 *onset, as expected given that the classifiers were trained on data from the fourth TR*
 *following stimulus onset (Fig. 2b).”*

**Q3:** *Regarding the repetition trials, if one reads first the main manuscript, then the methods*
 *and finally the supplementary methods, it is easy to get confused about the structure of the*
 *repetition trials. It is unclear in the main manuscript how were the sequences constructed.*
 *In figure 1c, it is said that “Repetition trials were always fast (32 ms ISI) and contained two*
 *visual images of which either the first or second was repeated eight times”. The same structure*
 *is described on Line 237 “we investigated classification time courses in repetition trials, in which*

*only two out of the five possible images were shown. Crucially, one image was repeated, while*
*the other one was shown only once”. Both figure 1c, and comments in the main manuscript,*
*make the reader understand that one of the categories was repeated only once and that happened*
*either at the start or at the end of the nine items long repetitions sequence. There is a mention*
*in line 260: “Additional conditions with intermediate levels of repetitions are reported in the SI*
*(Fig. S1e)”. I would suggest a bit more explanation regarding the different ‘levels of repetitions’*
*in this line of the main manuscript, as well as making explicit in the methods (line 574) that*
*only two repetition patterns were presented in the main manuscript among all the ones described*
*in the methods (line 574). It is not mentioned in the Methods and it becomes confusing. Only*
*by reading Fig S1.e can one understand well.*

**A3:** We thank the reviewer for this comment, which was also raised by Reviewer 1 (see Q9
of Reviewer 1) and apologize for the confusion. Following the reviewers’ suggestions, we now
added the following sentences to the main manuscript:

*“Note that our analyses focused on the two extreme cases of repetition trials with*
*one versus eight repetitions of the first versus second item (or vice versa) while the*
*experiment also included repetition trials with intermediate levels of repetitions (see*
*SI). Specifically, other repetition trials included cases in which the second item be-*
*gan to appear at each possible position from 2 to 9. The other repetition trials could*
*therefore include, for instance, three repetitions of the first and six repetitions of*
*the second image, or four repetitions of the first and five repetitions of the second*
*item, etc. The results reported in the SI indicate that effects in these trials smoothly*
*transition between the extremes shown in the main manuscript.”*

The following sentences were added to the Methods section:

*“Repetition trials included varying repetitions of two images in a sequence of nine*
*items in total. All analyses reported in the Results section focused on the two most*
*extreme cases, (1) the first image shown once followed by eight repetitions of the*
*second image, and (2) eight repetitions of the first image followed by the second*
*image shown once. Analyses of all intermediate levels of repetitions are reported in*
*the SI.”*

**Q4:** *I understand that the 13 TR reported in many figures (e.g, fig 3a) includes all the time*
*elapsed between the start of each sequence until the test starts, right?*

**A4:** Yes, all analyses of sequence and repetition trials focused on the time period from sequence
onset until the response screen, which was set to 16 s, roughly spanning 13 TRs. We added
this information in parentheses to the relevant section in the manuscript (see highlighted text
below):

*“This delay between visual events and response (roughly spanning 13 TRs; see x-*
*axes in Fig. 3a-b) allowed us to measure sequence-related fMRI signals without*

*interference from following trials, while the upcoming question did not necessitate*
*memorization of the sequence during the delay period.”*

**Q5:** *While this manuscript is a nice methodological paper, it might have potential questions*
*about how the brain works, which would in fact shed light on the last claim of the authors that*
*this method could might detect neural replay. To investigate if their method can be applied*
*to investigate neural mechanisms like hippocampal replay, the authors construct artificial fast*
*events with different levels of signal-to-noise ratio, by embedding different numbers of sequence*
*events into resting-state recordings. While this analysis was convincing to me, I suggest the*
*authors perform a new analysis in their current data that could shed light onto this question: the*
*authors could perform the same analyses, but training and testing the probabilistic classifiers on*
*a hippocampus mask. After all, if this method should be able to detect replay after the stimulus*
*has been presented, at least it should be able to detect hippocampal activity that responds to*
*on-screen stimuli. Another question would be if there are learning effects. For instance, across*
*event repetitions, stimuli will be learned and hence, the hippocampal activity might change. For*
*instance, across repetitions, one would expect the hippocampus to anticipate the last items in a*
*sequence.*

**A5:** We thank Reviewer 3 for this comment and the suggestion to test our classification
analyses in a region of interest (ROI) centered on the hippocampus. We followed the reviewer’s
suggestion and performed the same analyses but trained and tested the probabilistic classifiers
on a hippocampus mask. To this end, we created participant-specific anatomical masks of the
hippocampus based on the same automated anatomical labeling of brain surface reconstructions
from the individual T1-weighted (T1w) reference images that was used to create the anatomical
masks of occipito-temporal brain regions (as described in the Methods in the main manuscript).
An important requirement of our analysis approach is sufficient decoding performance to assert
that our classification approach is able to accurately decode the stimulus categories. Using
the hippocampal masks in our leave-one-run-out cross-validation approach revealed that the
average classification accuracy ($M = 20.47\%$, $SD = 1.56\%$, range = 16–25%) did not differ
from the chance baseline of 20%, $t_{(35)} = 1.81$, $p = .08$, $d = 0.30$ (one-sided one-sample t -test,
testing the a-priori hypothesis that decoding accuracy would be higher than chance; see Fig.
R8). We now report these results in the main manuscript at the end of the paragraph on
“Training fMRI pattern classifiers on slow events”:

*“Decoding in an anatomical ROI of the hippocampus did not surpass chance level (de-*
*coding accuracy: $M = 20.47\%$, $SD = 1.56\%$; $p = .08$, compared to chance, $d = 0.30$;*
*using the same decoding approach, see SI for details).”*

Further details are provided in a novel dedicated section in the SI as follows:

*“We also conducted a separate leave-one-run-out classification analysis to decode the*
*five stimulus categories from activation patterns in the hippocampus. To this end,*
*the same decoding approach was used but activity patterns were extracted from an*

*anatomical ROI centered on the hippocampus. The ROI was based on the same auto-*
*mated anatomical labeling of brain surface reconstructions from the individual T1w*
*reference images that were used to create the anatomical masks of occipito-temporal*
*brain regions. No GLM-based feature selection was performed on activity patterns*
*from the hippocampus. Using the hippocampal masks in the leave-one-run-out cross-*
*validation approach revealed that the average classification accuracy ($M = 20.47\%$,*
*$SD = 1.56\%$, range = 16–25%) did not differ from the chance baseline of 20%,*
*$t_{(35)} = 1.81$, $p = .08$, $d = 0.30$ (one-sided one-sample t-test, testing the a-priori hy-*
*pothesis that decoding accuracy would be higher than chance; see Fig. S2). The*
*implications of this finding are further discussed in the main manuscript.”*

The fact that we did not achieve the necessary classification performance in the hippocampus
might not be surprising given that the slow trials used for training the classifiers did not require
any memorization and were explicitly designed to elicit object-specific activation patterns in
occipito-temporal regions. During slow trials, participants’ only task was to attend to the
stimuli and respond when a stimulus was presented upside-down. We deliberately chose to
develop and verify our methods based on neural data processing visual information to provide
optimized test conditions for the expected effects. Of note, the current analysis approach is
based on methods developed in our own previous work in which we demonstrated the success
of our methods in hippocampal data [8]. Successful decoding in the hippocampus has been
demonstrated in particular for tasks that used stimulus material known to elicit hippocampal
activity (e.g., visual scenes and landscapes) or for episodic memory paradigms [see e.g., 22].
Finally, we note that while we did not find sufficient classification performance in hippocampal
data, additional analyses in post-task resting state data as inquired by Reviewer 1 (see R1’s Q6)
demonstrate that our current analyses in occipito-temporal data are sensitive to presumable
reactivation events during post-task rest. Interestingly, this indicates that the reactivations in
occipito-temporal brain areas occurred independently of any task-related involvement of the
hippocampus. We mention this aspect now in a novel paragraph in the Discussion:

*“Of note, replay during rest reflected cortical reactivations in the occipito-temporal*
*brain regions. Given that we were not able to decode on-task stimulus representations*
*in the hippocampus, it remains unclear if reactivations occurred independently from*
*(task-related) involvement of hippocampus or if we were simply not able to detect*
*concurrent reactivation in the hippocampus. One potential reason why we found no*
*hippocampus involvement could be that the oddball detection paradigm used for slow*
*trials to train the classifiers involved no mnemonic task component and therefore*
*was not suitable to activate the hippocampus. Our previous work [53] has already*
*demonstrated the success of our methods in hippocampal data. Taken together, our*
*results indicate that our method allows the uncovering of fast task-related reactiva-*
*tions during rest and highlight the importance of task design for detecting replay in*
*humans using fMRI.”*

Figure R8: Classification accuracy in the hippocampal mask. (a) Cross-validated classification accuracy in decoding the five unique visual objects in hippocampal data during task performance (in %; y-axis). Chance level is 20% (dashed line). Each dot corresponds to averaged data from one participant. The error bar represents ± 1 SEM. (b) Time courses (in TRs from stimulus onset; x-axis) of probabilistic classification evidence (in %; y-axis) for all five stimulus classes. No probability increases for any stimulus presented (black lines) on a given trial (gray panels) were found. Each line represents one participant.

**Q6:** *On Line 243, the authors state that*

*“Finally, varying whether the second or first item is short allowed us to investigate*
 *if the ability to detect sequence elements is asymmetrical, and possibly favors the*
 *detection of late over early events.”*

*They write this later hypothesis but it is not explained why.*

**A6:** We apologize that the hypothesis about an asymmetry in the detection of earlier versus
 later sequential items has not become clear enough. We clarified our reasoning in the following
 paragraph, which replaces the originally cited sentence:

*“To ask if detection of brief activations is differently affected by events occurring*
 *before versus after the single event, we varied whether the second or first item is*
 *short. We pose this question because the backward effects were consistently larger*
 *than forward effects in our sequentiality analyses reported above (Fig. 3c), suggesting*
 *asymmetric detection sensitivity.”*

To further clarify to the reader what the hypothesis about asymmetric detection of early versus
 late sequence items would predict, we added the following sentence to the end of the relevant
 paragraph in the manuscript:

*“This implies that one briefly presented item at the end of a sequence will be easier*

*to detect than a briefly presented item at the beginning of a sequence, even though*
*both were equally close in time to another strong activation signal. In consequence,*
*short items followed by a longer item during repetition trials should be more difficult*
*to detect than those preceded by a long item. To test this idea, we considered the*
*above described two order conditions. We will term the case in which the first image*
*was shown briefly once and followed immediately by eight repetitions of a second im-*
*age the forward interference condition, because the forward phase of the sequential*
*responses suffers from interference. Correspondingly, trials in which the first image*
*was repeated eight times and the second image was shown once will be termed the*
*backward interference condition.”*

**Q7:** *In figure 4e, it is unclear what are “repetition 1” and “repetition 2”.*

**A7:** We thank the reviewer for noting that the *repetition* transition type in Fig. 4e is not
explained in the manuscript. We meant to refer to the *repetition of the first and second item*
respectively. We now clarified this by adding the following sentence to the end of the relevant
paragraph:

*“The full transition matrix is shown in Fig. 4e. Repetitions of the first or second*
*item are shown on the upper two diagonal elements (with all consecutive repetitions*
*of items labelled repetition in Fig. 4e), and were not considered in this analysis.”*

**Q8:** *Figure 5b and 5c of the main manuscript do not fit well. In figure 5c, I see higher absolute*
*slopes for the slow condition, followed by the fast condition, and then the rest condition. But*
*this is not what figure b shows. It is not well explained, neither in the text nor in the figure*
*explanation below.*

**A8:** We thank the reviewer for highlighting this difference, which might appear like an incon-
sistency. However, also after verifying our analyses again, we can provide an explanation for
the apparent difference between Fig. 5b and 5c: Fig. 5b in our submitted manuscript showed
the mean in *absolute* slopes across participants thus indicating the absolute size of apparent
sequentiality irrespective of its directionality. Fig. 5c, in contrast, showed the time courses
of the *non-absolute* slopes across all participants at every time point (at every TR), *not* the
*absolute* slope as the reviewer assumed, thus providing an impression about the directionality
of our sequence measure across time (and its consistency across participants). To clarify the
difference in absolute vs. non-absolute slopes to the reader, we adjusted the caption of Figure
5, as follows:

*“(b) Mean absolute regression slopes, as in (a). (c) Time courses of non-absolute*
*regression slopes in rest and sequence data. Vertical lines indicate trial boundaries.”*

**Q9:** *On Line 473, “Auditory feedback was used to anatomically separate the expected neural*
*activation patterns of visual stimuli and auditory feedback” is a bit unclear to me.*

**A9:** We thank the reviewer for this comment and gladly clarify. As described in the
 manuscript, an important goal of the study was to achieve optimal classifier performance in
 decoding the five visual objects used in the task. We suspected that using visual feedback
 stimuli like smileys or points, as commonly done in psychological and cognitive neuroscience
 experiments, would engage the same brain regions processing the five visual objects and could
 interfere with and potentially reduce classification performance. Thus, we decided to use au-
 ditory performance feedback (the coin and buzzer sounds for positive and negative feedback,
 respectively), which we expected to primarily activate different brain regions than visual stim-
 uli. Indeed, entering the search terms “visual” and “auditory” into the meta-analysis database
 Neurosynth [23, 24, see <https://neurosynth.org/>], which aggregates activation maps of brain
 regions preferentially related to the search term confirmed this impression: Studies related to
 the search term *visual* preferentially indicate occipital brain regions, while studies related to the
 search term *auditory* preferentially indicate temporal brain regions, with only minor overlap
 between these two activation maps (see Fig. R9). To clarify this aspect to the reader, we add
 the following sentence after the one cited by the reviewer:

“Auditory feedback was used to anatomically separate the expected neural activation
 patterns of visual stimuli and auditory feedback. **While auditory feedback is more
 likely to engage primarily temporal brain regions, visual stimuli are more likely to
 activate primarily occipital brain regions.**”

Figure R9: Automated Neurosynth (<https://neurosynth.org/>) activation maps based on an association test for the same coordinates. (a) Neurosynth activation maps based on > 3,000 studies displaying brain regions that are preferentially related to the search term “visual”. **(b)** Activation maps as in (a) but for the search term “auditory” and based on > 1,000 studies. According to the Neurosynth documentation, the *association test maps* shown here display “brain regions that are *preferentially* related to the search term” (here, “visual” and “auditory”). The website was assessed on August 5th 2020.

**Q10:** On Line 743 of the main manuscript, the authors state that “In order to ensure that
 the classifier estimates were not biased by relative differences in class frequency in the training
 set, the weights associated with each class were adjusted inversely proportional to the class
 frequencies in each training fold.” I understand that, given that there were 5 classes to decode,
 the frequencies used to adjust were 1/5 for the class of interest, and 4/5 for the ‘other’ class,

*comprising any other classes?*

**A10:** Yes, the reviewer is exactly right with this understanding. To further clarify for the
reader how the classifier weights were adjusted, we included an additional sentence to the
manuscript, following the reviewer’s paraphrasing (highlighted in yellow):

*“In order to ensure that the classifier estimates were not biased by relative differences*
*in class frequency in the training set, the weights associated with each class were*
*adjusted inversely proportional to the class frequencies in each training fold. Given*
*that there were five classes to decode, the frequencies used to adjust the classifiers’*
*weights were 1/5 for the class of interest, and 4/5 for the other class, comprising any*
*other classes. Adjustments to minor imbalances caused by the exclusion of erroneous*
*trials were performed in the same way.”*

**Q11:** *As I understand it, there were seven runs in this experiment. The probabilistic classifiers*
*applied in each run were different from those for the other runs, right? Because you applied a*
*leave-one-run-out cross-validation procedure. Line 799 of the main manuscript: “This procedure*
*resulted in fold-specific maps of t-values that were used to select voxels from the left-out run of*
*the cross-validation procedure. Note, that this approach avoids circularity (or so-called double-*
*dipping) as the selective analysis (here, fitting of the GLMs to the training set) is based on*
*data that is fully independent from the data that voxels are later selected from”. Were there*
*significant differences in decoding probability across-folds?*

**A11:** We thank Reviewer 3 for this comment and like to briefly clarify the decoding approach
used in the current study. The experiment consisted of eight task runs in total. We used a
cross-validated leave-one-run-out classification approach. To answer the reviewer’s question if
there were significant differences in decoding probability across folds, we calculated the mean
decoding accuracy in every run for every participant. The results are shown in Fig. R10.
The average fold-specific decoding accuracies were homogenous overall, ranging from 67.3% to
73.3%. A linear mixed effects (LME) model of the mean decoding accuracy scores including
the task run as the main fixed effect of interest and by-participant random intercepts revealed
a significant main effect of task run on mean decoding scores, $F_{7,245} = 2.38$, $p = .02$. Of note,
the mean fold-specific decoding accuracy was highest in task runs 1 (73.27%) and 5 (71.06%),
which were the first runs of each of the two sessions (each session consisted of four runs). Thus,
the minor differences in decoding accuracy between cross-validation folds were likely explained
by behavioral fatigue and slight decreases in attention to the stimuli rather than suggesting
issues with the classification approach.

**Q12:** *typos: On Line 268, the word ‘the’ should be removed. On Line 345 of the main*
*manuscript, the authors refer to fig 5f, when they are talking about figure 5h. On Line 348,*
*they again mention fig 5f when they are referring to figure 5i.*

**A12:** Thanks, we corrected these typos in the main manuscript as shown below. Please note,

Figure R10: Classification accuracy across cross-validation folds. Cross-validated classification accuracy in decoding the five unique visual objects in occipito-temporal data during task performance (in %; y-axis) for each cross-validation fold / task run (x-axis). Chance level is 20% (dashed line). Each dot and line corresponds to averaged data from one participant. Errorbar represents ± 1 SEM.

that the first typo was removed by adjusting the entire sentence.

*“Moreover, the probability of decoding within-sequence items depended on the con-*
 *dition and whether the item was repeated or not their position as well as the their*
 *duration (number of repetitions).”*

*“Inserting fast event sequences into rest led to power increases in the frequency*
 *range indicative of 32 ms events (~ 0.17 Hz, Fig. 5h-i, left panel), in line with our*
 *findings above.”*

*“Inserting slow (2048 ms) sequence events into the rest period showed a markedly*
 *different frequency spectrum, with an increase around the frequency predicted for*
 *this speed (~ 0.07 Hz, Fig 5h-i, right panel).”*

References

- [1] Karola Kaefer, Michele Nardin, Karel Blahna, and Jozsef Csicsvari. Replay of behavioral
sequences in the medial prefrontal cortex during rule switching. *Neuron*, Feb 2020. ISSN
0896-6273. doi: [10.1016/j.neuron.2020.01.015](https://doi.org/10.1016/j.neuron.2020.01.015). URL [http://dx.doi.org/10.1016/j.](http://dx.doi.org/10.1016/j.neuron.2020.01.015)
[neuron.2020.01.015](http://dx.doi.org/10.1016/j.neuron.2020.01.015).
- [2] H Freyja Ólafsdóttir, Francis Carpenter, and Caswell Barry. Coordinated grid and place
cell replay during rest. *Nature Neuroscience*, 19(6):792–794, Apr 2016. ISSN 1546-1726.
doi: [10.1038/nn.4291](https://doi.org/10.1038/nn.4291). URL <http://dx.doi.org/10.1038/nn.4291>.
- [3] Thomas J. Davidson, Fabian Kloosterman, and Matthew A. Wilson. Hippocampal re-
play of extended experience. *Neuron*, 63(4):497–507, Aug 2009. ISSN 0896-6273. doi:
[10.1016/j.neuron.2009.07.027](https://doi.org/10.1016/j.neuron.2009.07.027). URL [http://dx.doi.org/10.1016/j.](http://dx.doi.org/10.1016/j.neuron.2009.07.027)
[neuron.2009.07.](http://dx.doi.org/10.1016/j.neuron.2009.07.027)
[027](http://dx.doi.org/10.1016/j.neuron.2009.07.027).
- [4] Talia L. Retter, Fang Jiang, Michael A. Webster, and Bruno Rossion. Dissociable effects
of inter-stimulus interval and presentation duration on rapid face categorization. *Vision*
*Research*, 145:11–20, Apr 2018. ISSN 0042-6989. doi: [10.1016/j.visres.2018.02.009](https://doi.org/10.1016/j.visres.2018.02.009). URL
<http://dx.doi.org/10.1016/j.visres.2018.02.009>.
- [5] Amanda K. Robinson, Tjil Grootswagers, and Thomas A. Carlson. The influence of image
masking on object representations during rapid serial visual presentation. *NeuroImage*,
197:224–231, Aug 2019. ISSN 1053-8119. doi: [10.1016/j.neuroimage.2019.04.050](https://doi.org/10.1016/j.neuroimage.2019.04.050). URL
<http://dx.doi.org/10.1016/j.neuroimage.2019.04.050>.
- [6] Rachel I. Wilson and Roger A. Nicoll. Endogenous cannabinoids mediate retrograde sig-
nalling at hippocampal synapses. *Nature*, 410(6828):588–592, Mar 2001. ISSN 1476-4687.
doi: [10.1038/35069076](https://doi.org/10.1038/35069076). URL <http://dx.doi.org/10.1038/35069076>.
- [7] Masaya Misaki, Wen-Ming Luh, and Peter A. Bandettini. Accurate decoding of sub-TR
timing differences in stimulations of sub-voxel regions from multi-voxel response patterns.
*NeuroImage*, 66:623 – 633, 2013. ISSN 1053-8119. doi: [10.1016/j.neuroimage.2012.10.069](https://doi.org/10.1016/j.neuroimage.2012.10.069).
URL <http://www.sciencedirect.com/science/article/pii/S1053811912010737>.
- [8] Nicolas W Schuck and Yael Niv. Sequential replay of nonspatial task states in the human
hippocampus. *Science*, 364(6447):eaaw5181, 2019. doi: [10.1126/science.aaw5181](https://doi.org/10.1126/science.aaw5181).
- [9] Stefan Haufe, Frank Meinecke, Kai Görden, Sven Dähne, John-Dylan Haynes, Benjamin
Blankertz, and Felix Bießmann. On the interpretation of weight vectors of linear models in
multivariate neuroimaging. *NeuroImage*, 87:96–110, Feb 2014. ISSN 1053-8119. doi:
[10.1016/j.neuroimage.2013.10.067](https://doi.org/10.1016/j.neuroimage.2013.10.067). URL [http://dx.doi.org/10.1016/j.](http://dx.doi.org/10.1016/j.neuroimage.2013.10.067)
[neuroimage.](http://dx.doi.org/10.1016/j.neuroimage.2013.10.067)
[2013.10.067](http://dx.doi.org/10.1016/j.neuroimage.2013.10.067).

- [10] Miriam E Weaverdyck, Matthew D Lieberman, and Carolyn Parkinson. Tools of the
trade multivoxel pattern analysis in fMRI: A practical introduction for social and affective
neuroscientists. *Social Cognitive and Affective Neuroscience*, 15(4):487–509, Apr 2020.
ISSN 1749-5024. doi: [10.1093/scan/nsaa057](https://doi.org/10.1093/scan/nsaa057). URL [http://dx.doi.org/10.1093/scan/](http://dx.doi.org/10.1093/scan/nsaa057)
[nsaa057](http://dx.doi.org/10.1093/scan/nsaa057).
- [11] Chia-Yueh Carlton Chu. *Pattern recognition and machine learning for magnetic resonance*
*images with kernel methods*. PhD thesis, UCL (University College London), 2009. URL
<https://discovery.ucl.ac.uk/id/eprint/18519>.
- [12] Hans P. Op de Beeck. Against hyperacuity in brain reading: Spatial smoothing does
not hurt multivariate fMRI analyses? *NeuroImage*, 49(3):1943–1948, Feb 2010. ISSN
1053-8119. doi: [10.1016/j.neuroimage.2009.02.047](https://doi.org/10.1016/j.neuroimage.2009.02.047). URL [j.neuroimage.2009.02.047](http://dx.doi.org/10.1016/
.
- [15] Anna Gardumi, Dimo Ivanov, Lars Hausfeld, Giancarlo Valente, Elia Formisano, and
Kâmil Uludağ. The effect of spatial resolution on decoding accuracy in fMRI multi-
variate pattern analysis. *NeuroImage*, 132:32–42, May 2016. ISSN 1053-8119. doi:
[10.1016/j.neuroimage.2016.02.033](https://doi.org/10.1016/j.neuroimage.2016.02.033). URL [2016.02.033](http://dx.doi.org/10.1016/j.neuroimage.
.
- [17] J. D. Swisher, J. C. Gatenby, J. C. Gore, B. A. Wolfe, C.-H. Moon, S.-G. Kim, and
F. Tong. Multiscale pattern analysis of orientation-selective activity in the primary vi-
sual cortex. *Journal of Neuroscience*, 30(1):325–330, Jan 2010. ISSN 1529-2401. doi:
[10.1523/jneurosci.4811-09.2010](https://doi.org/10.1523/JNEUROSCI.4811-09.2010). URL [09.2010](http://dx.doi.org/10.1523/JNEUROSCI.4811-
.
- [19] Dietrich Samuel Schwarzkopf and Geraint Rees. Pattern classification using functional
magnetic resonance imaging. *Wiley Interdisciplinary Reviews: Cognitive Science*, 2(5):
568–579, Mar 2011. ISSN 1939-5078. doi: [10.1002/wcs.141](https://doi.org/10.1002/wcs.141). URL [http://dx.doi.org/](http://dx.doi.org/10.1002/wcs.141)
[10.1002/wcs.141](http://dx.doi.org/10.1002/wcs.141).
- [20] Lukas Kunz, Lorena Deuker, Hui Zhang, and Nikolai Axmacher. Tracking human engrams
using multivariate analysis techniques. In *Handbook of Behavioral Neuroscience*, volume 28,
pages 481–508. Elsevier, 2019. doi: [10.1016/b978-0-12-812028-6.00026-4](https://doi.org/10.1016/b978-0-12-812028-6.00026-4). URL [http://](http://dx.doi.org/10.1016/B978-0-12-812028-6.00026-4)
dx.doi.org/10.1016/B978-0-12-812028-6.00026-4.
- [21] James V. Haxby, M. Ida Gobbini, Maura L. Furey, Almit Ishai, Jennifer L. Schouten,
and Pietro Pietrini. Distributed and overlapping representations of faces and objects in
ventral temporal cortex. *Science*, 293(5539):2425–2430, Sep 2001. ISSN 1095-9203. doi:
[10.1126/science.1063736](https://doi.org/10.1126/science.1063736). URL <http://dx.doi.org/10.1126/science.1063736>.
- [22] Martin J. Chadwick, Heidi M. Bonnici, and Eleanor A. Maguire. Decoding information in
the human hippocampus: A user’s guide. *Neuropsychologia*, 50(13):3107–3121, Nov 2012.
ISSN 0028-3932. doi: [10.1016/j.neuropsychologia.2012.07.007](https://doi.org/10.1016/j.neuropsychologia.2012.07.007). URL [http://dx.doi.org/](http://dx.doi.org/10.1016/j.neuropsychologia.2012.07.007)
[10.1016/j.neuropsychologia.2012.07.007](http://dx.doi.org/10.1016/j.neuropsychologia.2012.07.007).
- [23] Tal Yarkoni, Russell A Poldrack, Thomas E Nichols, David C Van Essen, and Tor D
Wager. Large-scale automated synthesis of human functional neuroimaging data. *Nature*
*Methods*, 8(8):665–670, Jun 2011. ISSN 1548-7105. doi: [10.1038/nmeth.1635](https://doi.org/10.1038/nmeth.1635). URL [http://](http://dx.doi.org/10.1038/nmeth.1635)
dx.doi.org/10.1038/nmeth.1635.
- [24] T Yarkoni, RA Poldrack, TE Nichols, DC Van Essen, and TD Wager. Neurosynth, 2019.
URL <https://neurosynth.org/>.

Reviewer #1 (Remarks to the Author):

All of my concerns have been addressed.

I thank the authors for their thorough revision and a promising new technique for cognitive and systems neuroscience.

Reviewer #2 (Remarks to the Author):

Thank you for your quite thorough and more than satisfying responses to my concerns.

**Response to reviews**

**- Final revision -**

**Manuscript NCOMMS-20-12009A-Z**

*Faster than thought: Detecting sub-second activation*
*sequences with sequential fMRI pattern analysis*

Lennart Wittkuhn^{1,2*} and Nicolas W. Schuck^{1,2*}

¹Max Planck Research Group NeuroCode, Max Planck Institute for Human Development, Berlin, Germany

²Max Planck UCL Centre for Computational Psychiatry and Ageing Research, Berlin, Germany

*Correspondence to {schuck, wittkuhn}@mpib-berlin.mpg.de

**Overview**

We first reproduce each comment in blue. Our answers will follow immediately in black.

**Answers to Reviewer 1**

**Remarks to the Author**

*“All of my concerns have been addressed. I thank the authors for their thorough*
*revision and a promising new technique for cognitive and systems neuroscience.”*

We thank Reviewer 1 for a thorough review of our manuscript and are glad to hear that all
concerns have been addressed by our revision.

**Answers to Reviewer 2**

**Remarks to the Author**

*“Thank you for your quite thorough and more than satisfying responses to my con-*
*cerns.”*

We thank Reviewer 2 for a thorough review of our manuscript and are glad to hear that the
reviewer found our responses more than satisfying.